# Global aerosol simulations using NICAM.16 on a 14-km grid spacing for a climate study: Improved and remaining issues relative to a lower-resolution model

Daisuke Goto[1], Yousuke Sato[2,3], Hisashi Yashiro[1,3], Kentaroh Suzuki[4], Eiji Oikawa[5], Rei Kudo[6],

5   Takashi M. Nagao[4], Teruyuki Nakajima[1]

[1]National Institute for Environmental Studies, Tsukuba, Japan

[2]Faculty of Science, Hokkaido University, Sapporo, Japan

[3]RIKEN Center for Computational Research, Kobe, Japan

[4]Atmosphere and Ocean Research Institute, University of Tokyo, Kashiwa, Japan

10   [5]Research Institute for Applied Mechanics, Kyushu University, Kasuga, Japan

[6]Meteorological Research Institute, Tsukuba, Japan

*Correspondence to*: Daisuke Goto (goto.daisuke@nies.go.jp)

**Abstract.** High-performance computing resources allow us to conduct numerical simulations with a horizontal grid spacing that is sufficiently high to resolve cloud systems on a global scale, and high-resolution models (HRMs) generally provide better simulation performances than low-resolution models (LRMs). In this study, we execute a next-generation model that is capable of simulating global aerosols using version 16 of the nonhydrostatic icosahedral atmospheric model (NICAM.16). The simulated aerosol distributions are obtained for 3 years with an HRM using a global 14-km grid spacing, an unprecedentedly high horizontal resolution and long integration period. For comparison, a NICAM with a 56-km grid spacing is also run as an LRM, although this horizontal resolution is still high among current global aerosol climate models. The comparison elucidated that the differences in the various variables of meteorological fields, including the wind speed, precipitation, clouds, radiation fluxes and total aerosols, are generally within 10% of their annual averages, but most of the variables related to aerosols simulated by the HRM are slightly closer to the observations than are those simulated by the LRM. Upon investigating the aerosol components, the differences in the water-insoluble black carbon and sulfate concentrations between the HRM and LRM are large (up to 32%), even in the annual averages. This finding is attributed to the differences in the aerosol wet deposition flux, which is determined by the conversion rate of cloud to precipitation, and the difference between the HRM and LRM is approximately 20%. Additionally, the differences in the simulated aerosol concentrations at polluted sites during polluted months between the HRM and LRM are estimated with normalized mean biases of -19% for black carbon (BC), -5% for sulfate and -3% for the aerosol optical thickness (AOT). These findings indicate that the impacts of higher horizontal grid spacings on model performance for secondary products such as sulfate, and complex products such as the AOT, are weaker than those for primary products, such as BC. On a global scale, the subgrid variabilities in the simulated AOT and cloud optical thickness (COT) in the 1°×1° domain using 6-hourly data are estimated to be 28.5% and 80.0%, respectively, in the HRM, whereas the corresponding differences are 16.6% and 22.9% in the LRM. Over the Arctic, both the HRM and the LRM generally reproduce the observed aerosols, but the largest difference in the surface BC mass concentrations between the HRM and LRM reaches 30% in spring (the HRM-simulated results are closer to the observations). The vertical distributions of the HRM- and LRM-simulated aerosols are generally close to the measurements, but the differences between the HRM and LRM results are large above a height of approximately 3 km, mainly due to differences in the wet deposition of aerosols. The global annual averages of the effective radiative forcings due to aerosol-radiation and aerosol-cloud interactions (ERFari and ERFaci)

attributed to anthropogenic aerosols in the HRM are estimated to be $-0.293\pm0.001$ W m$^{-2}$ and $-0.919\pm0.004$ W m$^{-2}$, respectively, whereas those in the LRM are $-0.239\pm0.002$ W m$^{-2}$ and $-1.101\pm0.013$ W m$^{-2}$. The differences in the ERFari between the HRM and LRM are primarily caused by those in the aerosol burden, whereas the differences in the ERFaci are primarily caused by those in the cloud expression and performance, which are attributed to the grid spacing. The analysis of interannual variability revealed that the difference in reproducibility of both sulfate and carbonaceous aerosols at different horizontal resolution is greater than their interannual variability over 3 years, but those of dust and sea salt AOT and possibly clouds were the opposite. Because at least ten times the computer resources are required for the HRM (14-km grid) compared to the LRM (56-km grid), these findings in this study help modelers decide whether the objectives can be achieved using such higher resolution or not under the limitation of available computational resources.

## 1 Introduction

High-performance computing resources allow us to conduct numerical simulations with a horizontal grid spacing that is sufficiently fine to resolve cloud systems on a global scale. Suzuki et al. (2008) first performed a high-resolution global simulation while explicitly treating aerosol-cloud interactions (ACIs) and reproduced the interactions obtained from satellite measurements. For the past 10 years, various high-resolution models (HRMs) have been developed to address the heretofore unresolved mechanisms related to cloud processes; one example of a related outcome is the buffered system hypothesis (e.g., Stevens and Feingold, 2009; Malavelle et al., 2017). When modeling atmospheric pollutants, such as aerosols and short-lived gases, HRMs are believed to provide a better simulation performance than low-resolution models (LRMs). For example, Qian et al. (2010) showed that simulations of the trace gases and aerosols in the vicinity of Mexico City in March with a 3-km horizontal resolution are far more advantageous than simulations with 15-km and 75-km horizontal resolutions; this indicates that a high-resolution horizontal grid can resolve local emissions and terrain-induced flows along mountain ridges. Similarly, Ma et al. (2014) identified that the aerosols and clouds simulated over the Arctic in April at the finest resolution (10 km) are closer to the observations than those simulated at a coarser resolution (ranging from 20 km to 160 km). In addition, using a global model with a horizontal resolution varying from 3.5 km to 56 km, Sato et al. (2016) showed that fine-resolution grids can more realistically resolve low-pressure systems with vortexes at mid-latitudes, which result in the realistic transport of black

carbon (BC) to the Arctic in November, than can coarse-resolution grids. On a global scale, Sekiya et al. (2018) employed a global chemical transport model with an integration period of 1 year and provided a more realistic distribution of short-lived gaseous $NO_2$, especially in urban areas, with a horizontal resolution of approximately 60 km (0.56°×0.56°) than with horizontal resolutions of approximately 110 km and 300 km (1.1°×1.1° and 2.8°×2.8°). Furthermore, Schutgens et al. (2016) investigated the subgrid variability of simulated aerosols with a 10-km resolution in various domains and noted the importance of a fine grid sizes, and Goto et al. (2016) showed that 10-km grid simulations around Japan over an integration period of 3 years require a regional HRM to properly reproduce the concentrations of aerosols because such high concentrations in urban areas create health concerns for many people (Ezzati et al., 2002). The studies mentioned above focused on atmospheric pollutants and discussed the advantages of HRMs at various scales and among different seasons; nevertheless, with only a few exceptions, the models were not executed with horizontal grids finer than 50 km for adequately long periods on a global scale. For instance, Hu et al. (2018) successfully applied the Goddard Earth Observing System (GEOS)-Chem model with a 12.5-km horizontal grid to simulate aerosols and short-lived gases, and Sato et al. (2018) clarified the advantages of an HRM using a nonhydrostatic icosahedral atmospheric model (NICAM) with a 14-km horizontal grid to resolve ACIs. However, these two studies focused on study periods of just 1 year. The 1-year calculation cannot provide yearly variability; thus, clarifying whether the differences in the simulated results between HRMs and LRMs are caused by a difference in horizontal resolution or meteorological fluctuations among years is difficult. The NASA GEOS-5 aerosol forecasting system has been running at these high resolutions for several years (e.g., Gelaro et al., 2015), but to our knowledge, the published literature does not explain the difference in the simulated results between the HRM and LRM. As such, the merits of using HRMs with horizontal grid resolutions finer than 50 km to simulate aerosols in global and climatological fields remain ambiguous. Thus, it is very important to clarify this issue and to provide scientific evidence for our future; to achieve this goal, global calculations of air pollutants must be performed with HRMs using horizontal grids finer than 10 km.

Therefore, in this study, we investigate how much relatively high-resolution grids can improve the simulation results of aerosols and their interactions with clouds and radiation fluxes for climatological fields. For this purpose, we executed a NICAM with aerosol components on a 14-km horizontal grid for 3 years. This 14-km horizontal grid boasts the finest resolution among all global chemistry models and is generally finer than most regional chemistry models (Galmarini et al., 2018). To effectively

show the advantages in the simulated parameters related to aerosols in the HRM with a 14-km horizontal grid, we also executed an LRM with a 56-km horizontal grid, which is still finer than most global aerosol climate models (Myhre et al., 2013; Galmarini et al., 2018) but coarser than some of those used for operational global aerosol forecasting (Sessions et al., 2015). Some issues are still under debate in global aerosol models. For example, how well are atmospheric pollutants over the Arctic

reproduced (e.g., Shindell et al., 2008)? In addition, why do most global models overestimate BC (and possibly other species) in the middle and high troposphere over the remote ocean (e.g., Schwarz et al., 2013)? Finally, what are the aerosol radiative forcing (ARF) values estimated through aerosol-radiation interactions (ARIs) and ACIs using global cloud-system resolving models? Furthermore, it is also important to quantify the differences caused by the horizontal grid spacing or yearly variability of the meteorological fields.

In this paper, the models and observation datasets are described in section 2. Section 3 demonstrates the results of using the NICAM coupled to an aerosol module and compares the results with multiple measurements. The first part of section 3 illustrates the global distributions of meteorological fields such as winds, precipitation, clouds, and radiation, while the second part shows the results of evaluations with the HRM and LRM using multiple aerosol measurements. In section 4, the effects of different grid spacings on the aerosol fields, model evaluations over the Arctic, ARFs, interannual variabilities

over 3 year integration and required computational resources are discussed. Section 5 provides the summary of this work and the implications for future research on HRMs in the context of powerful computational resources.

## 2 Model descriptions and experimental design

### 2.1 NICAM

Aerosol simulations were performed with a nonhydrostatic icosahedral atmospheric model (NICAM) with a uniform grid

system (Tomita and Satoh, 2004; Satoh et al. 2008; 2014). The NICAM was executed with unprecedentedly high resolutions, namely, 0.87 km for 1 week (Miyamoto et al., 2012) and 14 km for 25 years under Atmospheric Model Intercomparison Project (AMIP)-like experiments (Kodama et al., 2015), although these studies did not consider aerosols. Subsequently, Suzuki et al. (2008) first conducted a global 7-km integration of aerosols for 1 week in July 2006 and validated the simulated

ACIs by comparing them with satellite measurements. Sato et al. (2016) performed a global 3.5-km integration of aerosols for 2 weeks in November 2011 to focus on the transport and deposition of BC over the Arctic. Jing et al. (2017) and Sato et al. (2018) analyzed cloud microphysics parameters simulated using a NICAM with aerosol components and a 14-km grid spacing for 1 year in 2012. Additionally, to analyze the transport of a simulated tracer in an HRM, Ishijima et al. (2018)

calculated a radon tracer that has a long lifetime in the atmosphere using a NICAM with a 14-km horizontal resolution for 3 years. However, these studies did not elucidate the distributions of the aerosol components on a global scale for more than 1 year. Therefore, the present study extends these studies by simulating aerosol components for 3 years to discuss them climatologically.

The NICAM, which corresponds to a dynamic core, simulates the basic prognostic variable, such as air temperature, wind,

water vapor, cloud, precipitation and radiation fluxes, by calculating different processes, such as advection and diffusion, and the corresponding physics. In this study, the NICAM developed in 2016 was used as NICAM.16. The options to use modules for these calculations in running the NICAM with a 14-km resolution are almost similar to those used in Kodama et al. (2015). The advection module is based on Miura (2007) and Niwa et al. (2011), and the diffusion module is the level-2 Mellor-Yamada-Nakanishi-Niino (MYNN) scheme (Mellor and Yamada, 1972; Nakanishi and Niino, 2004). The module for

calculating the land surface flux is the Minimal Advanced Treatments of Surface Interaction and Runoff (MATSIRO) model with boundary conditions, such as the land cover type, soil type, leaf area index and ground albedo (Takata et al., 2003). The Model Simulation Radiation Transfer code (MSTRN-X), which is based on the k-distribution scheme, is adopted for the radiation model to calculate the radiative fluxes by considering the scattering, absorption and emissivity of aerosols and clouds and their absorption by gases (Sekiguchi and Nakajima, 2008). The MSTRN-X also calculates the global, direct and

diffuse solar fluxes. The cloud microphysics module is the NICAM Single-Moment Water 6 (NSW6) scheme (Tomita, 2008), which prognoses the single-moment bulk amounts of 6 categorized hydrometeors, i.e., water vapor, cloud water, rain, cloud ice, hail and graupel. Cloud water and rain are fully interactive with cloud condensation nuclei (CCN), which are online calculated by the parameterization of Abdul-Razzak and Ghan (2000) as an indirect aerosol effect or aerosol-cloud interaction. The parameterization of aerosol activation considers the updraft velocity, aerosol sizes and aerosol chemical

compositions. Even in an HRM, the updraft velocity tends to be small; therefore, the updraft velocity is also parameterized

by the formulation proposed by Lohmann et al. (1999) using turbulent kinetic energy, and the minimum value of the updraft velocity is set to 0.1 m s$^{-1}$ (Ghan et al., 1997). The minimum number of CCN is set at 25 cm$^{-3}$, as defined in the previous studies (Jing et al., 2017; Sato et al., 2018). Under high-resolution horizontal grid simulations, a NICAM does not generally adopt a cumulus parameterization or define the cloud fraction (e.g., Satoh et al., 2010; Goto et al., 2015a; Goto et al., 2019).

This study defines a cloud frequency, which is set to 1 when (1) the cloud liquid water content exceeds 10$^{-3}$ kg m$^{-3}$, (2) the cloud optical thickness (COT) exceeds 0.2 or (3) the cloud ice water content exceeds 10$^{-3}$ kg m$^{-3}$, otherwise it is set to zero. Under the above conditions as well as the condition where the sum of all hydrometers except water vapor exceeds 10$^{-4}$ kg m$^{-3}$, the warm cloud frequency is set to 1 (Sato et al., 2018). The autoconversion rate from cloud to raindrops is parameterized by Berry (1967). The simulated relationship between cloud and precipitation with a 14-km grid spacing has already been

thoroughly evaluated in previous studies (Jing et al., 2017; Sato et al., 2018). In the sensitivity experiments for a comparison of aerosol mass concentrations over the Arctic in section 4.2, a cloud macrophysics module containing both a large-scale cloud condensation (Le Treut and Li, 1991) instead of the NSW6 cloud microphysics scheme and a cumulus parameterization (Chikira and Sugiyama, 2010) are adopted in the NICAM with 56-km and 220-km grid spacings. Hereafter, the sensitivity experiments are called low-resolution model (56-km) with the macrophysics module (LRM-macro) and very

low-resolution model (220-km) with the macrophysics module (VLRM-macro). The VLRM-macro results have been evaluated against measurements in previous studies (Dai et al., 2014; Goto et al., 2015b; Dai et al., 2018).

## 2.2 Aerosol module

The aerosol module, based on the Spectral Radiation-Transport Model for Aerosol Species (SPRINTARS) (Takemura et al., 2005), was implemented in NICAM by Suzuki et al. (2008), and the results have been sufficiently validated through

previous studies on a global scale with low-resolution (approximately 200 km) horizontal grids (Dai et al., 2014; Goto et al., 2015b; Dai et al., 2018) and on the regional scale with high-resolution (10-25 km) horizontal grids (Goto et al., 2015a; Goto et al., 2016; Goto et al., 2019); moreover, the results have been validated on a global scale with horizontal grids at high resolutions (ranging from 3.5 km to 14 km) but over a relatively short period of less than 1 month (Suzuki et al., 2008; Sato et al., 2016; Goto et al., 2017). The use and applications of this module are summarized in Goto et al. (2018). The aerosol

module considers major tropospheric aerosol species, i.e., BC, particulate organic matter (POM), sulfate, dust and sea salt. BC is a primary particle that is emitted from anthropogenic sources and biomass burning. One-half of all BC particles emitted from anthropogenic sources are assumed to be hydrophobic, whereas the remainder are assumed to be hydrophilic as internally mixed particles with POM without any atmospheric aging (Takemura et al., 2005). These emitted aerosols are

transported, diffused and removed through wet deposition in and below clouds by precipitation, dry deposition and gravitational settling, which are described elsewhere in the literature (e.g., Goto et al., 2015a; Goto et al., 2019). Especially, for the wet deposition of aerosols, the previous versions of the global climate model with a coarse resolution were updated to adapt to various assumptions to produce simulations with a finer resolution (Goto et al., 2019). In the wet deposition aerosols coexist in both interstitial and inside clouds, and the interstitial fractions of aerosols are tuning parameters and, in this study,

set at 0.5 for dust, 0.2 for sea salt, 0.5 for all POM, 0.9 for external BC and 0.5 for sulfate. The secondary aerosol sulfate (the main secondary aerosol considered in this study) is formed from chemical reactions, namely, the oxidation of $SO_2$ by OH, ozone and $H_2O_2$ in the atmosphere. The three-dimensional distribution of these oxidants is prescribed from the results of a chemical transport model, namely, the chemical atmospheric general circulation model (AGCM) for study of atmospheric environment and radiative forcing (CHASER), coupled to a conventional GCM named the Model for Interdisciplinary

Research on Climate (MIROC) (Sudo et al., 2002). The sizes of dust and sea salt are divided into 10 bins (the centers are from 0.13 μm to 8.02 μm) and 4 bins (the centers are from 0.178 μm to 5.62 μm), respectively, whereas those of BC, POM and sulfate are assumed to be monomodal with single fixed sizes (the radii are 0.1 μm for internally mixed BC with POM, 0.08 μm for secondary organic aerosols (SOA), 0.054 μm for external BC and 0.0695 μm for sulfate) and the width (1.8 for internally mixed BC with POM, 1.8 for SOA, 1.53 for external BC and 2.03 for sulfate). The sizes and widths are referred

from Hess et al. (1998), Moteki et al. (2007) and Goto et al. (2008). For internally mixed BC with POM, SOA, sulfate and sea salt, i.e., hygroscopic particles, the sizes are functions of the relative humidity (RH) (e.g., Table 2 in Goto et al., 2011). For all aerosols, their optical products, i.e., their extinction coefficient and AOT, are calculated by their mass concentrations and properties, such as size, RH and refractive index according to Mie scattering (Sekiguchi and Nakajima, 2008). These optical parameters at a wavelength of 550 nm are evaluated by measurements. The refractive indexes are 1.53-0.0055i for

dust, $1.50-10^{-8}i$ for sea salt, $1.43-10^{-8}i$ for sulfate, 1.53-0.006i for pure POM and 1.75-0.44i for pure BC (Dai et al., 2014).

The refractive indexes for internally mixed BC with POM are calculated by the volume-weight average. All parameters used in the HRM aerosol module also apply to those used in the LRM aerosol module. To evaluate the aerosol direct effect in the NICAM, the instantaneous radiative forcing of the ARI (IRFari) is online calculated by the difference in the radiative fluxes with/without aerosol species in MSTRN-X; the effective radiative forcing of the ARI due to anthropogenic aerosols (ERFari) is also calculated by the difference in the radiative fluxes between two different experiments with/without anthropogenic aerosol species (but the emissions from biomass burning do not change in our assumption). In these two experiments with/without anthropogenic sources, the effective radiative forcing of the ACI due to anthropogenic aerosols (ERFaci) is also calculated by the difference in the cloud radiative fluxes; the method for calculating the ERFaci as an ERFari is derived from Ghan (2013). Unfortunately, the calculations of the ERFaci under the preindustrial era and the IRFari associated with each aerosol component under the present era are performed for only one year because of limitations of available computer resources. Therefore, the ERFaci value and the IRFari value associated with each aerosol component are calculated using the one-year integration

## 2.3 Experimental design

Numerical experiments with the HRM (14-km horizontal grid) are carried out for 3 years, and experiments are also carried out with the LRM (56-km horizontal grid) for the same period. In both the HRM and the LRM, the number of vertical layers is set at 38, which is relatively small but has been used in previous studies (Kodama et al., 2015; Sato et al., 2016; Sato et al., 2018). The heights of the layers are 80.8 m at the bottom to 36.7 km at the top of the model domain; 10 layers are used below a height of approximately 2 km. The timestep is set at 1 minute in both the HRM and the LRM, and the initial conditions are prepared by the meteorological fields estimated from the National Centers for Environmental Prediction (NCEP)-Final (FNL) (Kalnay et al, 1996) data on November 2011 for the model spinup. The analysis is initiated at the beginning of January 2012 and terminates at the end of December 2014. The model runs without nudging the meteorological fields, i.e., in a free run. The sea surface temperature (SST) and sea ice are nudged by the results of the NICAM from Kodama et al. (2015).

The emission amounts of total BC were 5.6 Tg yr$^{-1}$ from anthropogenic sources in 2010 according to the Hemispheric Transport of Air Pollution (HTAP)-v2.2 emission inventory (Janssen-Maenhout et al., 2015) and a climatological average of 1.8 Tg yr$^{-1}$ from biomass burning over 2005-2014 from the Global Fire Emission Database version 4 (GFEDv4; van der Werf et al., 2017). The interannual variabilities of the emission from the biomass burning are shown in the supplemental figures (Figures S1 and S2) to show impacts of the climatological averages on the results in a specific year, indicating that the impacts can be mostly ignored with only a few exceptions: AOT over Canada and Siberia in 2012-2014 average (mainly section 3.2) and BC mass concentrations over the Pacific and over the Arctic in March-April 2008 (section 4.3). The injection height is set at the surface for anthropogenic sources and 1 km for biomass burning in this study. Organic carbon (OC) is composed of both primary and secondary components; the emission amounts of primary OC were 20.3 Tg yr$^{-1}$ for anthropogenic sources (HTAP-v2.2) and 39.7 Tg yr$^{-1}$ from biomass burning (GFEDv4). These OC values are converted by multiplying the corresponding values for particulate organic matter (POM) by 1.6 for anthropogenic sources and 2.6 for biomass burning sources, whose values are used in several global aerosol models (Tsigaridis et al., 2014). SOAs are assumed to be particles by multiplying the emission fluxes of isoprene and terpenes provided by the Global Emissions Initiative (GEIA) (Guenther et al., 1990) using scaling factors. As a result, the amount of emitted SOAs was 22.2 Tg yr$^{-1}$, which is comparable to the best estimates from recent studies (Tsigaridis et al., 2014). Sulfate is a secondary species formed from a precursor of SO$_2$, of which 108.1 Tg yr$^{-1}$ is emitted from anthropogenic sources (HTAP-v2.2), 2.2 Tg yr$^{-1}$ is emitted from biomass burning (GFEDv4), and 3.1 Tg yr$^{-1}$ is emitted from volcanic eruptions (Diehl et al., 2012). Some SO$_2$ is formed from dimethyl sulfide (DMS), which is mainly emitted from oceans and is calculated as a function of downward solar fluxes (Bates et al., 1987); it is estimated to be 26.2 Tg yr$^{-1}$ (HRM) and 24.9 Tg yr$^{-1}$ (LRM) in this study. Dust and sea salt are primary particles, which are calculated inside the model using the wind speed at a height of 10 m. The emission flux of dust depends on the cube of the wind speed and empirical coefficients, which are determined by the soil moisture, the land use, snow cover and tuning coefficients depending on the region (Takemura et al., 2000). The tuning parameters used in the HRM also apply to those used in the LRM. Over the sea surface without sea ice, the emission flux of sea salt depends on a power of 3.41 (Monahan et al., 1986), which is comparable to the best estimate of 3.5 (Grythe et al., 2014). The estimated emission fluxes for dust and sea salt are shown in section 3. In the preindustrial era to estimate both ERFari and ERFaci, the

emission fluxes from anthropogenic sources and biomass burning for BC, POM and $SO_2$ are set to zero, but those from other sources, i.e., all natural sources, are identical to those used in the present era.

## 2.4 Data description

Table 1 summarizes the measurements used in this study for the model evaluation. Satellite observations greatly assist in better understanding the global model performance of optical properties. The Moderate Resolution Imaging Spectroradiometer (MODIS), a sensor on board the polar-orbiting satellites Terra and Aqua, observes both aerosols and clouds. The cloud products, i.e., COT only for warm-topped clouds and cloud fraction (CF) for all types of clouds, and the aerosol products, i.e., AOT, in collection 6 are retrieved with a grid of $1° \times 1°$ by a NASA algorithm (Platnick et al., 2015). For clouds, the MODIS-retrieved COT has some positive biases especially in high latitudes, due to high solar zenith angle (Grosvenor and Wood, 2014; Lebsock and Su, 2014). For the AOT, the combination method of Dark Target (DT) and Deep Blue (DB) is used and can retrieve AOTs even over the desert areas (Levy et al., 2013), but it does not retrieve AOTs over high-albedo areas covered by snow and some specific areas, which include caatinga/cerrado surfaces over eastern Brazil in June-July-August and over Australia in all seasons (Sayer et al., 2014). In addition, the vertical profiles of the aerosol extinction coefficients are derived from Cloud‑Aerosol Lidar with Orthogonal Polarization CALIOP/Cloud‑Aerosol Lidar and Infrared Pathfinder Satellite Observations (CALIPSO) version 3 provided by the NASA Langley Research Center (LaRC) after an averaging operation with a grid of $1° \times 1°$ under clear-sky conditions (Winker et al., 2013). The top-of-atmosphere (TOA) radiation fluxes, i.e., outgoing shortwave radiation (OSR), outgoing longwave radiation (OLR) and shortwave and longwave cloud radiative forcing (CRF), prepared by the CERES_EBAF_Ed2.8 level-3 product are obtained from a sensor of the Clouds and the Earth's Radiant Energy System (CERES) experiment onboard Terra and Aqua using $1° \times 1°$ grids by considering the diurnal variations of clouds (Loeb et al., 2009). The Baseline Surface Radiation Network (BSRN) observes surface radiation fluxes at sites worldwide (Ohmura et al., 1998). The data collected by the BSRN cover the period of 2008-2012; these data are climatologically averaged while considering missing data and are then converted to the selected 25 sites (surface solar radiation) and 20 sites (direct and diffuse radiations) in this study. The reanalyzed wind at

a height of 10 m, which is important for analyzing the emissions of dust and sea salt in the NICAM, is prepared in a product with a grid of 2.5°×2.5° by the NCEP/National Center for Atmospheric Research (NCAR) reanalysis 1 (Kalnay et al., 1996). Precipitation, which directly causes the wet removal of aerosols, is compared with a product provided by the reanalysis data of the Global Precipitation Climatology Project (GPCP) (Adler et al., 2003). The abovementioned measurements can provide

a global map of the horizontal distributions of these parameters, whereas the following measurements are performed at ground-based sites spread around the world (Figure 1). The Aerosol Robotic Network (AERONET) (Holben et al., 1998) and SKYNET radiometer network (Nakajima et al., 1996) observe the AOT at sites worldwide, but only 135 AERONET sites and 5 SKYNET sites are used in this study. For the selection of these data, monthly mean values are calculated using more than 90 samples in one month, and the annual mean values are averaged using more than 7 months of data at each site. The

China Aerosol Remote Sensing Network (CARSNET) also observes the AOT at 50 sites in China (Che et al., 2015) and directly provides climatological values for the period of 2002-2013. The AOT and extinction values are calculated at the wavelength of 550 nm in the NICAM, whereas these values are retrieved at the wavelengths of 550 nm in MODIS, 500 nm AERONET and 532 nm in CALIOP. This study ignores the differences in the AOT values among the wavelengths of 500 nm, 532 nm and 550 nm because the magnitude is small (less than several percent). In addition, we compared the simulated

AOT and aerosol extinction coefficient under all-sky conditions with the satellite-retrieved AOT and coefficients under clear-sky conditions because the differences in the simulated AOT between all-sky and clear-sky conditions are within 0.01 or 10% at a global scale (Figure S3), which is consistent with the previous study (Dai et al., 2015), but it should be noted that regionally the differences reach up to 0.1 over some regions, such as the North Atlantic (Figure S3). The difference would be generally lower than that between the NICAM and satellite results. Aerosol mass concentrations are observed by multiple

networks, namely, the Interagency Monitoring of Protected Visual Environments (IMPROVE) program, European Monitoring and Evaluation Programme (EMEP), Acid Deposition Monitoring Network in East Asia (EANET), China Meteorological Administration Atmosphere Watch Network (CAWNET) and University of Miami network. The IMPROVE-observed BC, POM and sulfate over North America are used at approximately 190 sites, whereas the EMEP-observed BC, POM and sulfate over Europe are employed at approximately 50 sites. Over Asia, the EANET-observed

sulfate is used at only 35 sites, whereas the CAWNET-observed BC, POM and sulfate are used at 14 sites, but only in China

(Zhang et al., 2012). The network managed by a group at the University of Miami releases both dust and sea salt mass concentrations at sites worldwide (e.g., Prospero et al., 1989), but only the 16 sites shown in Liu et al. (2007) are used in this study. BC measurements, especially over the Arctic, are obtained by applying an aethalometer or a particle soot absorption photometer (PSAP), which may include some biases (Sinha et al., 2017; Sharma et al., 2017); nevertheless, these

measurements can still be a good reference for the evaluation of global models. Aircraft measurements of BC using a single-particle soot photometer (SP2; Schwarz et al., 2006) are also used for a one-year model evaluation (January, March, June, August and November) in 2009 over the Pacific Ocean under the High-performance Instrumented Airborne Platform for Environmental Research (HIAPER) Pole-to-Pole Observations (HIPPO) campaign (Schwarz et al., 2010; Wofsy et al., 2012), in March-April 2012 under the Aerosol Radiative Forcing in East Asia (A-FORCE) campaign (Oshima et al., 2012), and in

April-May and July-August 2008 under the Arctic Research of the Composition of the Troposphere from Aircraft and Satellites (ARCTAS) campaign (Jacob et al., 2010). The uncertainties in the observed products used in this study are shown in each reference.

## 3 Results

First, the meteorological fields relevant to the aerosol distribution are compared between the satellite and reanalysis data.

Aerosols are transported in the atmosphere by wind and are removed from the atmosphere mainly by wet deposition associated with precipitation; however, some aerosols, i.e., dust and sea salt, are emitted through surface friction by winds. Therefore, the simulated variables of wind, precipitation and clouds are evaluated. Second, the aerosols simulated by the HRM and LRM are compared with the multiple observations described in section 2.4. When measurements are not available in the model evaluation, only the difference between the HRM and LRM is discussed.

### 3.1 Meteorological fields

Figure 2 illustrates the annual, January and July averages of the wind directions and speeds at a height of 10 m using the HRM-simulated, LRM-simulated and NCEP-reanalyzed winds. The global distribution of the statistical metrics, i.e., Pearson correlation coefficient (PCC), normalized mean bias (NMB) and root-mean-square error (RMSE), for the annually averaged

wind speeds between the NICAM simulations and NCEP-reanalysis data are illustrated in the supplement (Figure S4). The global annual averages of both the HRM-simulated and LRM-simulated wind speeds (approximately 4.2 m s$^{-1}$; 4.169 m s$^{-1}$ for HRM and 4.242 m s$^{-1}$ for LRM) are slightly lower than those of the NCEP-reanalyzed wind speeds (4.487 m s$^{-1}$). The differences in the wind speed between the models and NCEP over land are smaller than those over ocean. The correlations between the NICAM (both the HRM and the LRM) and NCEP are moderate with a PCC of approximately 0.58 (0.577-0.580) for the global averages, whereas the differences in the PCC between land (0.582 for HRM and 0.590 for LRM) and ocean (0.576 for HRM and 0.577 for LRM) are small. The global annual average RMSEs between the NICAM simulations and NCEP are calculated to be approximately 1.45 m s$^{-1}$ (1.446 m s$^{-1}$ for HRM and 1.461 m s$^{-1}$ for LRM), approximately one-third of the global averages. The RMSEs are relatively high over the Southern Ocean (45°S-70°S), with values of at most 5.0 m s$^{-1}$. The NMBs are calculated to be -7.6% (HRM) and -5.8% (LRM) for the global averages. The RMSE and NMB over land are smaller than those over ocean. In Figures 2(a), 2(b) and 2(c), the spatial patterns of the HRM-simulated and LRM-simulated winds are generally in agreement with those obtained from the NCEP-reanalysis data, although there are some differences between the models and the NCEP reanalysis. The difference is predominantly caused by an underestimation over the Southern Ocean (within 45°S-70°S) with lower correlation (PCC), higher uncertainty (RMSE) and more negative bias (NMB) than other areas (Figure S4). More negative NMB values over land are also found in both HRM and LRM (Figure S4(g) and S4(h)), even though the NCEP wind speeds are generally less than 3 m s$^{-1}$ over land. In January and July, the global averages of both the HRM-simulated and the LRM-simulated wind speeds are also lower than those of the NCEP-reanalyzed wind speeds by at most 10%. In January, the differences in the global averages of wind speed between the models and the NCEP reanalysis over land are larger than those over ocean, whereas in July those over land are smaller than those over ocean. In regions where sea salt is dominant over the Southern Ocean in both January and July and dust is the dominant over the Sahara in July, the differences in wind speeds between the HRM and LRM are relatively large. Although we expected the HRM-simulated wind speeds to be higher than the LRM-simulated wind speeds, our results do not confirm this behavior because the wind speed can be influenced by several complex mechanisms, such as clouds and radiation.

The precipitation simulated with the NICAM (both the HRM and the LRM) is generally comparable to the GPCP-reanalyzed precipitation, especially over the mid-latitudes and high latitudes (Figure 3a). The strongest precipitation is found at Intertropical Convergence Zone (ITCZ), where the precipitation simulated by the NICAM is overestimated compared to that reanalyzed by the GPCP, and the HRM-simulated precipitation is closer to the GPCP-reanalyzed precipitation than the

LRM-simulated one. The annual global mean precipitation rates are 2.64 mm day$^{-1}$ (HRM-simulated), 2.81 mm day$^{-1}$ (LRM-simulated) and 2.64 mm day$^{-1}$ (GPCP-reanalyzed); the difference arise primarily because the LRM occasionally provides stronger precipitation along some coastlines than the HRM (supplemental Figures S5(a,b,c)). The HRM precipitation is closer to the GPCP precipitation than that of the LRM. Due to the coarseness of the horizontal grid spacing, the LRM tends to reproduce unrealistically strong convective clouds compared to the HRM. Such convective clouds can lead to strong

precipitation.

The simulated clouds are also evaluated by zonal averages based on a comparison with satellite observations (MODIS/Terra and MODIS/Aqua). Because the cloud liquid water path (LWP) retrieved from satellites is highly uncertain (e.g., Lebsock and Su, 2014) and the simulated LWP is strongly correlated to precipitation (not shown), the comparison of the simulated precipitation shown in Figure 3(a) can be considered one of a validation of cloud parameters. In Figure 3(b), the warm-

topped COTs are shown, and their global averages are estimated to be 7.9 (HRM), 10.2 (LRM), 15.1 (MODIS/Terra) and 15.0 (MODIS/Aqua). The distributions of both the HRM and LRM results are also not very close to the MODIS retrievals (Figure 3(b) and supplemental Figures S5(d,e,f)). The possible reasons are the underestimation of warm-topped COT itself in the NICAM and the overestimation of warm-topped COT in MODIS, especially in high latitudes (Grosvenor and Wood, 2014; Lebsock and Su, 2014). Another possible reason is that a bias of the simulated cloud height in the NICAM. The

differences in warm-topped COT (Figure 3(b)) between thee HRM and LRM are consistent with those of precipitation (Figure 3(a) and supplemental Figures S5(d,e,f)). Figure 3(c) illustrates zonal averages of cloud fraction (CF) for all types of cloud (not just warm-topped clouds). The global averages of the CF are 0.63 (HRM), 0.59 (LRM), 0.74 (MODIS/Terra), and 0.75 (MODIS/Aqua). Both simulated CFs are underestimated compared to the MODIS result, but the HRM results tend to be closer to the MODIS results than the LRM results over low latitudes from 30°S to 30°N as well as high latitudes from 60°N

to 90°N, whereas the LRM results tend to be closer to the MODIS results than the HRM results over higher latitudes from

90°S to 30°S. These differences can be found in the global distribution shown in supplemental Figures S5(g,h,i). Such discrepancy in clouds between global models, including the NICAM and the observations, can be found in previous studies (e.g., Nam et al., 2012; Kodama et al, 2015); therefore, our case also includes some common problems.

For aerosol wet removal, the ratio of precipitation to cloud water (RPCW) is one of the important variables, although this is not a pure ratio of both variables but the conversion ratio from cloud to precipitation. The RPCW at a height of 2 km is calculated online using the model and plotted in Figure 3(d) and supplemental Figures S5(j,k). The global average of the RPCW at a height of 2 km is calculated to be 0.143 (HRM) and 0.181 (LRM), which can be explained by the tendency that the LRM reproduces stronger convective clouds and precipitation, thus providing a quicker conversion from cloud to precipitation in the LRM than in the HRM. Therefore, the wet removal rate in the HRM is slower than that in the LRM. This result is very important for determining the aerosol distributions.

To further evaluate the climatic impacts of clouds on the radiation field in the models, the TOA shortwave radiative fluxes and CRF between the NICAM and the satellite-based results are shown in Figures 3(e) and 3(f). Their global distributions are shown in the supplement (Figure S5(l,m,n,o,p,q)). The relevant TOA parameters for aerosols are the OSR and shortwave CRF (SWCRF) simulated by the HRM and LRM and retrieved by CERES. As shown in Figures 3(e) and 3(f), the global averages of these variables in the LRM appear closer to those in CERES than the averages of the HRM, which is caused by the results over the mid-latitudes from 60°S to 30°S or from 60°N to 30°N, where the CF in the LRM is close to the MODIS results shown in both Figures 3(c) and supplemental Figures S5(l,m,n,o,p,q). Over the low latitudes from 30°S to 30°N, where the CF in the HRM is close to the MODIS results shown in Figures 3(c), the global distributions of the simulated SWCRF and OSR in the HRM are much closer to the those in CERES than those in the LRM shown in Figures 3(e,f) and supplemental Figures S5(g,h,i). Interestingly, these shortwave radiative fluxes in both the HRM and LRM are closer to the fluxes retrieved by CERES than those shown in the NICAM without aerosol components by Kodama et al. (2015). This finding indicates that aerosols and their interactions with clouds primarily affect low-level clouds (mainly water-phase clouds) and provide better results than previous results without aerosols. Such effects were considered in a very recent study by Kodama et al. (2020). Although there are some differences between the NICAM and CERES, these estimates are

generally within multimodel uncertainties (8 W m$^{-2}$ for the SWCRF and up to 11 W m$^{-2}$ for the LWCRF) derived from the current global climate models (Lauer and Hamilton, 2013).

To perform a precise validation of radiative fluxes, the surface shortwave radiative fluxes simulated by the NICAM are evaluated using in situ observations in Figure 4, which illustrates the scatterplots of the surface solar radiation (SSR) and direct and diffuse radiation between the observations and NICAM simulations under all-sky conditions at the ground-based BSRN sites (nearly 20 sites around the world, i.e., North America, Europe, North Africa, Asia, and Oceania). The NICAM-simulated SSRs are very similar to the observations, exhibiting high PCCs (PCC=0.89 in both the HRM and the LRM), a low NMB in both models (NMB=1.1% in the HRM and NMB=-0.3% in the LRM) and low uncertainties signified by small RMSEs (RMSE=-32 W m$^{-2}$ in both the HRM and the LRM). When the SSR is decomposed into direct and diffuse fluxes, however, the NICAM-simulated direct radiation fluxes are overestimated compared to the observations, while the NICAM-simulated diffuse radiation fluxes are underestimated. The correlations of these decomposed radiation fluxes are still high, except for the diffuse radiation in the LRM (PCC=0.63, which is still moderate). Moreover, the biases of the decomposed radiation components are much larger than the bias of the SSR; the NMBs in the direct radiation are 28.2% (HRM) and 26.7% (LRM), whereas those in the diffuse radiation are -18.3% (HRM) and -20.4% (LRM). The differences in the SSR between the HRM and LRM are very small, but the HRM-simulated direct and diffuse radiations are slightly better than the LRM-simulated radiation fluxes. These results at the surface may not be consistent with the results of the clouds and TOA radiation fluxes shown in Figures 3(e) and 3(f), respectively, which is likely because the number of samples from the BSRN in Figure 4 is smaller than that of the satellites in Figures 3(e) and 3(f). In addition, the BSRN sites do not cover the oceans, which cover a considerable proportion of the globe, thereby not exactly being consistent with the global average obtained from the satellites. Nevertheless, considering the model performance, the simulated clouds and radiation fluxes are generally acceptable for use in global aerosol simulations with a climate model.

**3.2 Aerosol fields**

Figure 5 shows the global distributions of the annual, January and July averages of the HRM-simulated, LRM-simulated, MODIS/Aqua-retrieved, and MODIS/Terra-retrieved AOTs. The global annual averages of the HRM-simulated AOT

(0.175) and the LRM-simulated AOT (0.170) are within the differences between the MODIS/Aqua-retrieved AOT (0.163) and MODIS/Terra-retrieved AOT (0.184). The same tendencies are also found over land (0.157 for HRM, 0.152 for LRM, 0.145 for MODIS/Aqua and 0.166 for MODIS/Terra) and ocean (0.227 for HRM, 0.221 for LRM, 0.217 for MODIS/Aqua and 0.234 for MODIS/Terra). Regionally, however, the spatial distributions of both the HRM-simulated and LRM-simulated AOTs are different from those of the MODIS-retrieved AOTs. In the Southern Ocean, for example, the NICAM-simulated AOT is overestimated compared to the MODIS-retrieved AOT by at most 0.2. In July, the NICAM-simulated AOT over the Arabian Sea is largely overestimated compared to the MODIS-retrieved AOT. Over land, where the MODIS-retrieved AOT is the most uncertain, the NICAM-simulated AOT is overestimated in the Saharan Desert in July and underestimated in China in January. As a result, over land, both the HRM-simulated and LRM-simulated AOTs are underestimated in January and overestimated in July in comparison with the MODIS retrievals. Over Canada and Siberia where biomass burning often occurs in summer, the NICAM-simulated AOT tends to be largely underestimated compared to the MODIS retrievals, partly due to use of climatological emission inventories for the biomass burning as pointed out in section 2.3 and Figure S2. The global distributions of the statistical metrics, i.e., PCC, RMSE and NMB, for the annually averaged AOTs between the NICAM simulations and MODIS/Aqua retrievals are shown in the supplemental materials (Figure S6). The correlations between the NICAM (both the HRM and the LRM) simulations and MODIS/Aqua data are moderate with a PCC of approximately 0.47 (0.470 for HRM and 0.473 for LRM) for the global averages, PCCs ranging from 0.463 (LRM) to 0.473 (HRM) for the land averages and PCCs ranging from 0.480 (HRM) to 0.499 (LRM) for the ocean averages. The spatial distribution of the PCC shows mostly positive values but displays negative values in some regions, such as Eastern Europe and oceans, at high latitudes. The global annual average RMSEs between the NICAM simulations and MODIS/Aqua retrievals are calculated to be 0.134 and 0.140 (HRM and LRM, respectively), which are lower than the global AOT averages (0.175 for HRM and 0.170 for LRM). The RMSEs are higher than those in other regions with relatively high AOTs, such as western Africa and western Asia near the Arabian Sea, over the oceans within 45°S-70°S where the NICAM-simulated sea salt seems to be overestimated, and in eastern China and central Russia where the NICAM-simulated AOT is highly underestimated compared to the MODIS-retrieved AOT. The RMSEs over land (0.210 for HRM and 0.222 for LRM) are higher than those over the oceans (0.095 for HRM and 0.096 for LRM), primarily because the AOTs over the oceans are

lower than those over land but also because those over deserts are higher due to the presence of dust. The NMBs are calculated to be 6.8% (HRM) and 3.7% (LRM) for the global averages, 4.5% (HRM) and 1.9% (LRM) for the land averages and 7.9% (HRM) and 4.6% (LRM) for the ocean averages. High positive biases are found in the same regions with relatively high RMSEs. In the regions where both the bias and the uncertainty are high, the differences in the RMSE and NMB between the HRM and LRM are small; therefore, the high bias and high uncertainty in western Africa, western Asia, the North Atlantic and the oceans within 45°S-70°S cannot be solved by employing finer horizontal resolutions. As mentioned in section 2.4, because the NICAM-simulated AOT under the all-sky condition and the MODIS-retrieved AOT under the clear-sky condition are compared, the differences in the AOT between the NICAM and MODIS may be partly explained by the differences in the AOT between under the all-sky and clear-sky conditions, especially over the North Atlantic where the HRM-simulated AOT under the all-sky condition is larger than that under the clear-sky condition by up to 0.1 (Figure S3). Over the oceans within 45°S-70°S, however, there are no clear tendency, with a mixture of positive and negative biases (Figure S3). The largest difference in the NMBs between the HRM and LRM is found in the vicinity of the western Pacific and the northern Indian Ocean, where the difference in the precipitation between the two is also large, which is partly shown in Figure 2(a). In these regions, although the AOTs and their RMSEs are lower than those in other regions, aerosols could be important because they act as a main trigger for the onset of the monsoon season (e.g., Li et al., 2016) and because sporadic biomass burning occurs throughout the dry season.

Although polar-orbiting satellites cover large areas and provide global AOT distributions, the accuracy of satellite-retrieved AOTs is lower than that retrieved from ground-based measurements. Figure 6 shows scatterplots of the AOTs between the NICAM simulations and in-situ observations, including AERONET, SKYNET and CARSNET, whose site locations are shown in Figure 1. A comparison of the AOTs between the NICAM simulations and satellite retrievals shows almost no differences between the HRM and LRM, but a comparison with in-situ measurements shows differences between the HRM and LRM. The HRM-simulated AOTs have a higher correlation (PCC=0.471), lower uncertainty (RMSE=0.21), and lower bias (NMB=-20.2%) in the annual averages than the LRM-simulated AOTs with PCC=0.356, RMSE=0.24 and NMB=-26.6%. Furthermore, the tendencies obtained in the annual averages are similar to those obtained in the January and July averages. When the HRM-simulated AOTs with a grid converted to 0.5°×0.5° by averaging 16 pixels of 0.125° grids are

evaluated using the in-situ measurements, the statistical metrics are worse than those in the original grids, i.e., lower PCC (-0.014), higher RMSE (0.003) and larger NMB (-0.8%) with regard to the annual averages (Table S5), but still higher than those in the LRM results. This finding suggests that the 0.5° grid is not fine enough to correspond to the representative value at the observation sites and the differences in AOTs between the HRM and LRM are not due to the grid conversion but the model resolution itself. More details of the differences between the HRM and LRM are discussed in section 4.1.

To further investigate these differences in the aerosol components, the mass loadings of the aerosol components are directly compared between the HRM and LRM in Figure 7. For additional references, the differences in the decomposed AOT components as well as the aerosol surface mass concentrations between the HRM and LRM are shown in the supplemental Figures S7 and S8. The global and annual differences in the dust mass loadings are very small (0.32 mg m$^{-2}$ or 0.6%), although the regional differences are not small in the outflow regions, such as the Arabian Sea. The difference in the sea salt mass loadings between the HRM and LRM is larger than that in the sea salt AOTs (supplemental Figure S7(c)) by more than a factor of two, which probably cancels the difference in the mass loadings by the RH difference. The difference in the sulfate mass loadings (-0.48 mg m$^{-2}$ or -15.7%) is larger than that in the sulfate AOTs (-11.3%) shown by supplemental Figure S7(d). The carbon components can be decomposed into POM, water-soluble BC (WSBC) and water-insoluble BC (WIBC). The global and annual averages of the differences in these mass loadings between the HRM and LRM are all negative and are calculated to be -0.20 mg m$^{-2}$ (-9.9%) for POM, -0.01 mg m$^{-2}$ (-10.4%) for WSBC and -0.04 mg m$^{-2}$ (-32.1%) for WIBC. The regional differences in POM and WSBC are noticeable near the source regions, whereas those in WIBC are found to be not only near the source regions but also largely distributed even throughout the Arctic. These comparisons of aerosol mass loadings show that the differences in the components, especially WIBC, sulfate, WSBC and POM, between the HRM and LRM are remarkable.

The global budgets of these aerosols are summarized in Table 2, which includes the mass loading or column density, chemical budget (emissions and deposition through dry processes, gravitational settling and wet processes), and atmospheric lifetime. To support the analysis, global distributions of the differences in these budgets between the HRM and LRM are shown in the supplement (Figures S9-S16). These values of the global budgets are generally within the variabilities and uncertainties estimated by other global models (e.g., Textor et al., 2006), except for the lifetimes of some aerosols. The

lifetime is defined as a ratio of column burden to emission or total deposition fluxes in a global average (e.g., Seinfeld and Pandis, 2006; Textor et al., 2006); therefore, the differences in the lifetime between the HRM and LRM are caused by those in the column burden or the emission flux. The global annual sums of the emission fluxes are almost identical to those of the total deposition fluxes in global annual averages for usual global models (e.g., Textor et al., 2006; Matsui and Mahowald,

2017). While the differences in the emission fluxes of POM, BC, WSBC and WIBC between the HRM and LRM are almost zero (relative difference of less than 1%, as shown in Table 2 and Figure S9), those for dust, sea salt and the sulfate are not zero (relative difference of more than 3%, as shown in Table 2 and Figure S9) because these emissions are calculated online. This means that the differences in the lifetimes of POM, BC, WSBC and WIBC between the HRM and LRM are mainly caused by the differences in the column burdens, whereas differences for dust, sea salt and sulfate are caused by differences

in both their column burdens and their emission fluxes.

The lifetime of sea salt is approximately 0.2 days (both the HRM and the LRM), which is at the lower limit of prior studies (0.20-0.98 days by Textor et al., 2006; Matsui and Mahowald, 2017; Bian et al., 2019). Among the differences in the budgets between the HRM and LRM, those of the wet deposition flux of sea salt are large over the most of ocean and estimated to be 20% globally (Table 2 and Figure S11(c)). This is mainly due to the larger RPCW values shown in Figure 3(d). For dust, the

differences in the dust column and budgets as well as the lifetime between HRM and LRM are very small in a global average (Table 2) but regionally large (Figure S10). Because the lifetimes of POM and BC are within the variabilities reported from previous studies, the wet deposition fluxes, especially over the oceans, seem to be larger (Table 2 and Figures S13 and S14), which is consistent with the results of sea salt and mainly due to the larger RPCW values. The lifetimes of WIBC are comparable to those proposed by a previous study by Goto et al. (2012) but much longer than that of the previous studies

that considered the atmospheric aging (Chung and Seinfeld, 2002; Goto et al., 2012). The differences in the lifetimes between the HRM and LRM are large and estimated to be -22% for BC, -10% for WSBC and -33% for WSBC globally. The differences in their lifetimes or their column burdens between the HRM and LRM are mainly caused by wet deposition (Table 2 and Figures S14, S15 and S16). The wet deposition fluxes for aerosols in the HRM are generally smaller than those in the LRM, because the RPCW values in the HRM are smaller than those in the LRM. Therefore, over the outflow region,

the wet deposition fluxes for BC, WSBC and WIBC in the HRM are smaller than those in the LRM. However, over land

where the aerosol concentrations are large, the wet deposition fluxes in the HRM are larger than those in the LRM because the wet deposition fluxes are proportional to the aerosol concentrations (e.g., Seinfeld and Pandis, 2006). Near the source region of BC, for example in China, wet deposition in the HRM is larger than that in the LRM (Figure S14), mainly due to the larger concentrations, even though the RPCW values in the HRM are larger than those in the LRM.

The lifetimes of sulfate are estimated to be 2.38 days (HRM) and 2.05 days (LRM), which are smaller than those (ranging from 3.3 days to 4.9 days) in the literature (Textor et al., 2006; Matsui and Mahowald, 2017). Sulfate aerosols are produced through $SO_2$ oxidation in the gas and aqueous phases. The sulfate production through both phases in the HRM is generally larger than in the LRM. The global annual relative differences through the gas and aqueous phases between the HRM and LRM are estimated to be -0.5 TgS yr$^{-1}$ (-3.6%) and -1.1 TgS yr$^{-1}$ (-2.5%) (Table 2, Figure S9(c) and Figure S9(d)), but their

differences vary regionally, especially in East Asia. These differences between the HRM and LRM can be explained by the concentrations of both $SO_2$ and clouds, although the HRM-simulated clouds tend to be smaller than the LRM clouds, as shown in Figure 3. Therefore, these differences between the HRM and LRM are solely due to $SO_2$ concentration. This is also why the sulfate production rates through both the gas and aqueous phases in the HRM are greater than those in the LRM (Figure S9(d) and S9(e)). As a result, the HRM-simulated sulfate concentrations increase, but the wet deposition for sulfate

in the HRM is larger than that in the LRM (Table 2 and Figure S12), as explained for BC that the wet deposition fluxes are proportional to the aerosol concentrations, even though the RPCW values in the HRM are larger than those in the LRM. In the end, the HRM-simulated sulfate in terms of the column burden is larger than in the LRM by 16% in a global average (Table 2), which mainly determines the differences in the lifetimes for sulfate. Therefore, the impact of the horizontal resolution (14-km and 56-km grid spacings), which determines the meteorological parameters including wind, vertical

mixing, diffusion, clouds and precipitation fluxes, on dust is very small, but sea salt, sulfate and BC are strongly influenced. Because almost all aerosols are emitted from the surface, evaluations of the surface mass concentrations of those aerosols are important. The supplement (Figure S8) shows the global distributions of the differences in the annual averages of the aerosol surface mass concentrations between the HRM and LRM. Compared to the differences in the AOTs (Figure S7) and mass loadings (Figure 7), the differences in the surface mass concentrations are generally smaller but have different signs for

carbonaceous aerosols, i.e., POM, WSBC and WIBC. This is probably because the NICAM-simulated biomass burning-

emitted aerosols, i.e., parts of carbonaceous aerosols, are ejected at a height of 1 km (not the surface). For dust, sea salt and sulfate, the horizontal patterns of the differences in their surface mass concentrations between the HRM and LRM are similar to those of the AOTs and mass loadings. The simulated surface mass concentrations are evaluated by the measurements described in section 2.4, and the results are shown in Figures 8 and 9, which illustrate scatterplots of the annually averaged

surface mass concentrations of the aerosol species between the satellite measurements and NICAM simulations. The annual averages of three compounds, i.e., sulfate, BC and POM, are compared over North America, Europe and Asia, whereas not only the annual averages but also the January and July averages of dust and sea salt are compared at sites worldwide due to their large seasonal variabilities. The statistical metrics for the comparison are also shown in Table S6 to Table S8. The model results in both the HRM and the LRM exhibit a high correlation, low uncertainty and low bias, except for the

relatively high negative bias for BC and POM with NMBs ranging from 46% to 56%. Although the differences in the statistical metrics between the HRM and LRM are very small, the metrics of the HRM are generally better than those of the LRM. As mentioned in the AOT comparison using in-situ measurements, although a 0.5° grid may not be fine enough to represent the observation sites, the difference between the HRM and LRM is not due to the analysis grid size but the model resolution itself (Table S6). BC and most POM simulated in the NICAM are primary compounds that tend to be localized

near the source region; thus, the simulated BC and POM distributions with finer grid spacing are expected to be better. The differences in the simulated sulfate, which is a secondary component, between the HRM and LRM are caused both by differences in the transport of $SO_2$ and sulfate and by the cloud distributions related to sulfur chemistry (Goto et al., 2015b). The lower conversion ratio of the simulated precipitation to the simulated clouds (Figure 3(d)) in the HRM than that of the LRM results in a longer lifetime for sulfate (Table 2) and provides larger values for the HRM-simulated sulfate. Even the

HRM provides large underestimations of the simulated BC and POM, which is mainly because of their underestimation in China. The possible reasons for this phenomenon are probably the underestimation of BC and POM emissions and possibly the excessive localization of measured values. These findings are consistent with the results of the AOT underestimation in Asia (Figure 6).

The annual and January averages of both the HRM-simulated and the LRM-simulated dust mass concentrations at the

available sites are comparable to the measurements. The correlations are high to moderate (the PCCs of the annual averages

are approximately 0.9, and the PCCs of the January averages are approximately 0.65), the uncertainty is relatively small (the RMSEs of the annually averaged HRM- and LRM-simulated concentrations are less than 4 µg m$^{-3}$, while the January-averaged HRM-simulated concentration is 9 µg m$^{-3}$, and the January-averaged LRM-simulated concentration is 4 µg m$^{-3}$) and the bias is relatively low (the NMBs range from -22% to +37%). However, the uncertainty and bias in the July averages are higher than those in the annual and January averages, mainly because the emission fluxes from the Saharan and Arabian Deserts are larger in summer (July). The NMBs are calculated to be -64.8% (HRM) and -55.6% (LRM), the RMSEs are calculated to be 10.6 µg m$^{-3}$ (HRM) and 10.2 µg m$^{-3}$ (LRM), and the correlations are high (PCC=0.75 for the HRM and PCC=0.68 for the LRM).

For sea salt, the correlation is poor, except for the HRM in January, where the correlation is moderate with PCC=0.62. Because the emission fluxes of sea salt are strongly correlated with winds (the power of 3.41 mentioned in section 2.3), the differences in the simulated wind speeds shown in Figure 2 strongly affect the reproductivity of sea salt. The difference at the wind speed of 1.5 m s$^{-1}$ provides the error in the sea salt emission flux of approximately 4. Therefore, a small error in the simulated wind speed can easily cause biases in the simulated sea salt emissions and its mass concentrations. Nevertheless, the bias and uncertainty of the NICAM-simulated sea salt are not large. The RMSE ranges from 7.7 to 8.2 µg m$^{-3}$ for the HRM and from 7.9 to 10.6 µg m$^{-3}$ for the LRM, while the NMB ranges from -29% to -18% for the HRM and from -41% to -31% for the LRM. Therefore, without nudging the meteorological fields, it is difficult to obtain results similar to the measurements in the sea salt simulation, even with the fine horizontal grid spacing of 14 km in this study.

In summary, both the HRM-simulated and the LRM-simulated aerosols are generally close to the MODIS-retrieved results and in-situ measurements, although differences in the column burden between the HRM and LRM are found for sulfate (11.3%), WIBC (32.1%), POM (9.9%) and WSBC (10.4%). These are mainly caused by the modification of aerosol-cloud-precipitation interactions through wet deposition under the different horizontal grid spacings. The above verification of the relevant variables suggests that both the HRM and LRM can be applied for a current global climate aerosol model. However, several important differences between the HRM and LRM have not been addressed in detail; therefore, section 4 discusses the remaining issues associated with using the HRM relative to the LRM.

**4 Discussion**

In section 3, the modeled results using the HRM and LRM are shown as annual or monthly averages and/or global distributions of aerosol species using multiple measurements, i.e., MODIS, AERONET, IMPROVE, etc, against multiple variables, i.e., AOT and surface aerosol mass concentration for each aerosol component. This finding indicates that the results of both the HRM and LRM are generally within the uncertainties of the measurements and other global models; furthermore, the differences in these variables between the HRM and LRM are not large in terms of the annual and global averages. However, some remarkable differences are found at the regional scale and in the results for sulfate and BC, but these differences and their mechanisms have not been thoroughly investigated. In section 4, more detailed comparisons are carried out to reveal both the differences in the simulated variables between the HRM and LRM and the advantages of using the HRM.

**4.1 Effects of a fine grid spacing on aerosol fields**

A high-resolution horizontal grid spacing has the potential to provide more realistic values of model subgrid variability and possibly more realistic averages, for example, for aerosol concentrations in highly polluted areas, because most aerosols are emitted from heterogeneous hotspots on the surface. Figure 10 shows the mass concentrations of both BC and sulfate and the AOTs at the relevant sites, which are selected from the most polluted sites in the monthly averages within typical domains such as the United States, Europe and China. These results are derived from the results shown in Figures 6 and 8. In Figure 10, three sets of results are compared: the HRM with the original grid of 0.125°×0.125°, the HRM with the grid converted to 0.5°×0.5° and the LRM with the original grid of 0.5°×0.5°; hereafter, these models are referred to simply as HRM, HRM-0.5° and LRM, respectively. As already mentioned in section 3, since an exact comparison using two models requires the same grid size, the HRM-0.5° is introduced to clarify the differences caused only by the grid size. The results show that the HRM-simulated BC concentrations are the largest among the simulations because BC is a primary aerosol and the relevant

sites are located near BC emission sources. The HRM-0.5°-simulated BC concentrations are larger than the LRM-simulated BC concentrations. For example, during April in Chengdu, China, the simulated BC concentrations are 3.3 µg m$^{-3}$ (HRM), 2.8 µg m$^{-3}$ (HRM-0.5°) and 2.2 µg m$^{-3}$ (LRM), which indicates that the difference among the simulated BC concentrations is approximately 35%. The differences, i.e., the relative ratios of the HRM-0.5° or LRM to the HRM results, range from -63%

(PHOE1, United States, December) to -2.5% (ATLA1, United States, September). Because the range represents the spatiotemporal variability of the selected sites and months, an estimation of bias, i.e., NMB, is meaningful and is found to be -18.4%. Compared with the measurements, especially in China, even the HRM-simulated results tend to be underestimated. This is probably because the BC emission inventory in China is underestimated (e.g., Goto et al., 2015b) or the 14-km grid spacing is not sufficient to resolve such high concentrations in highly dense urban areas. For sulfate, which can serve as a

representative secondary aerosol, and the AOT, which is highly influenced by RH, the HRM-simulated results are generally the best among the simulations, and the HRM-0.5°-simulated results are larger than the LRM-simulated results. For example, during August at BALT1 in the United States, the sulfate concentrations simulated by the HRM and HRM-0.5° range from 9.6 to 10.0 µg m$^{-3}$, whereas that simulated by the LRM is 0.9 µg m$^{-3}$, which is very different from the measurement (7.6 µg m$^{-3}$). The differences in the simulated sulfate concentrations among the simulations at all sites range from -91% (BALT1,

United States, August) to +18% (ATLA1, United States, May), with an NMB of -5.3%. Underestimated simulated sulfate concentrations are also found in China and Vietnam, whereas such underestimations are not generally found in the United States or Europe. At some sites, the LRM-simulated AOTs are the largest among the simulations. These complex results imply complicated situations where the AOT depends on not only the aerosol burden but also the RH, whereas the BC mass concentration near the surface strongly depends on BC emissions. The differences in the simulated AOT concentrations

among the simulations at all sites range from -49% (Nanging, China, August) to +223% (Nanging, China, August), with an NMB of -2.6%. The NMB values of the differences in the AOT among the simulations are smaller than those in sulfate by 5% and those in BC by 18%. This finding suggests that the primary product, i.e., BC, is the most influenced by the grid size, but the secondary product, i.e., sulfate formed from oxidation of SO$_2$ (this is a primary product) and removal through precipitation, and the complex product, i.e., AOT comprising various aerosols including primary and secondary particles and

being highly dependent on RH, are less influenced by the grid size. Therefore, the impacts of higher horizontal grid spacings

on model performance for secondary products, such as sulfate, and complex products, such as AOT, are weaker than those for primary products, such as BC.

In addition to the impact of the model grid size on the monthly averages of the aerosol concentrations at the relevant sites, the temporal variations in the aerosol concentrations are also investigated. Such comparisons were carried out by Lin et al. (2017), who investigated the marine aerosol subgrid variability using a regional HRM over the southern Pacific Ocean. Lin et al. (2017) estimated variabilities in the aerosol mass concentrations of 15% near the surface and 50% in the free troposphere in a 180-km×180-km domain using 3-hourly 3-km×3-km original grids for October 2008. In this study, these variabilities of the AOT, CCN at a height of 2 km, COT and precipitation are calculated on a 1°×1° domain using 6-hourly 14-km×14-km original grids for one year (Figure 11). The global and annual averages of the ratio for the AOT are calculated to be 28.5% (HRM) and 16.6% (LRM). The value obtained from only the HRM ranges between the two values obtained by Lin et al. (2017). For the CCN at a height of 2 km, the values are relatively small (7.6% for the HRM and 4.1% for the LRM), partly because the simulated CCN may be underestimated compared to the measurements, which show at least 100 cm$^{-3}$ even over the oceans (e.g., Heintzenberg et al., 2000). Clouds and precipitation are also strongly influenced by subgrid variability (e.g., Pincus et al., 1999; Hakuba et al., 2013; Boutle et al., 2014). The global and annual averages of the ratio for the COT are calculated to be 80.0% (HRM) and 22.9% (LRM), whereas the global and annual averages of the ratio for precipitation are calculated to be 216.2% (HRM) and 77.9% (LRM). These values for clouds and precipitation are much larger than those obtained for aerosols. The relative differences in these parameters between the HRM and LRM are calculated to be 1.7 (AOT), 1.9 (CCN), 3.5 (COT) and 2.8 (precipitation). These results clearly indicate the importance of high-resolution simulations, especially for reproducing extreme phenomena related to aerosol, clouds and precipitation such as in the Amazon where the subgrid variabilities of both COT and precipitation in the HRM are high.

## 4.2 Arctic

Aerosols over the Arctic, especially BC, are very important due to their impact on climate change (e.g., Willis et al., 2018). Unfortunately, it is generally difficult for global models to properly reproduce aerosols over the Arctic (e.g., Shindell et al.,

2008; Eckhardt et al., 2015; Sand et al., 2017). To solve this issue, many improvements to BC models have been applied by previous studies to microphysics processes, including aging and wet deposition processes (e.g., Liu et al., 2011; Lund and Berntsen, 2012; Marelle et al., 2017), and to the horizontal resolution to resolve the fine structures of clouds (e.g., Ma et al., 2014; Sato et al., 2016; Raut et al., 2017). Figure 12 illustrates the monthly variations in the BC and sulfate concentrations at three sites over the Arctic using four simulations: the HRM, LRM, LRM-macro (56-km grid spacing but using a cloud macrophysics module described in section 2.1) and very low-resolution NICAM model (220-km grid spacing using a cloud macrophysics module described in section 2.1). Similar to previous studies (e.g., Shindell et al., 2008; Eckhardt et al., 2015; Sand et al., 2017), the LRM-macro- and VLRM-macro-simulated BC concentrations are also very different from the measurements. At Alert and Zeppelin, for example, the LRM-macro- and VLRM-macro simulated BC concentrations are highly underestimated, and the observed variation cannot be reproduced. However, both the HRM and LRM with the cloud microphysics module succeed in simulating the observed seasonal variation (with the maximum in spring and the minimum in summer), but the HRM results are closer to the observations than the LRM results as a result of WIBC, as shown in Figure 7. At Barrow, the finer grid spacing of the LRM-macro-simulated BC provides better results than the VLRM-macro-simulated BC, but the former is still underestimated compared to the measurements, especially in spring. The good performance of the HRM and even the LRM can be found in the simulation of sulfate. Between the HRM and LRM, the largest difference in the surface BC mass concentrations between the HRM and LRM reaches 30% in spring. The differences in the simulated BC and sulfate concentrations between the HRM, LRM, LRM-macro and VLRM are mainly explained by differences in the CF and aerosol wet deposition, as shown in Sato et al. (2016). Near the aerosol source region, the simulated aerosol concentrations are strongly affected by their emission fluxes, but in remote areas such as the Arctic, aerosol wet deposition, which is directly related to cloud and precipitation processes, becomes important for their atmospheric lifetimes. In the LRM-macro and VLRM-macro simulations, the wet deposition process in winter results in unrealistic seasonal variations over the Arctic. The importance of wet deposition over the Arctic has also been noted by previous studies, such as Garrett et al. (2011), whose findings are consistent with our study. In addition, our results clearly show the importance of using numerical models with a cloud microphysics module, which introduces prognostic

precipitation and does not diagnose the assumed CF used in the macrophysics cloud module. In summary, these processes related to hydrometeors and thus aerosol wet deposition strongly affect the aerosol simulations, especially over the Arctic.

## 4.3 Vertical distributions of aerosols

Thus far, the horizontal distributions of the aerosols and their species are compared between the HRM and LRM and are validated using available measurements, but their vertical distributions are important for radiative forcings, especially BC (e.g., Haywood and Shine, 1997; Samset et al., 2013), although the model variability of BC is large (e.g., Textor et al., 2006; Kipling et al., 2016). Figure 13 shows the vertical profiles of the simulated and CALIPSO-retrieved aerosol extinction coefficients in 12 different regions, which are generally based on the definition in Koffi et al. (2016) by comparing Aerosol Comparisons between Observations and Models (AeroCom) models with CALIOP. The results of the HRM and LRM are generally comparable to those retrieved from CALIOP, but remarkable differences between the NICAM simulations and CALIOP retrievals are found in various regions, such as South America (panel k in Figure 13) and North Africa (panel h in Figure 13). In South America, the plume height is approximately 4 km in the NICAM simulations but approximately 2 km in the CALIOP measurements. As a result, the aerosol extinction coefficients of both the HRM and the LRM are underestimated below a height of 3 km. This may be consistent with the AOT results shown in Figures 5 and 6, which show the underestimation of the AOT in the NICAM simulations compared to the MODIS and AERONET retrievals caused by the underestimation of biomass burning emissions or the overestimation of transport to upper-level areas. In northern Africa, where dust is a major component but the simulations exhibit large variabilities among the global models (e.g., Kim et al., 2014), both the HRM-simulated and the LRM-simulated extinction coefficients are overestimated compared to those retrieved from CALIOP, although the vertical profiles simulated by both the HRM and the LRM are comparable to those retrieved from CALIOP. This is also consistent with the overestimation of the AOT shown in Figures 5 and 6. The reason for this overestimation is probably attributed to the overestimation of dust emission fluxes in the NICAM simulations, which can be attributable to several sources: the overestimation of the wind speeds at a height of 10 m (Figure 2), the underestimation of soil moisture, and the failure to appropriately tune the model for dust emissions, although the global amount of emitted dust is within the variability estimated by other global models shown in Table 2. In addition, this finding suggests that the

difference in the transport processes between different horizontal grid spacings is very small. Below a height of 5 km, the differences in the extinction coefficients between the HRM and LRM are small in all regions except for East China and the northwestern Pacific. It should be noted that the simulated extinction coefficient may not be overestimated above a height of 5 km because optically thin aerosols are often undetected by CALIOP in the upper troposphere and the CALIOP regionally averaged extinction coefficient tends to be underestimated above 5 km (Watson-Parris et al., 2018).

Vertical observations of aerosol species are still limited, but recent measurements of vertical BC by flight campaigns such as HIPPO are available for a model evaluation (e.g., Schwarz et al., 2013; 2017; Samset et al., 2014; Lund et al., 2018). Figure 14 shows the NICAM-simulated vertical BC profiles and the measured vertical BC profiles from various missions in different regions: flights in HIPPO for annual averages over the Pacific, ARCTAS in spring and summer over the Arctic region where CALIOP does not generally detect aerosol signals, and A-FORCE in spring over East Asia where anthropogenic BC is likely transported to the Arctic (and which can be an important source of BC over the Arctic) (e.g., Ikeda et al., 2017). The NICAM-simulated BC vertical profiles are generally comparable to those observed by the flights and generally closer to the observations than other global models (Koch et al., 2009; Samset et al., 2014; Matsui and Mahowald, 2017; Kaiser et al., 2019; Tegan et al., 2019). Over the majority of the Pacific Ocean (60°S-60°N, 160°E-150°W), the NICAM-simulated BC concentrations show as annual averages below a height of 3 km (approximately 700 hPa) in Figure 14 (a) to (c) that are generally within the uncertainties obtained from the variability of the measurements, whereas the differences in the BC concentrations between the HRM and LRM are very small. Above approximately 700 hPa, however, the differences between the HRM and LRM become large, which is consistent with the results of the comparison with CALIOP in Figure 13. Moreover, because the LRM-simulated BC concentrations are lower than those in the HRM, those in the LRM are closer to the observations than those in the HRM. As already mentioned in Table 2 in section 3.2, the differences in the BC concentrations between the HRM and LRM are caused by differences in the BC lifetime, especially for WIBC. In addition, the HRM-simulated BC concentrations and even the LRM-simulated BC concentrations around the equator (20°S-20°N) are overestimated compared to the measurements, which has been noted by previous studies as one of the current problems among global aerosol models (e.g., Koch et al., 2009; Samset et al., 2014; Schwarz et al., 2017). The reason for this overestimation is possibly the overestimation of the BC atmospheric lifetime, which must be smaller than 5

days (Lund et al., 2018) but larger than 5 days in the HRM and other global models (Table 2). The overestimation of the BC lifetime may be attributed to the underestimation of the wet deposition of WIBC in the HRM, overestimation of the convective mass flux above 500 hPa, which may be improved by increasing the number of vertical layers in the model (Allen and Landuyt, 2014), possible overestimation of the climatological BC emission from biomass burning (Figure S1),

and a lack of secondary aerosol activation by convective clouds and associated removal by precipitation (Yu et al., 2019).

Over the Southern Ocean (60°S-80°S, 160°E-150°W), where aerosols are transported from other areas, the observed BC concentrations are 10-50 ng kg$^{-1}$ near the surface and more than 1 ng kg$^{-1}$ at approximately 300 hPa. The surface BC concentrations are much lower than those in other areas, but those at 300 hPa are comparable to those in HIPPO-P2 (20°N-60°N) and HIPPO-P4 (60°S-20°S) and higher than those in HIPPO-P3 (20°S-20°N). These features of the observations are

generally reproduced by the NICAM simulations, but the simulated BC concentrations tend to be overestimated compared to the measurements. Although previous studies have offered only a limited discussion of BC transport to the Antarctic, this overestimation may be caused by the underestimation of BC wet deposition and possibly the overestimation of the horizontal transport of simulated BC to the Antarctic.

As discussed in section 4.2, both the HRM and the LRM successfully reproduce the aerosols over the Arctic. In Figure 14 (e),

(g) and (h), where the vertical BC profiles over the Arctic region (>60°N) are shown, the NICAM-simulated BC concentrations near the surface are generally comparable to the measurements except for the July-August average in panel (g). The observed BC concentrations in July-August seem to be inconsistent with the results in Figure 12, where BC over the Arctic reaches a maximum concentration in spring (February-April) and a minimum in summer (June-October). This may be caused by specific smoke plumes during the observation period (Liu et al., 2011; Allen and Landuyt, 2014); however, these

disturbances are not considered in our simulations because climatological emission fluxes are employed in this study. In fact, intensive biomass burning was observed in Russia and North America in the target year (2008) of the measurement period compared to the climatological years (Yasunari et al., 2018). In addition, the simulated BC concentrations in July-August are underestimated not only at the surface but also at all heights compared to the ARCTAS-B flight measurements. In the annual averages shown in panel (e), the simulated BC concentrations generally match the measurements, but above approximately

300 hPa, both the HRM and LRM overestimate the BC concentrations, which is also the case in other regions (panels (a)-(d)

in Figure 14). In spring (March-April) over the Arctic, both the HRM-simulated and LRM-simulated BC concentrations are generally comparable to the measurements, even those obtained by the field campaign flights (Figure 14(g)). In the main source regions of Arctic BC, i.e., East Asia, both the HRM-simulated and the LRM-simulated BC concentrations exhibit better agreement in the measurements (Figure 14(f)). In the middle troposphere (approximately 400-800 hPa) over the Arctic, however, both the HRM-simulated and LRM-simulated BC concentrations are underestimated compared to the ARCTAS-A measurements. These findings may suggest that the HRM with O(10-km) grid spacing cannot resolve the lifting process of aerosols along the Arctic front as pointed out by Quinn et al. (2011). Even when a source-receptor analysis of BC concentrations is used to identify the sources, the results remain highly uncertain, and no clear conclusions have been reached among previous studies. For example, Ikeda et al. (2017) employed the GEOS-Chem model and concluded that BC is mainly contributed by East Asian anthropogenic sources, whereas Matsui et al. (2011) used backward trajectories with both ARCTAS measurements and Weather Research and Forecasting (WRF) simulations and concluded that BC over the Arctic is mainly contributed by biomass burning from Russia, North America and Europe. The differences between these models and measurements can be partly caused by a sampling problem without using exact grids and periods (Schutgens et al.,2016). In conclusion, a high-resolution grid system resolves one of the major issues regarding the distribution of BC, namely, the overestimation in the upper troposphere over the Pacific Ocean, but it does not solve the issue of its underestimation in the middle to upper troposphere over the Arctic.

## 4.4 Aerosol radiative forcing (ARF)

ARFs, which are complicated by various aerosol parameters, are the most important factors for estimating the impacts of aerosols on climate. Figure 15 shows the global and annual average ARFs due to the direct and indirect effects of anthropogenic and all aerosols, i.e., IRFari, ERFari, and ERFaci, under all-sky and clear-sky conditions. The values of the ERFari due to anthropogenic aerosols under all-sky conditions with uncertainties, which represent global confidence intervals with a significance threshold of 95%, are estimated to be $-0.293\pm0.001$ W m$^{-2}$ (HRM) and $-0.239\pm0.002$ W m$^{-2}$ (LRM), which are within the AeroCom estimates (from $-0.58$ W m$^{-2}$ to $-0.02$ W m$^{-2}$ with a mean of $-0.20$ W m$^{-2}$ and a standard deviation of 0.15 W m$^{-2}$ by Myhre et al., 2013) but slightly smaller than the estimation by the Max Planck Aerosol

Climatology version 2 (MACv2) (-0.35 W m$^{-2}$) by Kinne (2019). Under clear-sky conditions, the values of the ERFari due to anthropogenic aerosols are estimated to be -0.567±0.001 W m$^{-2}$ (HRM) and -0.479±0.004 W m$^{-2}$ (LRM), which are also within the AeroCom estimates (from -1.01 W m$^{-2}$ to -0.35 W m$^{-2}$ with a mean of -0.71 W m$^{-2}$ and a standard deviation of 0.18 W m$^{-2}$ by Myhre et al., 2013) but smaller than the MACv2 estimate (-0.69 W m$^{-2}$) by Kinne (2019). The differences in

the ERFari values between the HRM and LRM under both all- and clear-sky conditions are within 0.1 W m$^{-2}$, which is smaller than the standard deviation among the AeroCom models. The uncertainty of the HRM is smaller than that of the LRM becausee the number of samplings is 16 times higher (due to different number of grids). The values of IRFari due to all aerosols, i.e., anthropogenic and natural aerosols, under all-sky conditions are estimated to be -1.791±0.002 W m$^{-2}$ (HRM) and -1.697±0.010 W m$^{-2}$ (LRM). For only shortwave fluxes, the IRFari values are estimated to be -2.019±0.003 W m$^{-2}$

(HRM) and -1.927±0.011 W m$^{-2}$ (LRM), which are comparable to the measurement-based estimates using CALIOP (Oikawa et al., 2018) and MACv2 (Kinne, 2019) within approximately 0.2 W m$^{-2}$ but smaller than the assimilated estimate of -3.1 W m$^{-2}$ (Su et al., 2013). Table 3 shows the TOA and surface components of the IRFari under all-sky and clear-sky conditions. First, the largest difference in the TOA IRFari between the HRM and LRM under all-sky conditions is found for sulfate (-0.048 W m$^{-2}$), whereas the differences in the other components between the HRM and LRM are within 0.020 W m$^{-2}$. This is

also consistent with the differences in the AOT and column burden shown in Figures S7 and 7. Second, the largest difference in the surface IRFari between the HRM and LRM under all-sky conditions is found for WIBC (-0.061 W m$^{-2}$), whereas that in the TOA IRFari is only 0.020 W m$^{-2}$. Under clear-sky condition, the largest difference in the RFari for WIBC is -0.004 W m$^{-2}$ at the TOA and -0.074 W m$^{-2}$ at the surface. The differences in the IRFari at the surface are consistent with those in the column burden of WIBC; thus, these differences between the TOA and surface can be explained by the stratification of

WIBC and clouds (e.g., Haywood and Shine, 1997). Although the differences in the IRFari due to BC between the HRM and LRM are found, both the HRM and LRM estimated a positive IRFari due to the WIBC and even the WSBC seems to be underestimated compared to the observation-based studies by Oikawa et al. (2018) and Kinne (2019). This is supported by the fact that the differences in the IRFari between all-sky and clear-sky conditions are lower than those by Oikawa et al. (2018) and Kinne (2019), which is probably because the light-absorption of carbonaceous aerosols is underestimated due to

the underestimation of cloud scattering. Third, the values of the shortwave IRFari due to all aerosols at the surface under all-

sky conditions are estimated to be -3.330±0.005 W m$^{-2}$ (HRM) and -3.272±0.022 W m$^{-2}$ (LRM), the absolute values of which are smaller than those in previous studies based on satellites (-4.23 W m$^{-2}$ to -7.79 W m$^{-2}$, summarized by Korras-Carraca et al., 2019) and the MACv2 estimate (-4.0 W m$^{-2}$) by Kinne (2019). This is probably because the dust shortwave IRFari values in both the HRM and LRM have larger negative values among the aerosol species due to the overestimation of

the single scattering albedo (SSA) over desert areas (0.96-0.97 in this study) than those in other studies based on AERONET retrievals (0.92 in Giles et al., 2012). Another reason is the underestimation of ground surface albedo, which is a tendency of the NICAM, and our previous study (Dai et al., 2018) also showed negative IRFari values even over desert areas. Fourth, the IRFari due to sea salt under all-sky conditions is estimated to be -0.474±0.000 W m$^{-2}$ (in both the HRM and LRM), which is comparable to the values reported in previous studies (-0.21 W m$^{-2}$ to -2.21 W m$^{-2}$ in Partanen et al., 2014; -0.31 W m$^{-2}$ in

Takemura et al., 2002; -0.55 W m$^{-2}$ in Jacobson, 2001), even though the simulated AOTs over the oceans tend to be larger than the satellite results. Again, this gap in the IRFari between the model-based and observation-based estimates cannot be solved using a finer grid resolution in global models.

Using the method proposed by Ghan (2013), the values of the ERFaci due to the anthropogenic aerosols are estimated to be -0.919±0.004 W m$^{-2}$ (HRM) and -1.101±0.013 W m$^{-2}$ (LRM). These values are comparable to those in another study (-1.06

W m$^{-2}$) by Jing and Suzuki (2018) and slightly larger than the values published in the Fifth Assessment Report by the Intergovernmental Panel on Climate Change (IPCC-AR5) (-0.45 W m$^{-2}$ with a 90% uncertainty range from 0 W m$^{-2}$ to -1.2 W m$^{-2}$), albeit within uncertainty. However, it should be noted that our estimates are still uncertain. First, this is because the biomass burning emissions in this study are assumed to be zero during the preindustrial era. Second, the minimum CCN value in this study is set at 25 cm$^{-3}$, which strongly affects the ERFaci (Hoose et al., 2009). Third, the interannual variability

among different years can influence the results, as discussed in section 4.5. Furthermore, the magnitude of the difference between the HRM and LRM is estimated to be 0.179 W m$^{-2}$, which is larger than that of the IRFari. The HRM-simulated ERFaci is generally lower than the LRM-simulated ERFaci, which is partly explained by the following: the HRM-simulated CCN concentrations are larger than the LRM-simulated CCN concentrations (Figure 11), and the ERFacci generally becomes smaller as the aerosol concentrations become larger (e.g., Carslaw et al., 2013). In the total effect, because some of

the ERFari and ERFaci cancel out, the difference due to both ARIs and ACIs attributable to forcing in the HRM is calculated to be -0.125 W m$^{-2}$.

## 4.5 Interannual variability

The interannual variabilities of aerosols for 3 years are discussed and quantified by comparing the differences in the aerosols between the HRM and LRM. Figure 16 shows the global annual averages for the relevant parameters (emission fluxes for dust and sea salt, column aerosol burdens, AOT, and IRFari at the TOA) using the HRM and LRM results. The annual averages include 3-year averages as well as 1-year averages in both the HRM and LRM to realize whether the differences between the HRM and LRM are greater or less than the maximum and the minimum difference between each 1-year average of the 3 years. In Figure 16(b), for example, the difference in the 3-year averages of the emission flux for sea salt between the HRM and LRM is estimated to be approximately 900 Tg yr$^{-1}$, which is much larger than the difference in the interannual variability in both the HRM and LRM (the maximum and the minimum difference is approximately 160 Tg yr$^{-1}$ for the HRM and approximately 240 Tg yr$^{-1}$ for the LRM, respectively). Therefore, the impact of the horizontal grid size on sea salt emissions is larger than that caused by interannual variability of the meteorological fields (mainly wind at a height of 10 m) over 3 years. In contrast, in Figure 16(a) the difference in the 3-year averages of the emission flux for dust between the HRM and LRM is smaller than the maximum and the minimum difference between each 1-year average of the 3 years caused by interannual variabilities over 3 years in the HRM and LRM. These may be explained by interannual variabilities in the simulated winds, which are strongly affected by the simulated surface temperature; the SST is fixed, but the temperature over land is a diagnostic variable. Therefore, the differences in the column burden and AOT for dust between the HRM and LRM are smaller than the interannual variabilities. The difference in the column burden for sea salt is slightly larger than that for dust, but the difference in the AOT for sea salt is smaller than the interannual variabilities over 3 years. In Figures 16(e) and 16(k), the differences in the 3-year averages of the column burden and AOT for sulfate between the HRM and LRM are larger than interannual variabilities, indicating that the difference in the clouds and precipitation between the different horizontal grid spacings (14 km versus 56 km) is larger than that among the interannual variabilities with the same horizontal grid spacings. This conclusion is also applicable to the carbonaceous parameters (Figure 16(l)). As a result,

because the contribution of dust and seasalt to the total AOT is larger than that of sulfate and carbonaceous aerosols, the difference in the 3-year averages of the total AOT between the HRM and LRM is smaller than that among the interannual variabilities (Figure 16(m)). For the RFari, the differences in the 3-year averages of the shortwave and total (shortwave plus longwave) RFari under all-sky conditions between the HRM and LRM are slightly larger than those among the interannual

variabilities in the HRM and LRM (Figures 16(n) and 16(r)), whereas the differences in the longwave or shortwave IRFari under clear-sky conditions, which are strongly related to dust, are generally smaller than those among the interannual variabilities in the HRM and LRM. This suggests that the clouds are also significantly modulated by the interannual variabilities, affecting the dust-induced changes in IRFari and diminishing changes in appearance. In summary, the interannual variability is mainly present in the winds over land and RH, which cause relatively larger variabilities in dust

(emission flux, column burden and AOT) and sea salt (mainly AOT). As a result, the total AOT and IRFari under clear-sky conditions and for longwave are more influenced by the interannual variabilities than those by the horizontal resolution. However, the other relevant parameters shown in Figure 16, i.e., the sea salt emission flux, column burdens for sulfate and carbon, including POM and BC, and total RFari under all-sky conditions, are influenced by the horizontal resolution, and discussions of the impacts of different horizontal grid spacings on these parameters can be facilitated using only a 1-year

integration.

## 5 Summary

What is the advantage of an *actual* HRM with aerosols? To address this question, we developed a global aerosol transport model using NICAM.16 with a 14-km horizontal grid spacing. Previous studies have spent considerable amounts of

resources to find the answer, but almost all of these studies were limited in terms of the domain (regional or urban scale) and period (several days to 1 month). Although previous studies have focused on the global scale, the horizontal grid spacing has been still coarse, i.e., more than 50 km. In this study, we execute a global cloud-system resolving model, NICAM, coupled to aerosol components with a 14-km grid spacing and evaluate the simulated aerosol distributions, their budgets and their interactions with clouds against multiple measurements and other models. For comparison, we also execute the NICAM

simulations with a 56-km grid spacing as an LRM, which still boasts a high resolution among the current global aerosol climate models, but coarser than some of those used for operational global aerosol forecasting (Sessions et al., 2015). The integration time is 3 years, which is very long with such an HRM.

The relevant variables, i.e., wind, clouds and precipitation, that strongly determine the aerosol distributions are evaluated using reanalysis data, satellite and in situ measurements. The differences in the global and annual averages between the HRM and LRM are generally within 10%, and both differences generally range within the uncertainties of the measurements and other global models. Our specific conclusions are described below.

- We expected the HRM-simulated wind speeds to be higher than the LRM-simulated wind speeds, but this is not always the case.

- The HRM-simulated precipitation is smaller than that simulated by the LRM because the LRM tends to reproduce unrealistically strong convective clouds compared to the HRM. Such convective clouds can provide strong precipitation due to the coarseness of the horizontal grid spacing. The warm-topped COTs simulated by the HRM are also smaller than the LRM results, but both simulated results are underestimated compared to the MODIS retrievals. In contrast, the HRM-simulated CF for all types of clouds is larger than the LRM-simulated results and closer to the MODIS retrievals.

- The HRM-simulated RPCW, which is very important for determining the aerosol wet removal rate, is smaller than that simulated by the LRM by approximately 20%, which means that the LRM-simulated aerosols are more quickly scavenged by precipitation than the HRM-simulated aerosols.

- Both the HRM-simulated and LRM-simulated TOA shortwave radiative fluxes are closer to the satellite measurements by CERES than are the results of a previous study using NICAM simulations with a 14-km grid spacing but without aerosol components (Kodama et al., 2015). The bias is less than 8 W m$^{-2}$, which is within the uncertainty among global models. However, the LRM reproduces the CERES-estimated radiative fluxes better than the HRM due to the larger clouds in the LRM.

- At the surface, the BSRN-observed SSR is sufficiently reproduced by both the HRM and LRM (PCC=0.9, NMB<1%, RMSE=-32 W m$^{-2}$), although diffuse and direct radiation fluxes have higher biases and uncertainties (reaching up to 30% for the NMB).

The conclusions for the simulated aerosol evaluations are described below.

- Both the HRM-simulated and LRM-simulated AOTs are generally close to the MODIS-retrieved AOTs with PCC>0.47, RMSE<0.14 and NMB<7% in global averages. A comparison using in situ measurements shows that the HRM-simulated AOTs are slightly closer to the measurements than are the LRM-simulated AOTs, as the former have a higher correlation (PCC=0.47), lower uncertainty (RMSE=0.21) and lower bias (-20%).

- The analysis of the chemical components of the AOTs and column burdens shows that the largest difference in the AOTs between the HRM and LRM is found for sulfate (15.7%), followed by all carbonaceous aerosols (5.2%). Large differences in the column burden are found for sulfate (11.3%), WIBC (32.1%), POM (9.9%) and WSBC (10.4%). Differences in sulfate and WIBC occur over a large area.

- The global budgets of aerosol species in both the HRM and LRM generally range within those obtained from other global models, except for the atmospheric lifetime of sulfate, whose lifetime is estimated to be 2.4 days (HRM) and 2.1 days (LRM), whereas it ranges from 3.3 days to 4.9 days in other global models. This tendency is also found for sea salt, whose lifetime is 0.23 days (HRM) and 0.21 days (LRM), whereas it ranges from 0.2 days to 1.0 day in other global models. Between the HRM and LRM, some remarkable differences in the wet deposition flux of sea salt and the lifetime of BC of more than 20% are observed. These results suggest that aerosol-cloud-precipitation interactions through wet deposition are modified in the models with different horizontal resolutions.

- The simulated surface aerosols for fine-mode particles, such as sulfate, POM and BC, are generally in agreement with the measurements except in China, where the simulated results are underestimated. This suggests that the emission inventory in China is underestimated or the 14-km grid spacing is not sufficient to resolve such high concentrations in highly dense urban areas.

- The simulated surface aerosols for dust are generally in agreement in the measurements but not for sea salt. This is probably because the slight bias in the wind causes considerable bias in the sea salt emission flux; the 1.5 m s$^{-1}$ difference in the wind speed at a height of 10 m provides a 4-fold difference in the sea salt emission flux.

The verification of the relevant variables suggests that both the HRM and LRM can be applicable for a current global aerosol climate model. However, several important differences between the HRM and LRM have not been addressed in detail; therefore, section 4 discusses the following six issues to clarify the remaining issues relative to the LRM.

**What is the impact of the high-resolution grid on the coarse-grid average used in global aerosol models (section 4.1)?**

At the polluted sites during polluted months, the differences in the simulated aerosol concentrations between the HRM and LRM are estimated with NMBs of -19% for surface BC, -5% for surface sulfate and -3% for AOT. On a global scale, the variabilities in the AOT are calculated to be 28.5% (HRM) and 16.6% (LRM); the ratio between the HRM and LRM is 1.7. For CCN, COT and precipitation, the ratios are calculated to be 1.9, 3.5 and 2.8, respectively. This clearly shows how the HRM reproduces such variability in relation to extreme weather phenomena.

**What is the impact of the high-resolution grid on the reproducibility of BC and sulfate over the Arctic (section 4.2)?**

Unlike previous global models and our model with a lower grid spacing and a cloud macrophysics module, both the HRM and LRM succeed in reproducing the observed BC and sulfate over the Arctic. Between the HRM and LRM, the difference in the BC concentration reaches 30% in spring, and the HRM results are better than the LRM results. Our sensitivity experiments show the importance of considering cloud microphysics processes, including prognostic precipitation, which is one of the processes related to the wet deposition of aerosols, as suggested by previous studies.

**What is the impact of the high-resolution grid on the vertical distribution of aerosols (section 4.3)?**

The differences in the column burdens influence sulfate and carbonaceous aerosols, but the corresponding changes in the vertical distribution are not discussed in section 3. Using CALIOP/CALIPSO satellite observations, both the HRM and LRM generally reproduce the vertical profiles of the CALIOP-retrieved aerosol extinction coefficients worldwide. The issue regarding the overestimation of aerosols in the mid-troposphere among the current global aerosol models is not found extensively in this study. However, the use of a high-resolution grid does not resolve one of the major issues pertaining to the BC distribution—the underestimation in the mid- to upper troposphere over the Arctic. In the mid- and upper troposphere,

especially above 3 km, the HRM-simulated aerosol concentrations tend to be higher than the LRM-simulated concentrations, and the HRM results are overestimated compared to the CALIOP measurements. The analysis of the column burden indicates that this difference is caused by WIBC and sulfate. This and the finding that the differences in the vertical profiles over dusty regions are very small suggest that wet deposition processes rather than the transport characteristics cause the

differences in the vertical profiles between the HRM and LRM. This is also suggested by the validation of the vertical BC profile, which shows better performance of the LRM, whose lifetime of BC (4.97 days) is smaller than that in the HRM (6.37 days) and closer to the reference value (less than 5 days) from previous studies (Lund et al., 2018).

**How are the ARFs modified using the HRM (section 4.4)?**

The ARFs, i.e., IRFari, ERFari and ERFaci, estimated from both the HRM and LRM are within the uncertainties obtained

from the observations and other global models. The largest difference in the IRFari between the HRM and LRM is 0.05 W m$^{-2}$ at TOA for sulfate and 0.06 W m$^{-2}$ at the surface for WIBC. Although the differences in the IRFari due to BC between the HRM and LRM are found, both the HRM and LRM estimated positive IRFari due to WIBC and even WSBC seems to be underestimated compared to the observation-based studies by Oikawa et al. (2018) and Kinne (2019). Both the HRM-estimated and LRM-estimated IRFari values due to dust are more negative than those in other models because of the

overestimation of the SSA and the underestimation of the surface albedo over desert areas. The large negative dust-related IRFari is responsible for the underestimation of the surface IRFari compared to the satellite results. For the ERFaci due to anthropogenic aerosols, the difference between the HRM and LRM is 0.18 W m$^{-2}$, which is larger than that obtained for the ERFari. This study indicates that a higher-resolution model provides a lower ERFaci that is closer to the reference value shown in the IPCC-AR5. However, it should be noted that several important assumptions used in this study can affect the

ERFaci values, so this process must be further developed and refined to properly estimate the ERFaci.

**Is the difference between the HRM and LRM larger than the interannual variability obtained by the 3-year integration (section 4.5)?**

The interannual variability is mainly reflect in the winds over land and RH, which cause relatively larger variabilities in dust (emission flux, column burden and AOT) and sea salt (mainly AOT). As a result, the total AOT and IRFari under clear-sky

conditions and for longwave are more influenced by the interannual variabilities than those by the horizontal resolution. This

suggests that the clouds are also significantly modulated by the interannual variabilities rather than the horizontal resolution. However, the results related to sulfate, POM and BC are strongly influenced by the horizontal resolution compared to the interannual variability, and discussions of the impacts of different horizontal grid spacings on these parameters can be facilitated using only a 1-year integration.

Finally, the following question "How high is the calculation cost when using the HRM?" is considered. This answer is balanced by the precision of the aerosol simulation. As the computational cost is shown in Table S9 in the supplement, the computer resources required by the HRM are more than ten times higher (theoretically 16 times, but approximately 10 times using the K computer, which is a high-performance computing resource with relatively high memory performance) than that required by the LRM when using the same supercomputer with the same number of processers. When focusing on extreme

phenomena related to clouds and precipitation and ACIs, a 14-km grid spacing (or finer) is needed to clearly resolve the scientific questions addressed in this study. In this case, various tuning parameters associated with the aerosol distributions using the LRM (56-km grid) can be directly applied to the HRM (14-km grid) simulations, as we did in this study. In contrast, when focusing on the general circulations of aerosols and related gases, a 56-km grid spacing with a cloud microphysics module is sufficient, and the results are generally similar to those with a 14-km grid spacing (with a difference of 10% on a

global average), but apparent differences are found in aerosol wet deposition between the different resolutions. If the available computational resources greatly increase in the near future, we hope these suggestions will become helpful for all modelers.

**Code and data availability**

The source codes of NICAM.16 can be obtained upon request under the general terms and conditions (http://nicam.jp/hiki/?Research+Collaborations). The data that support the findings of this study can be archived with DOI:10.5281/zenodo.3687323.

**Author contribution**

DG designed and operated the numerical experiments and analyses. YS, HY and KS coordinated the model configuration and prepared the external conditions of the experiments. RK, EO, and TMN prepared the observational datasets for the model evaluation. TN submitted the proposal for the computational resources. DG wrote the initial draft of the paper, and all coauthors participated in discussions over the results and commented on the original manuscript.

**Competing interests**

The authors declare that they have no conflicts of interest.

**Acknowledgements**

We acknowledge the developers and administrators of the NICAM (http://nicam.jp/), SPRINTARS (https://sprintars.riam.kyushu-u.ac.jp/indexe.html), and MODIS (https://modis.gsfc.nasa.gov/) and the relevant PIs of the AERONET (https://aeronet.gsfc.nasa.gov/), SKYNET (https://www.skynet-isdc.org/) and CARSNET sites. We are grateful to the NCEP-FNL group, the NCEP/National Weather Service/National Oceanic Atmospheric Administration (NOAA)/U.S. Department of Commerce (2000), the NCEP FNL Operational Model Global Tropospheric Analyses (continuing from July 1999), https://doi.org/10.5065/D6M043C6, and the NCAR Research Data Archive, Computational and Information Systems Laboratory, Boucher, Colo. (updated daily), last accessed 4 January 2020. The CERES datasets were obtained from the NASA LaRC Atmospheric Science Data Center. Some of the authors were supported by the Environment Research and Technology Development Fund (S-12) of the Environmental Restoration and Conservation Agency, Japan, and by JSPS KAKENHI grants (26740010, 15K17766, 17H04711, and 19H05669). Additionally, we were supported by the following projects: the Ministry of Environment (MOE)/GOSAT, the Japan Science and Technology (JST), CREST/EMS/TEEDDA, JAXA/EarthCARE, JAXA/GCOM-C, and NIES. The model simulations were performed using supercomputers: the RIKEN/K computer (hp160004, hp160231, hp170017, hp170232, hp180012, and hp180181), NIES/NEC SX-ACE and

JAXA/JSS2. We also acknowledge Drs. H. Matsui (Nagoya University), T. Seiki (JAMSTEC) and K. Ikeda (NIES) for their discussions and Profs. Y. Kondo (National Institute of Polar Research in Japan), M. Koike (University of Tokyo), and N. Moteki (University of Tokyo) and the NOAA Black Carbon Group for providing us with their aircraft BC measurements. Global maps in the figures are drawn using Grid Analysis and Display System (GrADS) (http://cola.gmu.edu./grads/).

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

Table 1. Details of the observation datasets

| Name | Product | Variables | Region | Period | Reference |
|---|---|---|---|---|---|
| **MODIS/Terra (MOD) and MODIS/Aqua (MYD)** | Satellite | Cloud optical thickness (COT), cloud fraction (CF), aerosol optical thickness (AOT) | Global (1°×1°) | 2012-2014 | Collection 6 for both clouds and aerosols retrieved from NASA |
| **CALIOP/CALIPSO** | | Vertical extinction coefficient for aerosols | Global (1°×1°) | 2012-2014 | Version 3 (Winker et al., 2013) |
| **CERES** | | Top-of-atmosphere radiation fluxes | Global (1°×1°) | 2012-2014 | CERES_EBAF_Ed2.8 provided by NASA/LaRC (Langley Research Center) Hampton |
| **NCEP** | Reanalysis | U and V (wind speed components) at a height of 10 m | Global (2.5°×2.5°) | 2012-2014 | NCEP/NCAR Reanalysis 1: Surface Flux |
| **GPCP** | | Precipitation | Global (2.5°×2.5°) | 2012-2014 | Version 2.3 by Adler et al. (2003) |
| **AERONET** | In situ measurement | AOT | Global | 2000-2015 | Level 2 daily version 2; accessed on 2015/06/27; Holben et al. (1998) |
| **SKYNET** | | | Asia and New Zealand | 2005-2015[#1] | Nakajima et al. (1996) |
| **CARSNET** | | | China | 2002-2013 | Che et al. (2015) |
| **BSRN** | | Surface solar radiation (SSR) and direct and diffuse radiation fluxes at the surface | Global | 2008-2012 | Ohmura et al. (1998) |
| **IMPROVE** | | Aerosol mass concentration at the surface | United States | 2006-2015[#1] | IMPROVE (Interagency Monitoring of Protected Visual Environments) |
| **EMEP** | | | Europe | 2007-2015[#1] | WMO Global Atmosphere Watch, World Data Center for Aerosols |
| **EANET** | | | Asia | 2005-2013[#1] | EANET (Acid Deposition Monitoring Network in East Asia; http://eanet.asia) |
| **CAWNET** | | | China | 2006-2007 | Zhang et al. (2012) |
| **University of Miami, US** | | | Global | Some in the 1980s and others after 2000 | Prospero et al. (1989); Liu et al. (2007) |
| **EMEP and WDCS** | | BC mass concentrations at the surface | Alert, Zeppelin, Barrow | 2007-2011 | http://ebas.nilu.no/ |
| **CABM** | | Sulfate mass concentrations at the surface | Alert | 2000-2006 | Canadian Aerosol Baseline Measurement (CABM) program |

| | | | | | |
|---|---|---|---|---|---|
| **EMEP** | | | Zeppelin | 2005-2013 | http://ebas.nilu.no/ |
| **Eckhardt et al. (2015)** | | | Barrow | 2008-2009 | Eckhardt et al. (2015) |
| **HIPPO** | Aircraft measurement | BC mass concentrations | Pacific Ocean | January, March, June, August and November in 2009 | High-performance Instrumented Airborne Platform for Environmental Research (HIAPER) Pole-to-Pole Observations (HIPPO) campaign (Schwarz et al., 2010; Wofsy et al., 2012) |
| **AFORCE** | | | East Asia | March-April, 2012 | Aerosol Radiative Forcing in East Asia (A-FORCE) campaign (Oshima et al., 2012) |
| **ARCTAS-A** | | | Arctic | March-April, 2008 | Arctic Research of the Composition of the Troposphere from Aircraft and Satellites (ARCTAS) campaign (Jacob et al., 2010) |
| **ARCTAS-B** | | | Arctic and North America | July-August, 2008 | |

[#1] The period depends on the site.

Table 2. Global aerosol budgets simulated by the HRM and LRM

| Species | Parameter | HRM | LRM | DIF* | Reference |
|---|---|---|---|---|---|
| **Dust** | Column [Tg] | 27.08 | 27.01 | 0 | 15.8 (6.8-29.5)[k],19.20 (11.5-26.9)[a], 28.5[b] |
| | Emission [Tg yr$^{-1}$] | 1805 | 1911 | 6 | 1123 (514-4313)[k],1840 (938-2742)[a], 2677[b] |
| | Dry Deposition [Tg yr$^{-1}$] | 342 | 363 | 6 | 396 (37-2791)[k] |
| | Grav. Deposition [Tg yr$^{-1}$] | 634 | 663 | 5 | 314 (22-2475)[k] |
| | Wet Deposition [Tg yr$^{-1}$] | 825 | 880 | 7 | 357 (295-1382)[k] |
| | Lifetime [Day] | 5.49 | 5.17 | -6 | 3.9[b], 4.14 (2.36-5.92)[a], 4.6 (1.6-7.1)[k] |
| **Sea salt** | Column [Tg] | 5.60 | 5.42 | -3 | 5.62[l], 6.8[c], 7.52 (3.5-11.6)[a], 13.6[b] |
| | Emission [Tg yr$^{-1}$] | 8856 | 9624 | 9 | 805 (378-1233)[e], 3529[l], 4015.5[c], 5039[b],10200[d], 16600±199%[a], |
| | Dry Deposition [Tg yr$^{-1}$] | 2272 | 2169 | -5 | 1313[l] |
| | Grav. Deposition [Tg yr$^{-1}$] | 1998 | 1951 | -2 | 327[l] |
| | Wet Deposition [Tg yr$^{-1}$] | 4586 | 5504 | 20 | 1889[l] |
| | Lifetime [Day] | 0.23 | 0.21 | -11 | 0.03-1.59[a], 0.48 (0.20-0.76)[a], 0.62[c], 0.80[l], 0.98[b] |
| **Sulfate** | Column [TgS] | 0.38 | 0.32 | -16 | 0.59 (0.34-0.93)[j], 0.66 (0.50-0.83)[a] |
| | Production [TgS yr$^{-1}$] | 58.4 | 56.7 | -3 | 37.6-61.1[l], 44.0[b] |
| | from the gas phase | 16.8 | 16.1 | -4 | 6.2[l] -17.4 [m] |
| | from the aqueous phase | 41.7 | 40.6 | -3 | 21.1[m]-58.8[l] |
| | Dry Deposition [TgS yr$^{-1}$] | 3.9 | 3.6 | -8 | 5.8-7.6[l] |
| | Grav. Deposition [Tg yr$^{-1}$] | 0.5 | 0.4 | -8 | 0.0[l] |
| | Wet Deposition [TgS yr$^{-1}$] | 52.0 | 50.4 | -3 | 31.8-53.5[l] |
| | Lifetime [Day] | 2.38 | 2.05 | -14 | 3.3[b], 4.12 (3.4-4.9)[a] |
| **POM** | Column [Tg] | 1.04 | 0.94 | -10 | 1.2[n], 1.6 (0.8-2.6)[i], 1.70 (1.24-2.16)[a], |
| | Emission [Tg yr$^{-1}$] | 82.2 | 81.9 | 0 | 96.6 (71.5-121.7)[a] |
| | Dry Deposition [Tg yr$^{-1}$] | 6.3 | 6.6 | 4 | approximately 15 (0.2-28)[i] |
| | Grav. Deposition [Tg yr$^{-1}$] | 3.7 | 3.9 | 5 | |
| | Wet Deposition [Tg yr$^{-1}$] | 72.6 | 71.4 | -2 | approximately 90 (approximately 50-140)[i] |
| | Lifetime [Day] | 4.60 | 4.17 | -9 | 5.3[n], approximately 6 (approximately 4-8)[i], 6.54 (4.77-8.31)[a] |
| **BC** | Column [Tg] | 0.13 | 0.10 | -23 | 0.11[b], 0.22[n], 0.24 (0.14-0.34)[a] |
| | Emission [Tg yr$^{-1}$] | 7.3 | 7.3 | -1 | 11.9 (9.2-14.6)[a] |
| | Dry Deposition [Tg yr$^{-1}$] | 0.8 | 0.8 | -1 | |
| | Grav. Deposition [Tg yr$^{-1}$] | 0.2 | 0.2 | 1 | |
| | Wet Deposition [Tg yr$^{-1}$] | 6.3 | 6.3 | -1 | |
| | Lifetime [Day] | 6.37 | 4.96 | -22 | <5 [f,g], 5.0[b], 6.4[n], 7.12 (4.77-9.47)[a], 7.4[h] |

| | | | | | |
|---|---|---|---|---|---|
| **WSBC** | Column [Tg] | 0.06 | 0.05 | -11 | 0.19[n] |
| | Emission [Tg yr$^{-1}$] | 4.5 | 4.5 | -1 | |
| | Dry Deposition [Tg yr$^{-1}$] | 0.4 | 0.4 | 3 | |
| | Grav. Deposition [Tg yr$^{-1}$] | 0.2 | 0.2 | 1 | |
| | Wet Deposition [Tg yr$^{-1}$] | 3.9 | 3.9 | -1 | |
| | Lifetime [Day] | 4.78 | 4.29 | -10 | 6.4[n] |
| **WIBC** | Column [Tg] | 0.07 | 0.05 | -33 | 0.03[n] |
| | Emission [Tg yr$^{-1}$] | 2.8 | 2.8 | -1 | |
| | Dry Deposition [Tg yr$^{-1}$] | 0.4 | 0.4 | -4 | |
| | Grav. Deposition [Tg yr$^{-1}$] | 0.0 | 0.0 | -6 | |
| | Wet Deposition [Tg yr$^{-1}$] | 2.4 | 2.4 | 0 | |
| | Lifetime [Day] | 8.95 | 6.04 | -33 | 1.0[n], 1.0-1.7[o], 9.6 (w/o aging)[o] |

* DIF is defined as (LRM-HRM)/HRM in percent.

[a] Textor et al. (2006); [b] Matsui and Mahowald (2017); [c] Bian et al. (2019); [d] Grythe et al. (2014); [e] Partanen et al. (2014); [f] Lund et al. (2018); [g] Samset et al. (2014); [h] Shindell et al. (2008); [i] Tsigaridis et al. (2014); [j] Myhre et al. (2013); [k] Huneeus et al. (2011); [l] Takemura et al. (2000); [m] Goto et al. (2011); [n] Chung and Seinfeld (2002); [o] Goto et al. (2012)

Table 3. IRFari [1] at the top of atmosphere (TOA) and the surface with the uncertainties[2] in units of W m$^{-2}$

| Wavelength | Species | All-sky | | Clear-sky | |
|---|---|---|---|---|---|
| | | HRM | LRM | HRM | LRM |
| **Top of Atmosphere (TOA)** | | | | | |
| **SW+LW** | Dust | -0.708 (±0.002) | -0.721 (±0.009) | -0.907 (±0.002) | -0.947 (±0.010) |
| | Sea salt | -0.474 (±0.000) | -0.470 (±0.002) | -0.735 (±0.001) | -0.755 (±0.003) |
| | Sulfate | -0.440 (±0.001) | -0.392 (±0.002) | -0.663 (±0.001) | -0.606 (±0.003) |
| | intBC+POM | 0.052 (±0.000) | 0.057 (±0.001) | 0.010 (±0.000) | 0.009(±0.001) |
| | SOA | -0.227 (±0.000) | -0.209 (±0.002) | -0.335 (±0.001) | -0.312 (±0.003) |
| | extBC | 0.086 (±0.000) | 0.066 (±0.000) | 0.052(±0.000) | 0.046 (±0.000) |
| | All | -1.717 (±0.002) | -1.670 (±0.010) | -2.585 (±0.003) | -2.565 (±0.012) |
| **SW** | Dust | -0.843 (±0.002) | -0.869 (±0.010) | -1.084 (±0.003) | -1.134 (±0.012) |
| | Sea salt | -0.521 (±0.000) | -0.517 (±0.002) | -0.851 (±0.001) | -0.883 (±0.003) |
| | Sulfate | -0.466 (±0.001) | -0.415 (±0.002) | -0.703 (±0.001) | -0.641 (±0.003) |
| | intBC+POM | 0.049 (±0.000) | 0.055 (±0.001) | 0.006 (±0.000) | 0.006 (±0.001) |
| | SOA | -0.231 (±0.000) | -0.212 (±0.002) | -0.341 (±0.001) | -0.318 (±0.003) |
| | extBC | 0.084 (±0.000) | 0.065 (±0.000) | 0.050 (±0.000) | 0.045 (±0.000) |
| | All | -1.936 (±0.003) | -1.895 (±0.011) | -2.936 (±0.003) | -2.925 (±0.013) |
| **Surface** | | | | | |
| **SW+LW** | Dust | -1.158 (±0.003) | -1.222 (±0.014) | -1.336 (±0.003) | -1.414 (±0.015) |
| | Sea salt | -0.296 (±0.000) | -0.306 (±0.001) | -0.293 (±0.000) | -0.304 (±0.002) |
| | Sulfate | -0.380 (±0.001) | -0.334 (±0.002) | -0.564 (±0.001) | -0.510 (±0.003) |
| | intBC+POM | -0.359 (±0.001) | -0.335 (±0.003) | -0.410 (±0.001) | -0.388 (±0.003) |
| | SOA | -0.316 (±0.001) | -0.292 (±0.002) | -0.415 (±0.001) | -0.388 (±0.003) |
| | extBC | -0.205 (±0.000) | -0.144 (±0.001) | -0.245 (±0.000) | -0.171 (±0.001) |
| | All | **-2.715** (±0.004) | -2.633 (±0.017) | -3.260 (±0.005) | -3.176 (±0.020) |
| **SW** | Dust | **-1.447** (±0.004) | -1.552 (±0.019) | -1.668 (±0.004) | -1.793 (±0.021) |
| | Sea salt | -0.535 (±0.000) | -0.530 (±0.002) | -0.862 (±0.001) | -0.892 (±0.003) |
| | Sulfate | -0.450 (±0.001) | -0.399(±0.002) | -0.673 (±0.001) | -0.613 (±0.003) |
| | intBC+POM | -0.371 (±0.001) | -0.342 (±0.003) | -0.424 (±0.001) | -0.399 (±0.004) |
| | SOA | -0.327 (±0.001) | -0.303 (±0.003) | -0.433 (±0.001) | -0.406 (±0.003) |
| | extBC | -0.208 (±0.000) | -0.146 (±0.001) | -0.248 (±0.000) | -0.173 (±0.001) |
| | All | **-3.330** (±0.005) | -3.272 (±0.022) | **-4.315** (±0.006) | -4.277 (±0.024) |

[1] The estimated IRFari are 1-year averages due to the limited computer resource. [2] The uncertainties are given as the global confidence intervals with a significance threshold of 95%.

(a) Observation sites for aerosol mass   (b) Observation sites for AOT and radiation

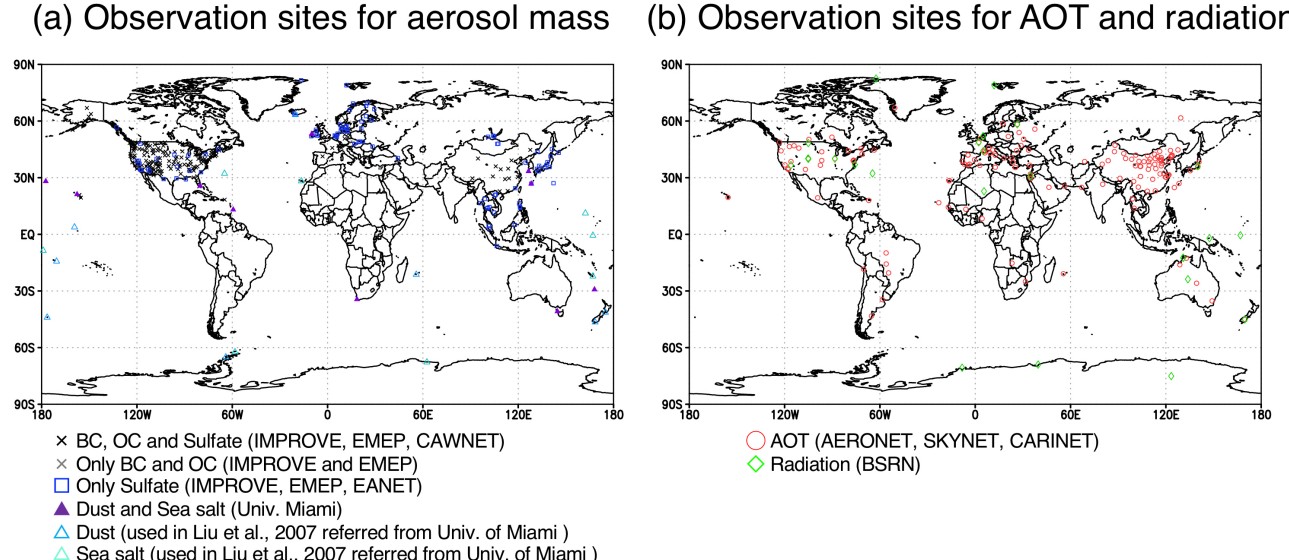

× BC, OC and Sulfate (IMPROVE, EMEP, CAWNET)
× Only BC and OC (IMPROVE and EMEP)
□ Only Sulfate (IMPROVE, EMEP, EANET)
▲ Dust and Sea salt (Univ. Miami)
△ Dust (used in Liu et al., 2007 referred from Univ. of Miami )
△ Sea salt (used in Liu et al., 2007 referred from Univ. of Miami )

○ AOT (AERONET, SKYNET, CARINET)
◇ Radiation (BSRN)

**Figure 1: Global distribution of observation sites used in the model evaluation. Detailed information on these sites is provided in Tables S1-S4 in the supplementary material.**

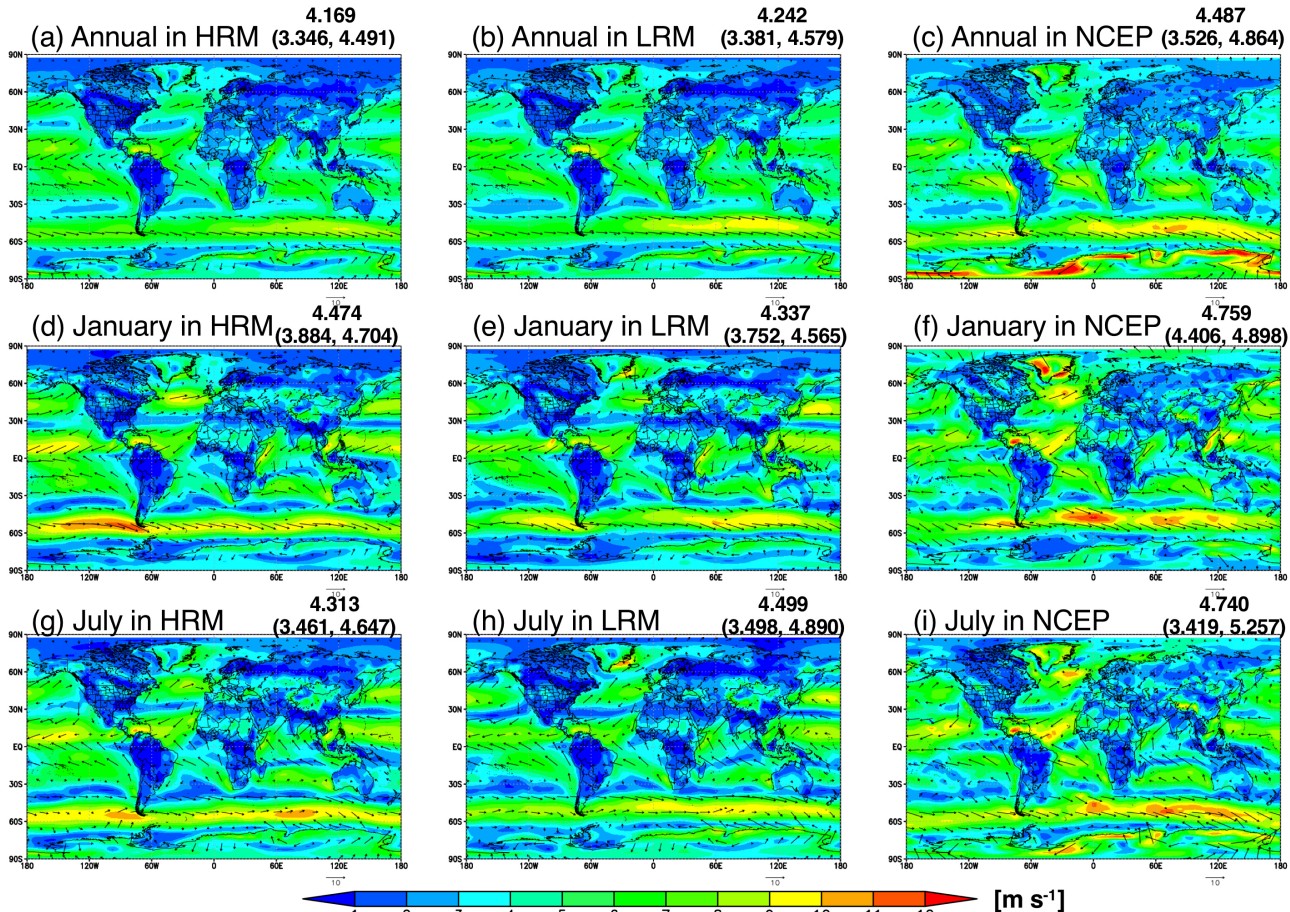

**Figure 2: Global distributions of the annual, January and July averages of the wind speed at a height of 10 m simulated by the HRM and LRM and reanalyzed by the NCEP. The colors and arrows represent the wind speed and wind direction, respectively. The model results in both the HRM and the LRM are horizontally interpolated onto the NCEP grids (2.5°×2.5°). The numbers shown in the upper-right corner in each panel represent the global averages (90°S-90°N); those in brackets represent the global land and ocean averages. All units are in m s$^{-1}$.**

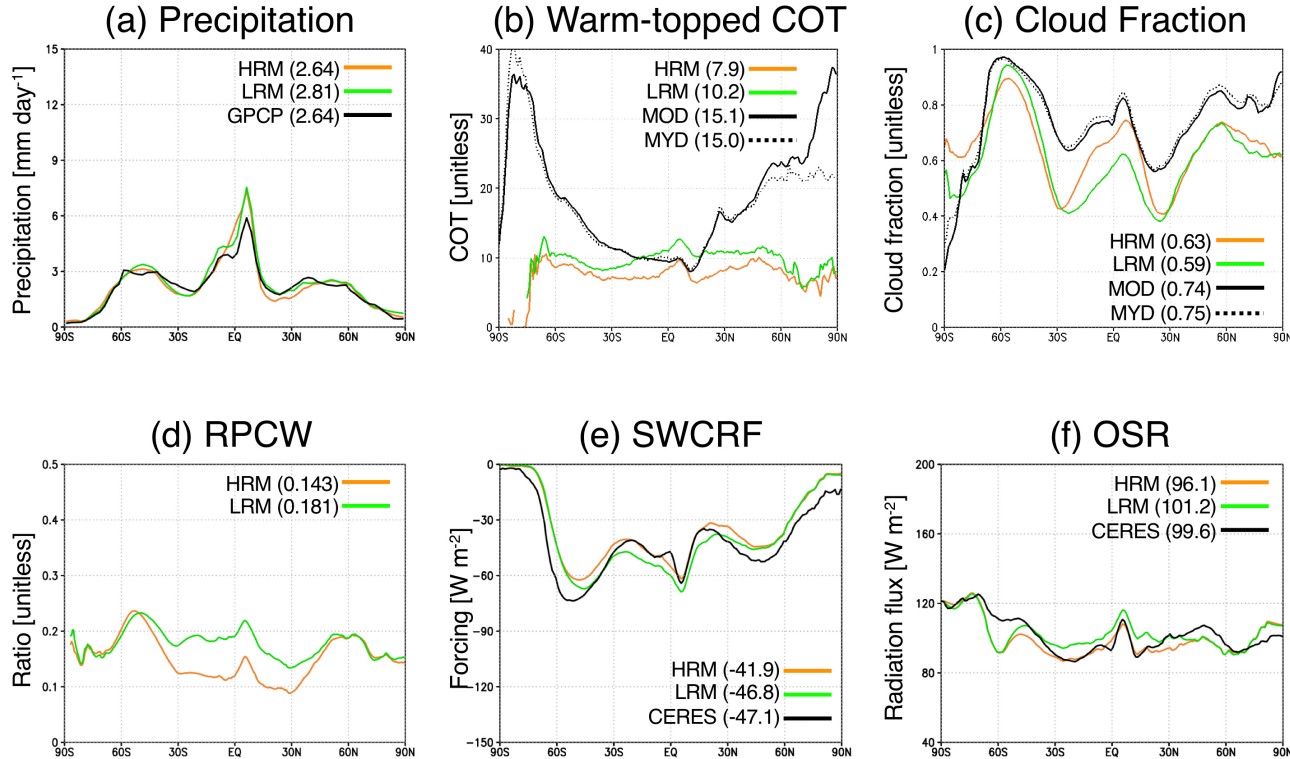

**Figure 3: Zonal distributions of the annual averages of (a) precipitation, (b) cloud optical thickness (COT) for warm-topped clouds, (c) cloud fraction (CF) in all types of clouds, (d) ratio of precipitation to total cloud water (RPCW) at a height of 2 km, (e) shortwave cloud radiative forcing (SWCRF) and (f) outgoing shortwave radiation flux (OSR) simulated by the HRM and LRM, reanalyzed by the GPCP only in (a), retrieved from both MODIS/Terra (MOD) and MODIS/Aqua (MYD) in (b) and (c), and estimated by CERES in (e) and (f). The annual averages of these variables except for CF and COT are calculated by a 3-year integration, whereas those in CF and COT are calculated by a 1-year integration using 6-hourly instantaneous clouds at 12:00 (local time) to more exactly compare them with the MODIS/Terra observation at approximately 10:30 (local time) and with the MODIS/Aqua observation at approximately 13:30 (local time). The numbers shown in the captions represent the global and annual averages for NICAM (HRM or LRM) and the reference data. The units are described in each panel.**

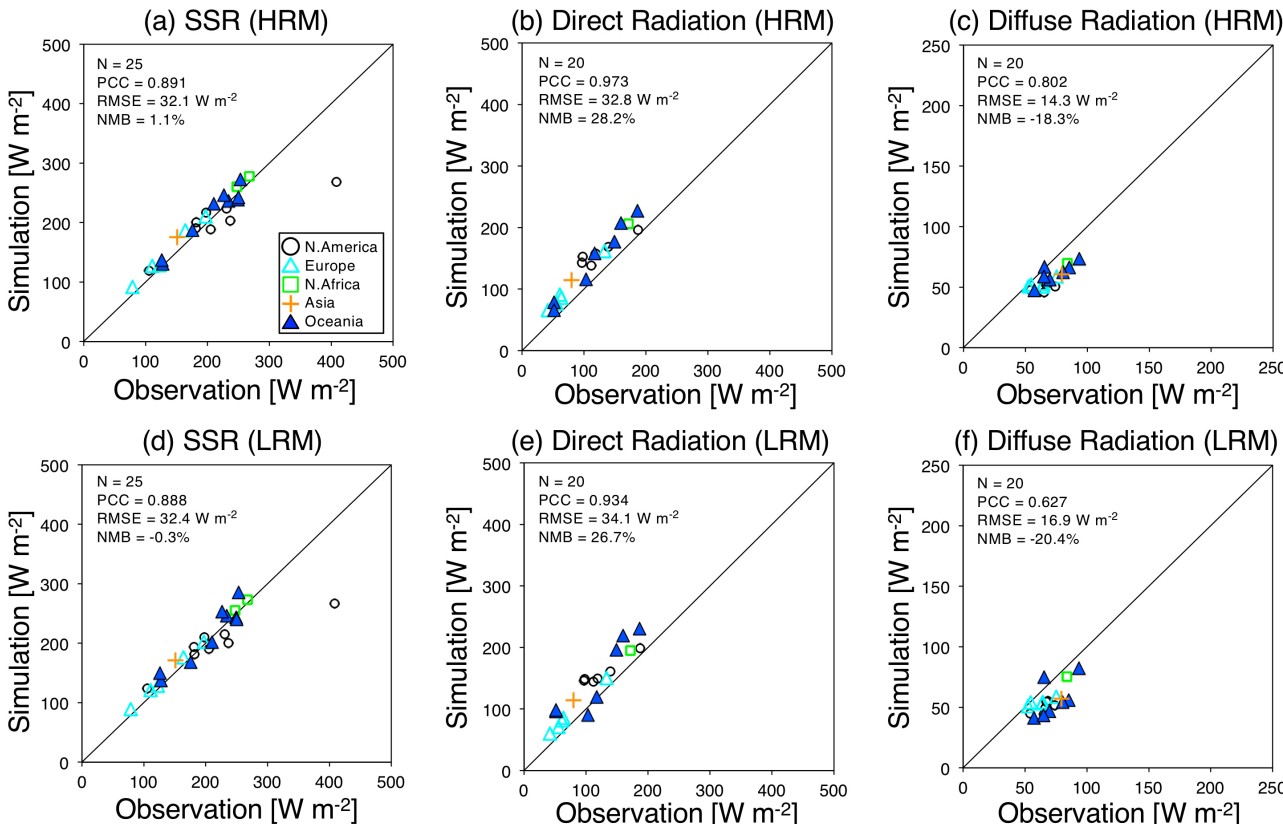

**Figure 4: Scatterplots of the (a,d) surface solar radiation (SSR), (b,e) direct and (c,f) diffuse radiation fluxes between the BSRN measurements and NICAM simulations (HRM and LRM) for global annual averages. The different colors and marks reflect the sites in the different regions explained in panel (a). The numbers located in the upper-left corner in each panel represent the statistical metrics: the sampling number (N), PCC, RMSE and NMB. All units are in W m$^{-2}$.**

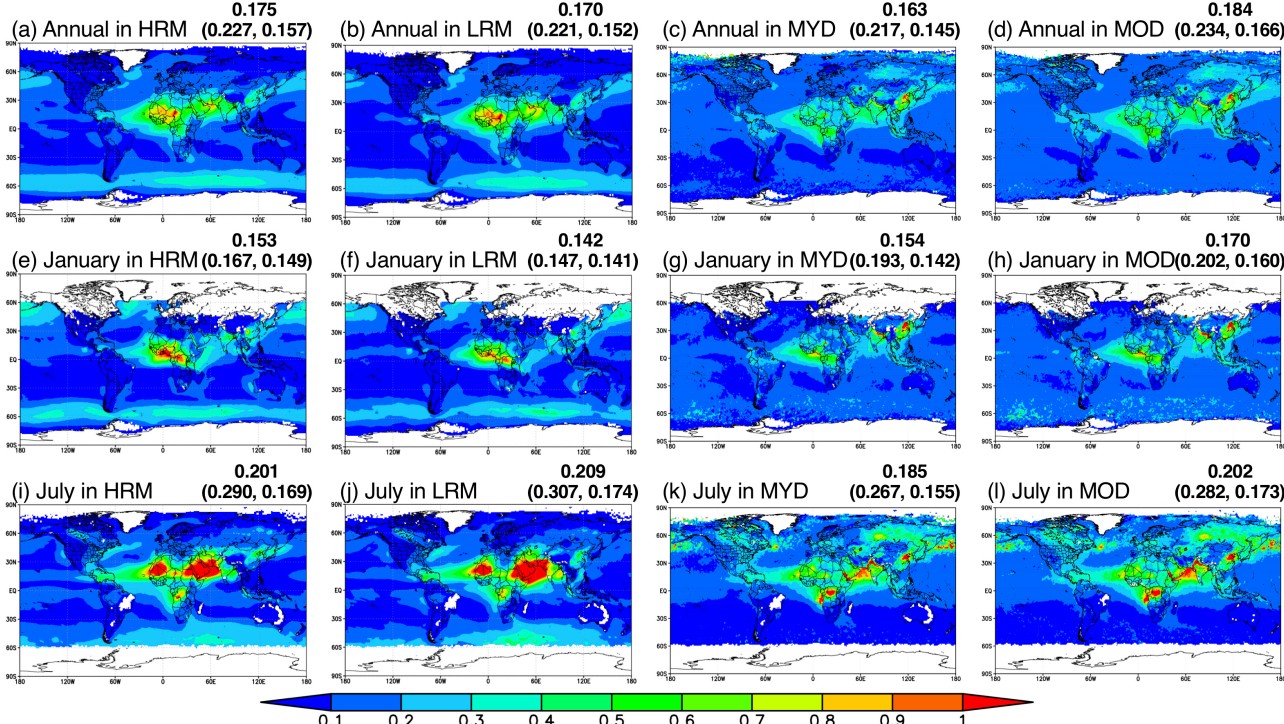

**Figure 5: Global distributions of the annual, January and July averages of the AOT simulated by the HRM and LRM and retrieved from MODIS/Aqua (MYD) and MODIS/Terra (MOD). The model results for both the HRM and LRM are horizontally interpolated onto the MODIS grids (1°×1°). The numbers shown in the upper-right corner in each panel represent the annual and semiglobal averages (60°S-60°N); those in brackets represent the global land and ocean averages.**

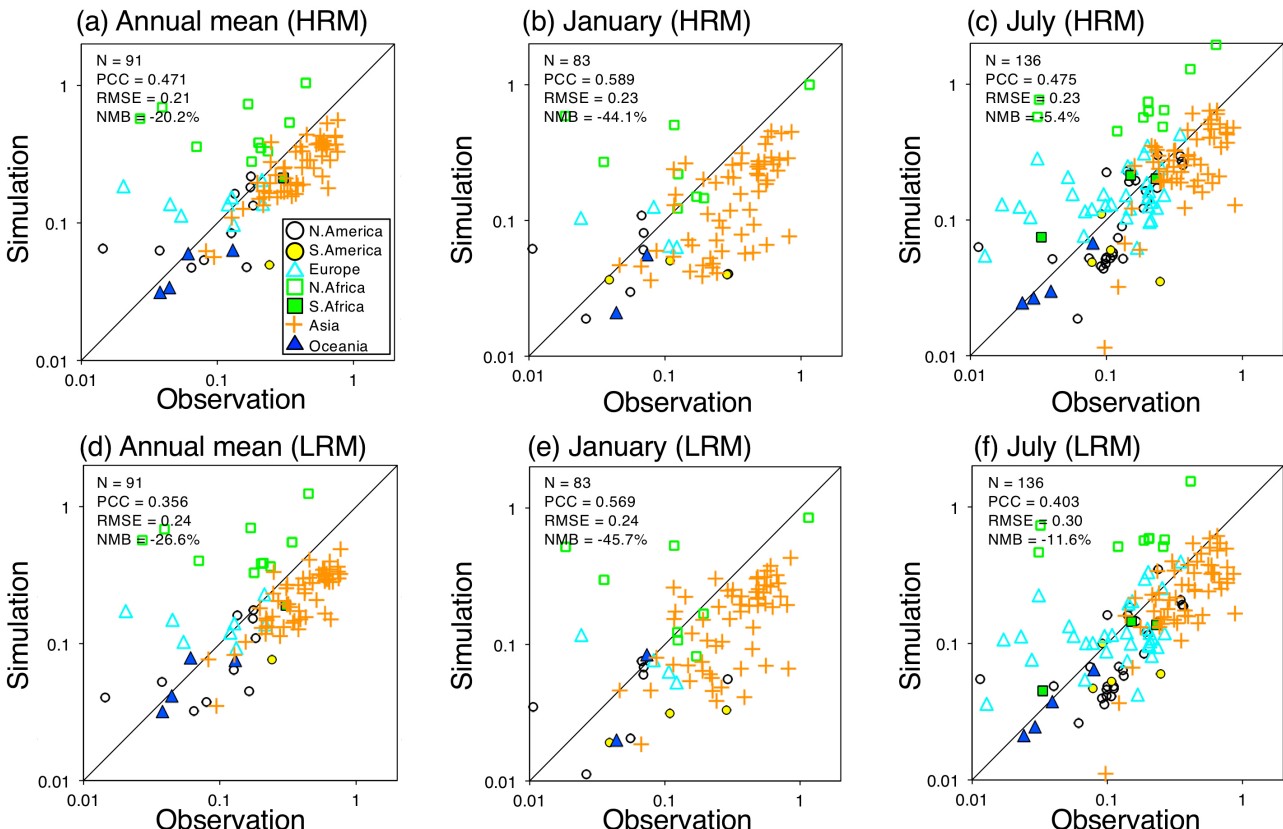

**Figure 6: Scatterplot of the AOT at a wavelength of 500 nm between satellite measurements (AERONET, SKYNET, and CARSNET) and the NICAM (HRM and LRM) simulations for the annual, January and July averages. The different colors and marks reflect the sites in the different regions explained in panel (a). The numbers located in the upper-left corner in each panel represent the statistical metrics: N, PCC, RMSE and NMB. The statistical metrics are also shown in Table S5. The sites used for the comparison are shown in Figure 1.**

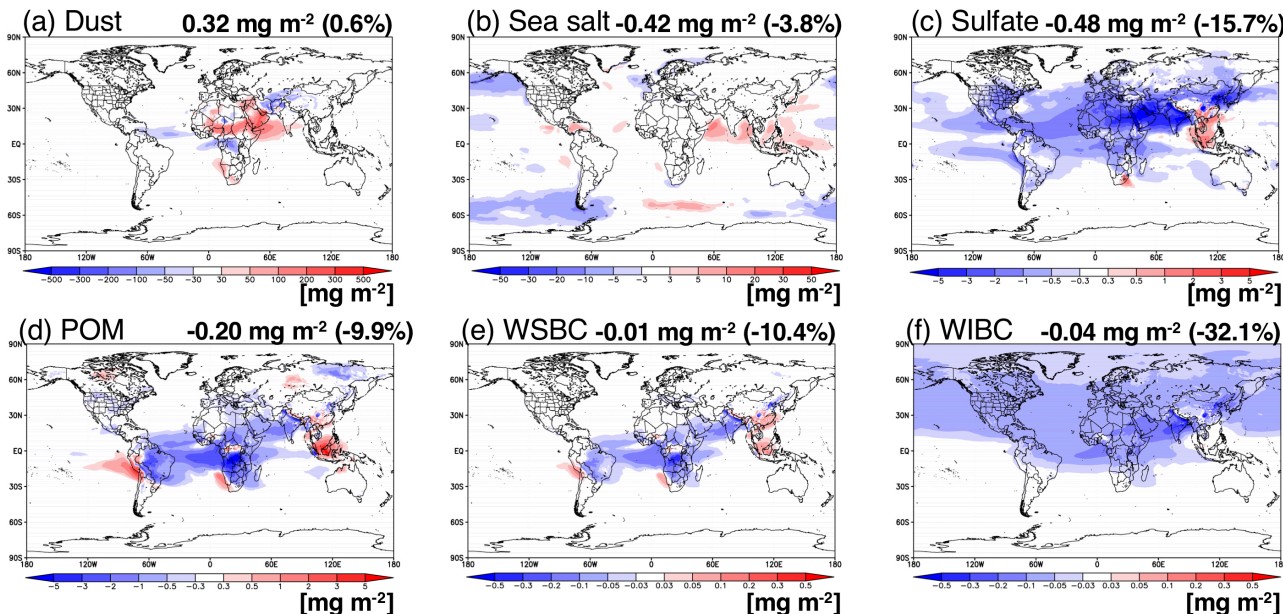

**Figure 7: Global distribution of the differences in the mass loadings of (a) dust, (b) sea salt, (c) sulfate, (d) POM, (e) WSBC and (f) WIBC between the HRM and LRM (LRM minus HRM) for the annual averages with a grid of 0.5°×0.5°. The numbers shown in the upper-right corner in each panel represent the annual and global averages of the difference in units of mg m⁻², and the numbers in brackets represent the annual and global averages of the relative difference in units of %.**

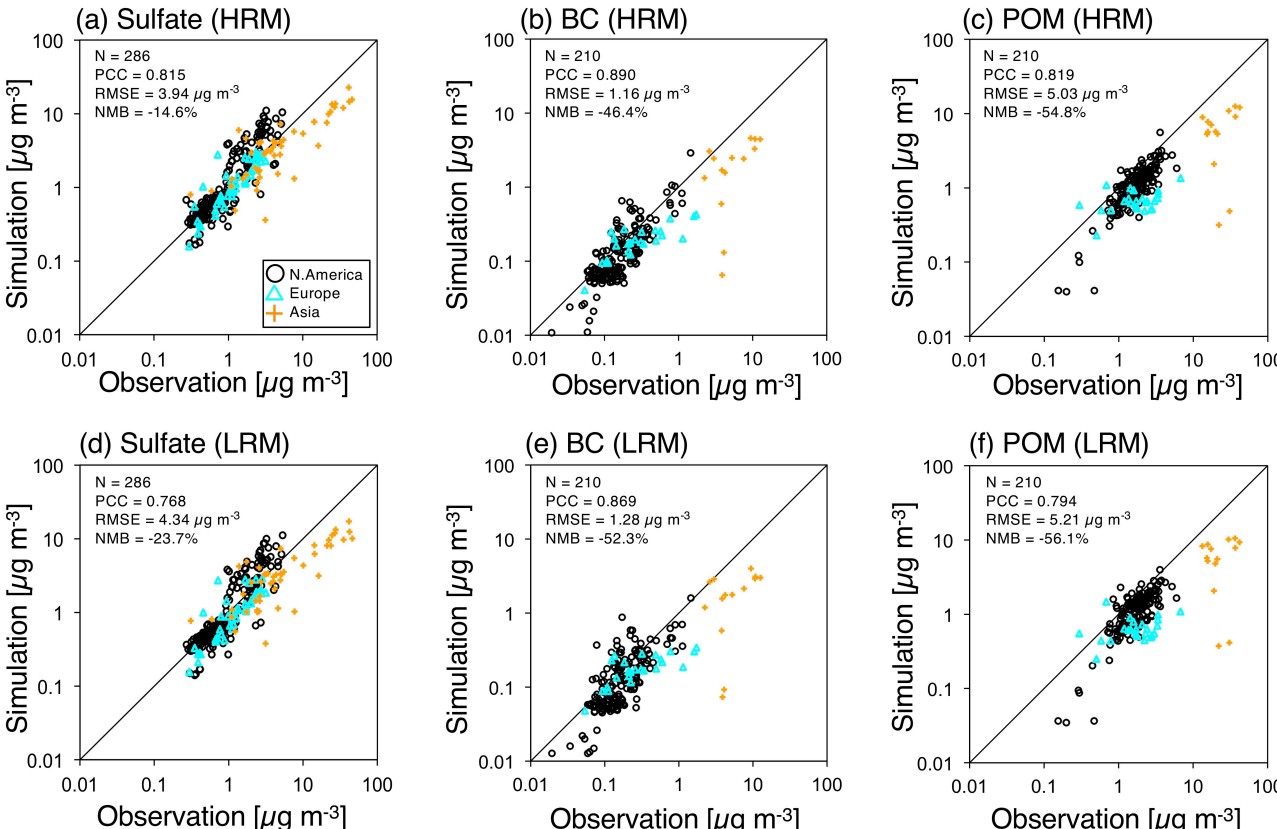

**Figure 8: Scatterplots of the surface aerosol mass concentrations (sulfate, BC and POM) between satellite measurements (IMPROVE, EMEP, EANET and CAWNET) and the NICAM (HRM and LRM) simulations for the annual averages. All units are in μg m⁻³. The different colors and marks reflect the sites in the different regions explained in panel (a). The numbers located in the upper-left corner in each panel represent the statistical metrics: N, PCC, RMSE and NMB. The statistical metrics are also shown in Table S6. The sites employed for the comparison are shown in Figure 1.**

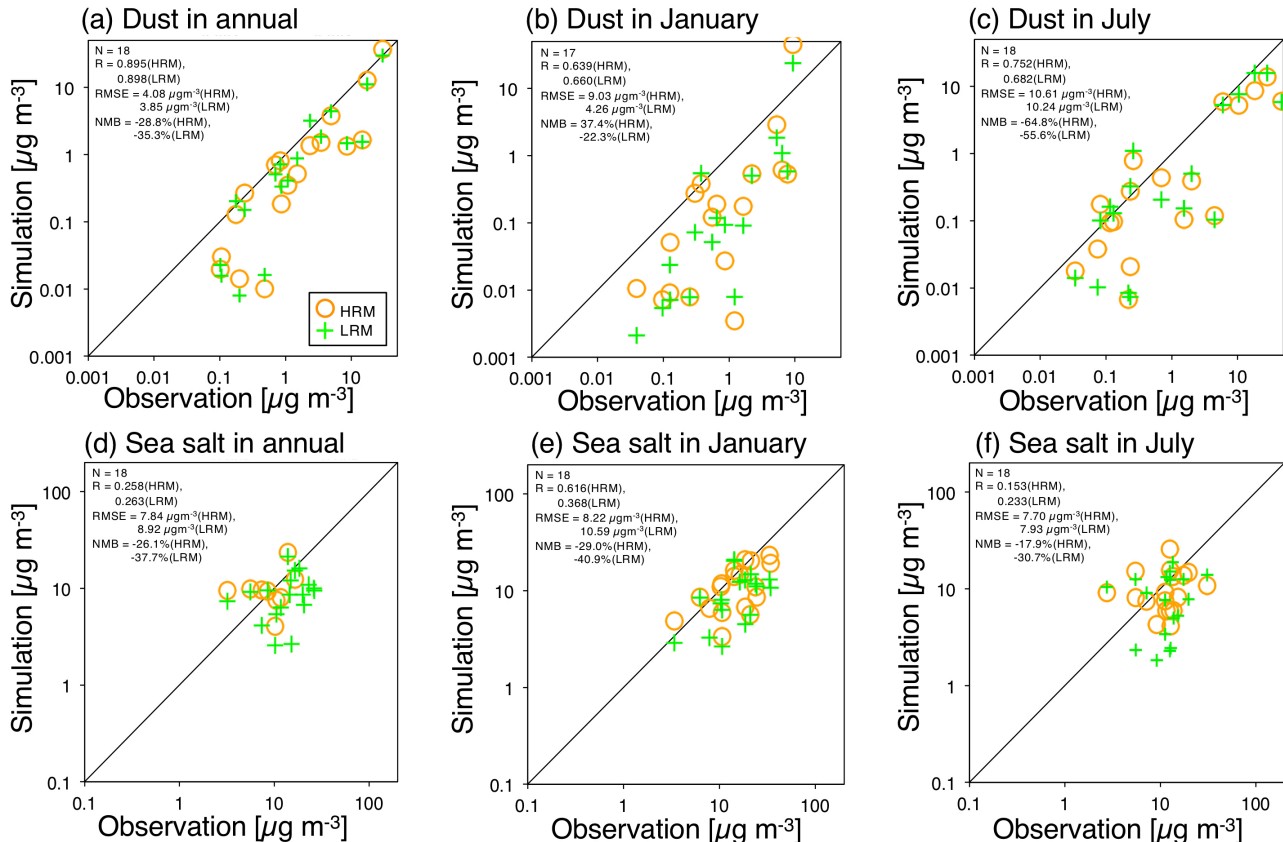

**Figure 9: Scatterplots of the surface aerosol mass concentrations (dust and sea salt) between the measurements (the network managed by the University of Miami) and NICAM simulations (HRM in orange and LRM in green) for the annual, January and July averages. All units are in μg m⁻³. The numbers located in the upper-left corner in each panel represent the statistical metrics: N, PCC, RMSE and NMB. The statistical metrics are also shown in Tables S7 and S8. The sites employed for the comparison are shown in Figure 1.**

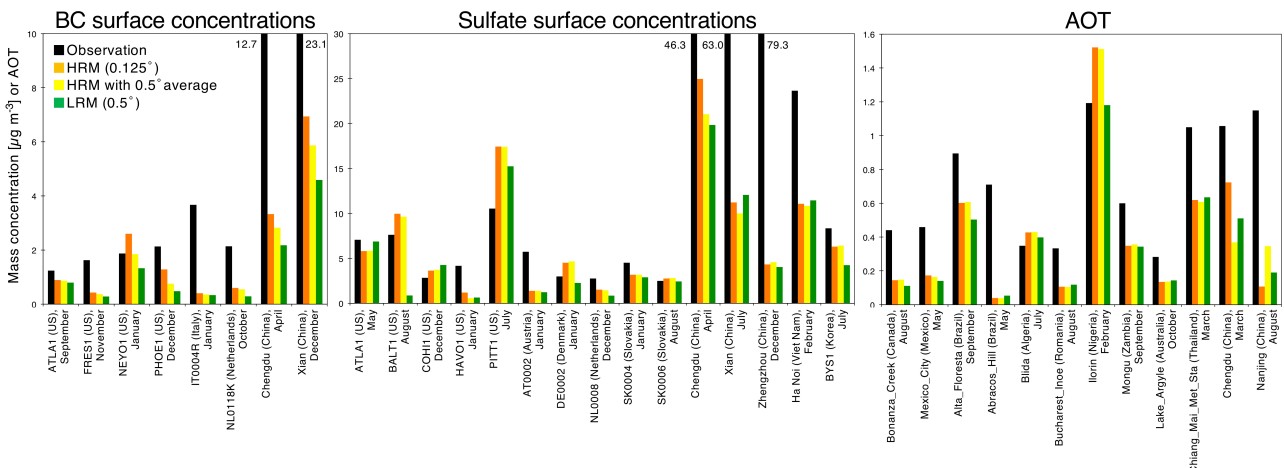

**Figure 10: Multiple comparisons of the BC and sulfate surface mass concentrations (μg m⁻³) and AOTs at the polluted sites using the HRM with an original grid of 0.125°×0.125°, the HRM with an interpolated grid of 0.5°×0.5° (denoted as 'HRM with 0.5° average'), the LRM with an original grid of 0.5°×0.5°, and the observations. The abscissa shows the selected sites, which were selected by choosing the highest values at the sites in each domain and month.**

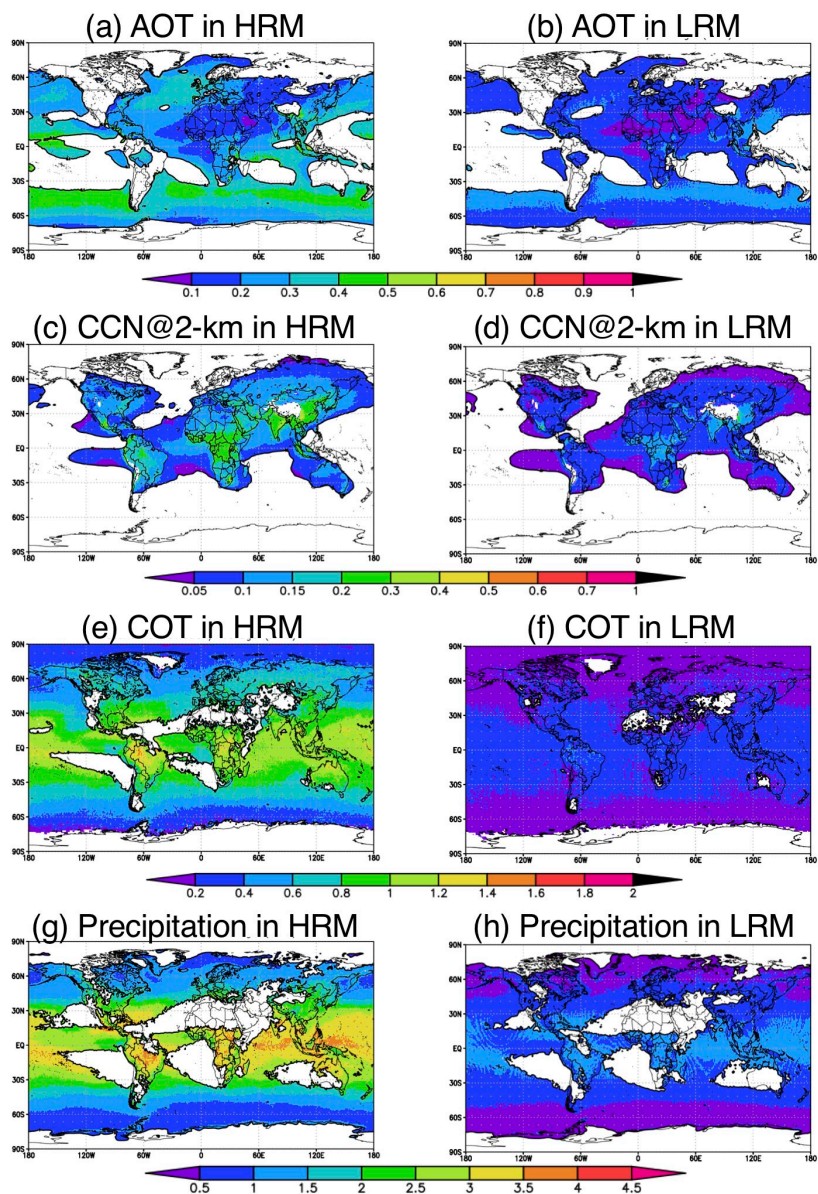

**Figure 11: Global distributions of the ratio of the standard deviation to the average for the (a,b) AOT, (c,d) CCN at a height of approximately 2 km, (e,f) COT and (g,h) precipitation on 1°×1° grids using the 6-hourly output of both the HRM and LRM for a 1-year integration period. All units are in %. The transparency represents lower absolute values of each parameter: AOTs of <0.1 in panels (a,b), CCN of <40 cm$^{-3}$ in panels (c,d), COTs of <5 in panels (e,f), and precipitation fluxes of <1 mm day$^{-1}$ in panels (g,h).**

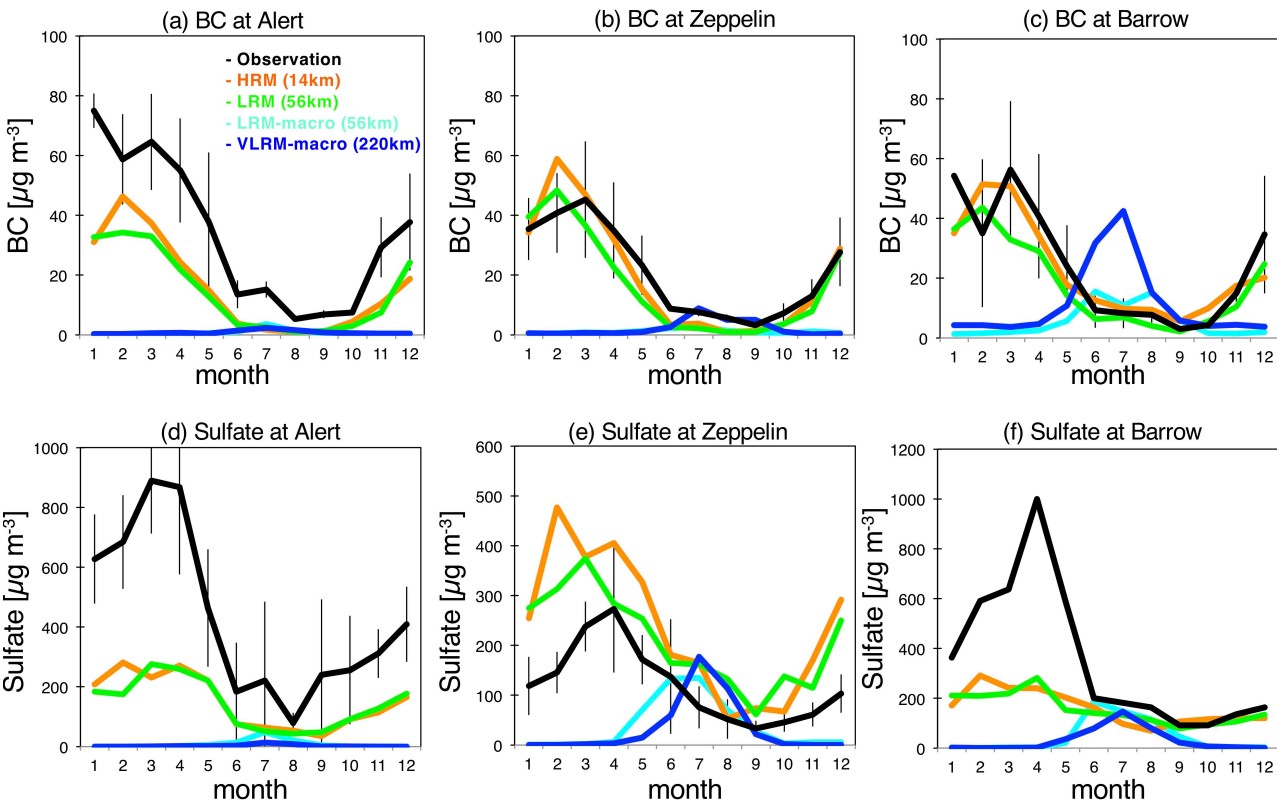

**Figure 12: Monthly averages of BC and sulfate concentrations simulated by the HRM, LRM, LRM with a cloud macrophysics module (LRM-macro, which is defined in section 2.1) and VLRM-macro (NICAM simulations using a horizontal grid spacing of 220 km with the cloud macrophysics module, which is defined in section 2.1) at three Arctic sites: Alert (62.3°W, 82.5°N), Zeppelin (11.9°E, 78.9°N) and Barrow (157.0°W, 71.3°N). The BC is measured as the equivalent BC at 530 nm by a particle soot absorption photometer (PSAP) for 2007-2011 under the EMEP and WDCS databases (http://ebas.nilu.no). The sulfate concentrations are averaged at Alert for 2000-2006 by the Canadian Aerosol Baseline Measurement (CABM) program, at Zeppelin for 2005-2013 by EMEP, and at Barrow for 2008-2009 by Eckhardt et al. (2015).**

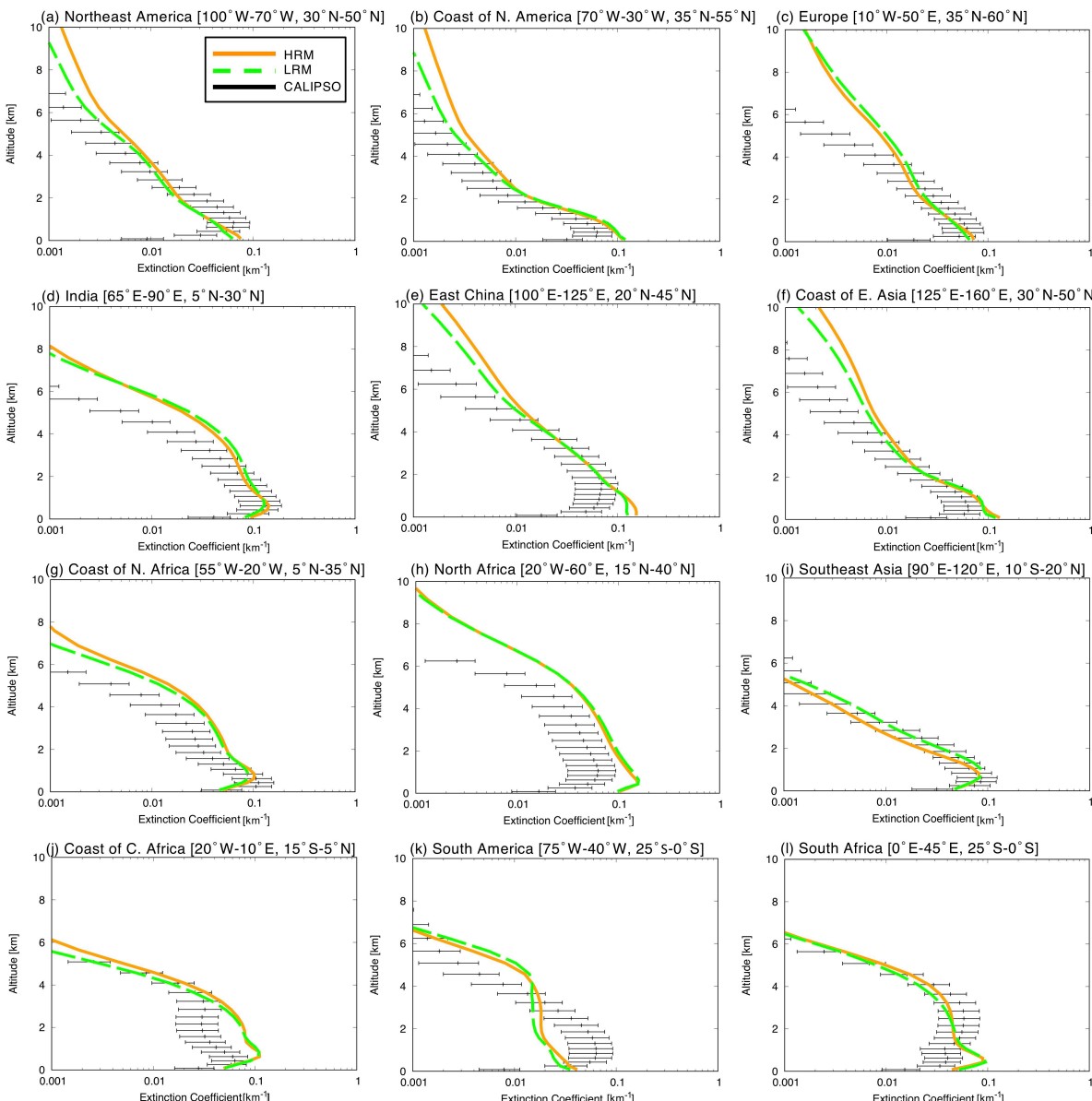

**Figure 13: Vertical distributions of the annually averaged aerosol extinction coefficients from the NICAM simulations (HRM in orange and LRM in green) and from CALIOP/CALIPSO observations (black) in 12 different regions. The definition of the region is based on Koffi et al. (2016) except for panels (i) Southeast Asia and (j) the coast of Central Africa. The CALIOP-retrieved results are shown as bars, which are the standard deviation of the results from a 3-year integration period.**

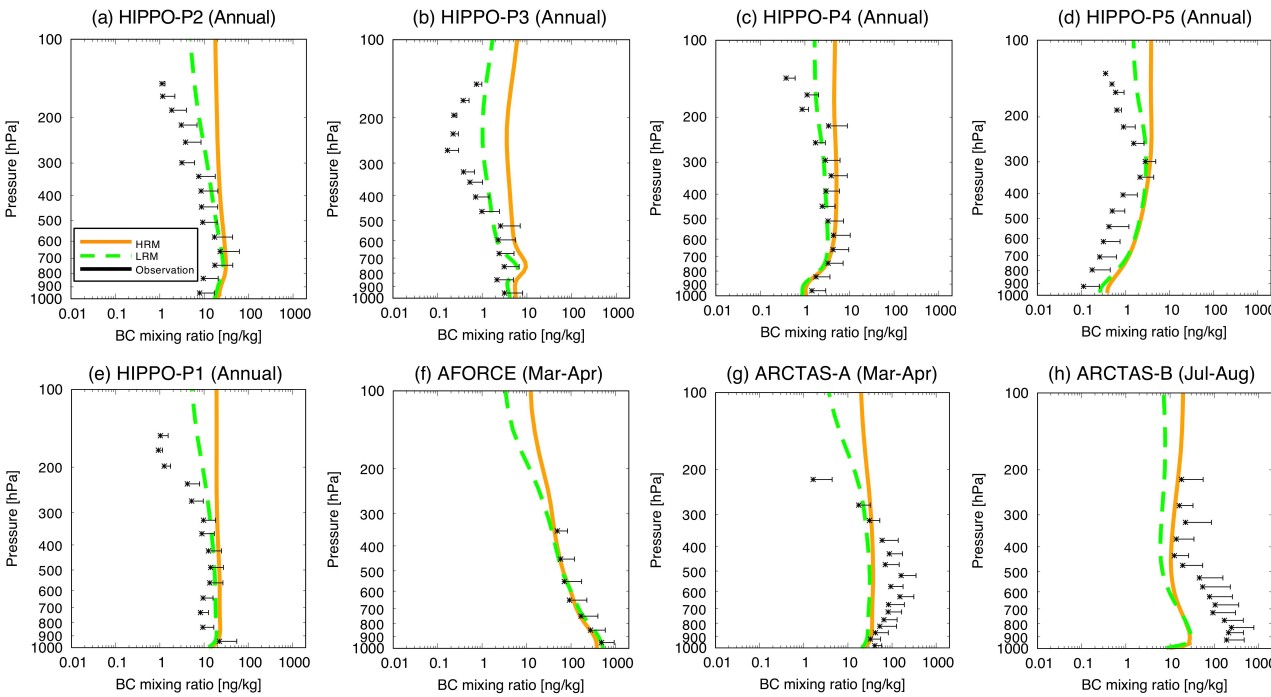

**Figure 14: Vertical distribution of the BC mass concentrations from the NICAM simulations (HRM in orange and LRM in green) and from flight campaign measurements (the groups by NOAA and the University of Tokyo) in various regions and seasons. The definitions of the target domain and period in each panel are as follows: (a) 20°N-60°N, 160°E-150°W, (b) 20°S-20°N, 160°E-150°W, (c) 60°S-20°S, 160°E-150°W, (b) 60°S-80°S, 160°E-150°W, and (e) 60°N-90°N, 160°E-150°W in annual averages of 2009, (f) 32°N-37°N, 122°E-126°E in March 2012 and 26°N-32°N, 126°E-132°E in April 2012, (g) 60°N-80°N, 165°W-70°W in March-April 2008 and (h) 45°N-87°N, 135°W-45°W in July-August 2008. The abscissa shows the mass concentration in units of μg m⁻³, and the ordinate shows the air pressure in units of hPa.**

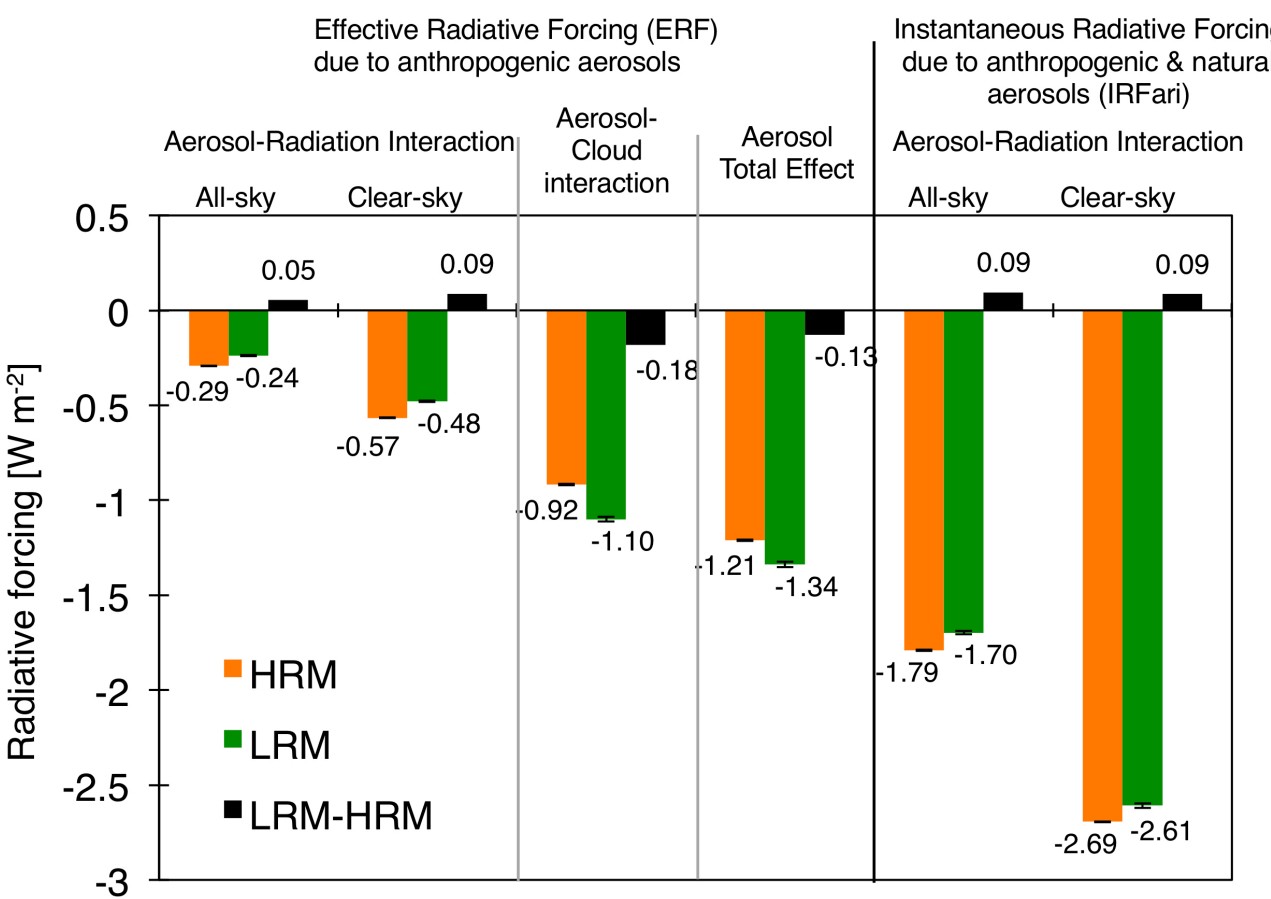

**Figure 15: Annual global average ARFs for both ARIs and ACIs, i.e., IRFari, ERFari, and ERFaci, against anthropogenic and total aerosols, i.e., anthropogenic and natural sources, under all-sky and clear-sky conditions at the TOA using the HRM (orange), the LRM (green) and the difference (LRM minus HRM in black). All units are in W m⁻². The uncertainties are given as the global confidence intervals with a significance threshold of 95%.**

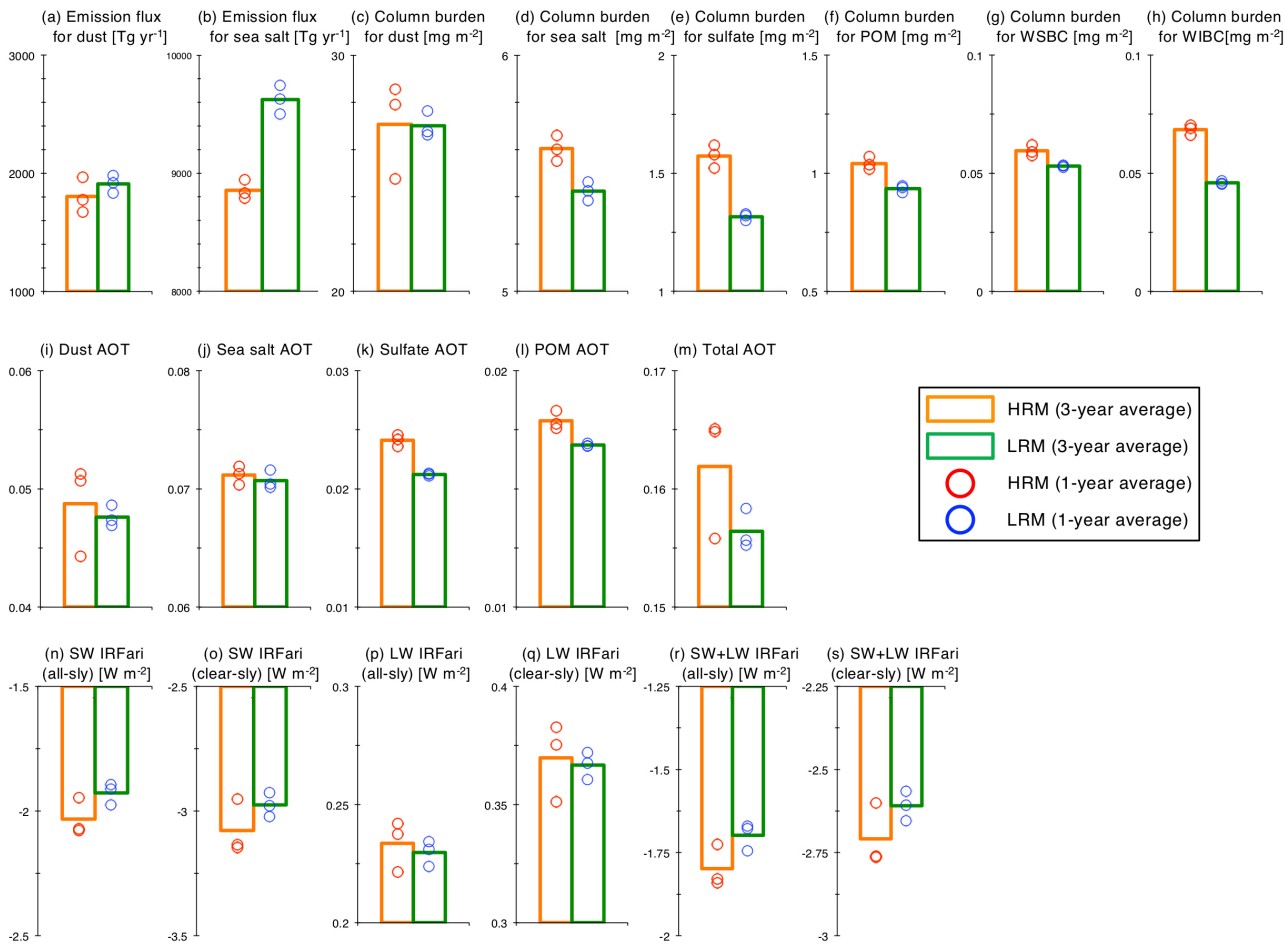

**Figure 16: Global annual averages of the emission fluxes (for dust and sea salt), column burdens (for dust, sea salt, sulfate, OC, WSBC and WIBC), AOTs (for dust, sea salt, sulfate, carbon and total amount) and direct ARF (IRFari) at the TOA (shortwave (SW), longwave (LW) and total (SW plus LW) under all-sky and clear-sky conditions) by the 3-year averages as well as each 1-year average in both the HRM and LRM.**