# Peer review of "Global aerosol simulations using NICAM.16 on a 14-km grid spacing for a climate study: Improved and remaining issues relative to a lower-resolution model"

_Geoscientific Model Development, 2020_

## Referee Comment (RC1) · Anonymous Referee #1 · 1 May 2020

This manuscript presents one of the first multi-year simulations with an aerosol–climate model at a resolution approaching the convective "grey zone", in this case using the NICAM atmospheric model with the SPRINTARS aerosol scheme at a grid spacing of 14km. The study compares the meteorological fields, aerosol mass and optical thickness, and aerosol radiative forcing, between this configuration and a lower-resolution version of the same model. They additionally consider the output from the high-resolution model coarsened to the lower resolution to distinguish the effects of simulation vs data resolution. In many respects, this is a therefore a significant and illuminating study and very much appropriate for publication in GMD. However, there is one major point as well as a number of minor points which should be addressed before acceptance:

**Major points**

The main area of concern with this study relates to the ability of a 3-year simulation to adequately capture interannual variability. The very large (and zero-containing) confidence intervals quoted up front in the abstract, e.g. $-91\%$ to $+18\%$ and $-49\%$ to $+223\%$ suggest that this is simply not a long enough period to adequately constrain what are presented as headline results. If this is to be presented as a major advance in its own right compared to single-year high-resolution simulations, rather than merely a technical step towards a long enough simulation to produce robust interannual statistics, this needs to be clearly quantified and justified in the study. Alternatively, it might be more appropriate to submit results when a longer and more statistically-significant dataset is available, if the conclusions drawn from 3 years do not add appreciable confidence to those from single-year studies.

There are also a very large number of figures in the paper, and I would strongly recommend condensing to a smaller number that illustrate the important results, and moving additional plots to the supplement if they do not contribute to the main conclusions.

**Minor points**

**p.2, line 7:** while 56km is a fine resolution compared to most global aerosol–*climate* models, it is coarser than some of those used for operational global aerosol *forecasting*, as collected in the ICAP initiative, for example.

**p.2, lines 12–13:** what does "column burden of the aerosol wet deposition" mean?

**p.2, line 13:** should this be "cloud to precipitation" rather than the other way around?

**p.2, lines 17–18:** it is not clear why "differences. . . between the different horizontal grid spacings are not explained simply by the grid size" follows from the results quoted above as suggested. Is this related to the coarsened-HRM data, which hasn't been mentioned yet at all in the abstract?

**p.2, line 25–p.3, line 1:** what are the confidence intervals on these ARF values, and therefore are the differences statistically significant?

**p.2, lines 4–6:** it is commonly found that tuning parameters require quite different values at different model resolutions; therefore any suggestion that tuning using the LRM can be applicable to the HRM or vice versa requires proper justification.

**p.8, line 2:** does this really mean monodisperse particles of a single fixed size as suggested, or the more usual unimodal size *distribution* with a fixedmode and width?

**p.10, line 10:** what is meant by "flux of the ARI"?

**p.9, lines 6–20:** given that a pre-industrial reference is mentioned elsewhere for the radiative-forcing calculations, please state what emissions are used for the pre-industrial case as well as for present-day/2010.

**p.9, line 8:** are biomass-burning emissions used for specific years in the simulation, or is this 2005–2014 period used to construct a climatology instead? If the latter, please consider the impact on the results (especially comparison to specific observations) of not capturing the real-year interannual variability in fire emissions.

**p.9, lines 11–13:** what does a conversion from "POM" to "particulate organic aerosols" mean? Why is such a factor necessary, and different for anthropogenic vs

biomass-burning? Or is that actually supposed to be referring to conversion between organic *carbon* (OC) and POM?

**p.10, line 4:** is it correct that only Terra is used for MODIS AOD, and not Aqua? If so, why? Both are mentioned below for the CERES_EBAF data. Also, are Dark Target or Deep Blue products used, or both?

**p.12, lines 11–23:** it's quite hard to identify the differences in these plots. Please include some kind of difference or statistical comparison plot (as is done in Fig. 6 for the aerosol fields).

**p.15, line 16:** please quote the actual MODIS AOT value here.

**p.16, line 23:** are't these are ground-based, not satellite, observations?

**p.16, line 25–p.17, line 3:** please consider the behaviour of the "coarsened" HRM data here, which should indicate whether the differences relate to the model resolution itself, or the fact that the observation sites are simply less representative of the coarser grid boxes.

**p.22, line 7:** why does underestimation of the emission inventory indicate the importance of using finer horizontal resolution? One does not obviously follow from the other.

**p.22, line 22:** please define what you mean by "secondary" and "tertiary" products.

**p.23, line 17:** "incredibly" is too strong a word here.

**p.23, line 24–p.24, line 1:** the description of LRM-macro is rather confusing here. Please introduce both LRM-macro and VLRM-macro properly in the main model-description section instead of suddenly throwing them in during the discussion section.

**p.26, line 1:** CALIPO $\longrightarrow$ CALIOP.

**p.26, lines 3–23:** Figure 17 suggests that the LRM consistently matches the SP2-observed profile shape better than the HRM, especially at higher altitudes (ARCTAS-B being the exception in absolute terms due to a bias although the *shape* is still a better match with the LRM). However, the discussion seems to gloss over this result which goes against the general theme of "HRM performs better". This needs to be identified in the text, and consideration given to how it fits with the rest of the results and conclusions.

**p.28, line 20–p.30, line 16:** the terminology (ARF, IARF etc.) used throughout this section is non-standard and confusing. Please consider using terminology like RFari, ERFaci etc. consistent with IPCC usage for clarity. Most of the values quoted in this section also need uncertainty estimates or some other method to enable an assessment of whether their differences are significant or not, otherwise they are not that meaningful.

**p.29, lines 13–14:** the second half of the sentence, "probably because the light-absorption..." doesn't make much sense. Please rephrase more clearly.

**p.30, lines 22–23:** this appear to quote variability of "meteorological fields" in $Tg\,yr^{-1}$, which doesn't make sense for any usual meteorological quantity. Please check what is actually meant here.

**p.31, lines 1–7 etc.:** please explain how the "difference caused by meteorological variabilities" is being quantified here as this is not at all clear. I think this is based on some measure of the variation over the three years of simulation, but please clarify explicitly *what* measure of this is being used, and also its associated uncertainty or confidence range given the small number of years.

**p.32, line 3:** 3 years is a long simulation at such resolution, however it's still very short for any study involving interannual climate variability, as discussed above. Al-

though this is one of the novel aspects of the work, describing it as "very pioneering" in this context is unjustified.

**p.33, line 6:** does "carbon" here refer to BC, OC, or both together?

**p.33, lines 18–19:** why should the lack of agreement with observations over China be attributed to an urban-density issue at this resolution that doesn't apply elsewhere, rather than to a possible error in the emissions inventory, for example?

**p.34, lines 2–6:** this terse list of "variabilities at relevant sites" isn't really appropriate for the presentation of conclusions. Please clarify what the actual conclusion is here instead.

**p.34, lines 13–24:** this needs to take account of the apparent better perfomance of the LRM with respect to SP2 profiles mentioned above.

**p.35, lines 6–7:** "negatively large[r]" is rather confusing. Does this mean "strongly/more negative"?

**p.35, lines 15–18:** again, this conclusion is rather unclear. What distinguishes the "relevant" fields from the "others"? Please rephrase and clarify.

**p.35, lines 23–25:** it's not clear what this means. What tuning exactly has been carried out here using the LRM for the HRM? Does this tuning have a bearing on the results presented in the manuscript?

---

## Referee Comment (RC2) · Anonymous Referee #2 · 18 May 2020

Review of "Global aerosol simulations using NICAM.16 on a 14-km grid spacing for a climate study: Improved and remaining issues relative to a lower-resolution model" by Goto et al. for publication in Global Model Development

The paper presents a fairly comprehensive overview of a pair of simulations performed with the NICAM.16 model with a coupled aerosol component based on SPRINTARS. Comparisons are made between a high-spatial resolution simulation performed at a horizontal14 km resolution and a lower resolution simulation at 56 km. In addition to presentation of such a high resolution simulation the main novel aspect of the paper

is that the simulation was run for several years, which is significant in terms of the comprehensiveness.

The paper is well written and fairly exhaustive in terms of comparisons made, but I am somewhat unsatisfied with the attribution aspects and suggest some needed modifications for publication. I do want to call out that I thought the presentation of the internal variabilities of the different resolutions was quite interesting and makes a case for bearing the costs of the higher resolution simulation, but it seems the conclusion of the paper is this is not necessary at the moment given the relative performances. Actually, I'm a little puzzled at what the overall conclusion is. Is it that the model performs well enough at the lower resolution to not justify the added cost? I wonder about specific case studies. Could the simulations be initialized to look at, say, a dust storm episode and dig more into the variability of such a case. For the overall conclusion, as stated what is recommended is that the tuning parameters associated with the aerosols in the LRM are applicable to those in the HRM. This is maybe true enough for dust and sea salt emissions, which are heavily tuned in most models. I wonder though if this is undermined by the apparent differences in the wet removal between the two runs. Further, I'm surprised the computational cost is only a factor of ten since the implied resolution differences suggest a factor of 16 more grid boxes in the high resolution run.

The discussion of the aerosol budget needs to be looked at more closely and given more discussion. In particular, I'm confused about what is shown in Table 2 especially with respect to black carbon components and for that matter the POM and sulfate. Differences in especially the lifetime of WIBC from nearly 9 to 6 days between the HRM and LRM runs are not explained by the budget given. Both runs have the same emissions, and the reported depositions for both runs are identical. So why is the lifetime different? Are there underlying mass conservation issues in the model that are not explored? WSBC has the same issue but the difference is less dramatic. For POM the lifetime also does not seem consistent with the loss and emissions numbers give. For sulfate I'm curious about the partitioning of aqueous production versus gas

production, which is not spelled out. My take on the paper is that most of what is different is due to wet processes, but the budget numbers don't clearly bear that out.

Most of my other comments are more minor or for clarification:

Could you please make explicit: are the aerosols radiatively coupled to the AGCM? Are they fully interactive with the cloud scheme?

Page 4, line 19: The NASA GEOS forecasting system is actually run at higher resolution in its operational forecasts with aerosols, and that system has been running for several years, although it is a quasi-operational system and so is not a single, consistent model experiment.

Page 7, line 24: 10 bins for dust is kind of a lot for this kind of model. You do not break down system costs, but how much compute could be saved by running half as many dust bins?

Page 8, line 2: "one modal" -> "monomodal"

Page 13, line 17-18: It is incorrectly stated that HRM is closer to data than LRM; the opposite appears to be true, or only at equator is HRM so close to data for COT. This is also stated on page 14 lines 9-10. I'm missing something here. Related, given the apparent discrepancies in the cloud fraction and COT I don't understand how the radiation parameter in 3E and 3F looks so good, and similarly for Figure 4.

Page 15, line 5: Please clarify use of word "global" here to refer to sum of diffuse+direct (i.e., could write: global (sum of diffuse+direct)). Later you refer to biasing of global averages (line 10) by the BSRN site locations. Where is the global averaged compared to BSRN even presented? I don't understand what you are trying to make a point of here.

Page 15, line 13 and Figure 5: The masking used here is curious since the simulations are AMIP runs untethered to actual events. Please explain the nature of the masking (presumably snow covered surfaces, although not sure about in Brazil). Another point

that bears some discussion about how the comparison is approached here: MODIS attempts to do a clear-sky aerosol retrieval, while presumably the model AOD is the all-sky AOD. In CTM-type runs where real events are simulated (and so, real clouds) it is found that by masking the model results with the MODIS cloud masks the AOD comparisons make more sense. You cannot do that here, although you could play games with excluding high cloud fraction grid cells from the comparisons. Or are you comparing a clear-sky calculated AOD (and how)? I suspect this is also relevant to the high AOD bias in the model over the southern ocean.

Page 15, line 21: over land AOD is "most uncertain" in MODIS products

page 17, line 15: strike "the most"

Page 18, lines 21-22: Here and elsewhere (like page 22, line 22) you implicate grid resolution as an explanation for differences but don't go far enough to say why. What process is different that you can point to?

Figure 15: What is going here with "macro"? Is this a separate model run? This isn't clear at all.

Page 23, line 22: the reference should be figure 15.

Page 32, line 13: The statement that the clouds are not underestimated with respect to MODIS is completely belied by Figure 3b and 3c. What am I not understanding here?

---

## Author Comment (AC1) · 12 Jun 2020

Referee #1

[C1-1] This manuscript presents one of the first multi-year simulations with an aerosol–climate model at a resolution approaching the convective "grey zone", in this case using the NICAM atmospheric model with the SPRINTARS aerosol scheme at a grid spacing of 14km. The study compares the meteorological fields, aerosol mass and optical thickness, and aerosol radiative forcing, between this configuration and a lower-resolution version of the same model. They additionally consider the output from the high-resolution model coarsened to the lower resolution to distinguish the effects of simulation vs data resolution. In many respects, this is a therefore a significant and illuminating study and very much appropriate for publication in GMD. However, there is one major point as well as a number of minor points which should be addressed before acceptance:

[A1-1] Thank you so much for reviewing our manuscript and give us much plentiful comments to improve our manuscript. According to your suggestions, we brushed up our manuscript. Through the revision, we moved and added some figures to the supplement including 9 tables and 16 figures. We also fixed errors in the calculation of global averages (and statistical parameters) used in wind speed, AOT and RF. In addition, we modified Figure 1 (typo: AOD→ AOT), Figure 2 (fixed errors of average values), Figure 3b (change: warm-topped cloud fraction → warm-topped COT), Figure 3c (change: warm-topped COT → cloud fraction in all clouds), Figure 4 (changed the name: Global Radiation→SSR), Figure 5 (added three panels obtained from MODIS/Aqua and fixed errors of average values), Figure 6 (typo: N=182, 186 and 272 → N=91, 83 and 136), Figure 7 (typo: Seasalt → Sea salt), Figure 9 (typo: Seasalt → Sea salt), Figure 15 (fixed errors of average values and added error bar) and Figure 16 (significantly modified, but the values used in this figure have not changed).

**Major points**
[C1-2] The main area of concern with this study relates to the ability of a 3-year simulation to adequately capture interannual variability. The very large (and zero-containing) confidence intervals quoted up front in the abstract, e.g. −91% to +18% and −49% to +223% suggest that this is simply not a long enough period to adequately constrain what are presented as headline results. If this is to be presented as a major advance in its own right compared to single-year high-resolution simulations, rather than merely a technical step towards a long enough simulation to produce robust interannual statistics, this needs to be clearly quantified and justified in the study. Alternatively, it might be more appropriate to submit results when a longer and more statistically-significant dataset is available, if the conclusions drawn from 3 years do not add appreciable confidence to those from single-year studies.

[A1-2] Thank you for your comments. We actually mentioned that "the differences in the simulated aerosol concentrations at polluted sites during polluted months between the HRM and LRM are estimated with medians of -23% (-63% to -2.5%) for BC, -4% (-91% to +18%) for sulfate and -1% (-49% to +223%) for the aerosol optical thickness (AOT)." The expression in the range has confused you. These ranges do not represent confidence intervals, but only the minimum and maximum values at various sites and months, as shown in Figure 13 in the original manuscript. Since the samples used in Figure 13 in the original manuscript were not randomly selected (just select the highest concentration for each region and month), the range from minimum to maximum represents their spatiotemporal variations. This range cannot be narrowed by the long-term integration of the model, since the spatiotemporal variations do not depend on the integration period. In addition, we thought that quantifying the difference between sites and time periods using the normalized mean bias (NMB) was more important than estimating the median values for different sites and time periods. Therefore, we newly calculated the NMB and modified the abstract as follows: "Additionally, the differences in the simulated aerosol concentrations at polluted sites during polluted months between the HRM and LRM are estimated with normalized mean biases of -19% for black carbon (BC), -5% for sulfate and -3% for the aerosol optical thickness (AOT)" Finally, as you pointed out, the advantage of the results of integration over 3 years instead of 1 year was not clearly quantified and justified in the abstract, although these were discussed in section 4.5. Therefore, we modified the contents in section 4.5 (please see the revised manuscript directly) and added the following conclusion to the abstract in the revised manuscript: "The analysis of interannual variability revealed that the difference in reproducibility of both sulfate and

carbonaceous aerosols at different horizontal resolution is greater than their interannual variability over 3 years, but those of dust and sea salt AOT and possibly clouds were the opposite."

[C1-3] There are also a very large number of figures in the paper, and I would strongly recommend condensing to a smaller number that illustrate the important results, and moving additional plots to the supplement if they do not contribute to the main conclusions.

[A1-3] Thanks for your suggestion. We considered this and decided to move Figure 6 (global distributions of the statistical metrics for AOT), Figure 8 (global distributions of the differences in the AOTs) and Figure 10 (global distributions of the differences in the surface mass concentrations) to supplement. Through the revisions, we prepared some additional figures to support the discussion and added them to supplement.

**Minor points**
[C1-4] p.2, line 7: while 56km is a fine resolution compared to most global aerosol–*climate* models, it is coarser than some of those used for operational global aerosol *forecasting*, as collected in the ICAP initiative, for example.

[A1-4] Yes, you are right. We first added "aerosol" to the abstract as follows: "For comparison, a NICAM with a 56-km grid spacing is also run as an LRM, although this horizontal resolution is still high among current global aerosol climate models." We also modified one sentence in section 1 as follows: "This 14-km horizontal grid boasts the finest resolution among all global chemistry models and is generally finer than most regional chemistry models (Galmarini et al., 2018). To effectively show the advantages in the simulated parameters related to aerosols in the HRM with a 14-km horizontal grid, we also executed an LRM with a 56-km horizontal grid, which is still finer than most global aerosol climate models (Myhre et al., 2013; Galmarini et al., 2018) but coarser than some of those used for operational global aerosol forecasting (Sessions et al., 2015)."

[Additional references]
Sessions, W. R., Reid, J. S., Benedetti, A., Colarco, P. R., da Silba, A., Lu, S., Sekiyama, T., Tanaka, T. Y., Baldasano, J. M., Basart, S., Brooks, M. E., Eck, T. F., Iredell, M., Hansen, J. A., Jorba, O. C., Juang, H. -M. H., Lynch, P., Morcrette, J. -J., Moorthi, S., Mulcahy, J., Pradhan, Y., Razinger, M., Sampson, C. B., Wang, J., and Westphal, D. L.: Development towards a global operational aerosol consensus: basic climatological characteristics of the International Cooperative for Aerosol Prediction Multi-Model Ensemble (ICAP-MME), Atmos. Chem. Phys., 15, 335-362, doi:10.5194/acp-15-335-2015.

[C1-5] p.2, lines 12–13: what does "column burden of the aerosol wet deposition" mean?

[A1-5] We changed from "column burden of the aerosol wet deposition" to "the aerosol wet deposition flux".

[C1-6] p.2, line 13: should this be "cloud to precipitation" rather than the other way around?

[A1-6] Thanks. This should be "cloud to precipitation".

[C1-7] p.2, lines 17–18: it is not clear why "differences. . . between the different horizontal grid spacings are not explained simply by the grid size" follows from the results quoted above as suggested. Is this related to the coarsened-HRM data, which hasn't been mentioned yet at all in the abstract?

[A1-7] Thanks for your comment. As you pointed out in [C1-21], the definition of secondary and tertiary products is unclear. After the revision of [A1-21], we modified it as follows: "These findings indicate that the impacts of higher horizontal grid spacings on model performance for secondary products such as sulfate, and complex products such as the AOT, are weaker than those for primary products, such as BC."

[C1-8] p.2, line 25–p.3, line 1: what are the confidence intervals on these ARF values, and therefore are the differences statistically significant?

[A1-8] The differences are statistically significant. To explain it, we calculated the confidence intervals using the annual average global grid data as a sample data. If we set at a global confidence intervals with a significance threshold of 95%, these ARF values with the uncertainties are estimated to be -0.293±0.001 Wm$^{-2}$ (RFari for HRM), -0.919±0.004 Wm$^{-2}$ (ERFaci for HRM), -0.239±0.002 Wm$^{-2}$ (RFari for LRM) and -1.101±0.013 Wm$^{-2}$ (ERFaci for LRM). The uncertainty of the HRM is smaller than that of the LRM becausee the number of samplings is 16 times higher (due to different number of grids). We incorporated these estimates to the abstract. We also added such uncertainties to section 4.4 in the revised manuscript.

[C1-9] p.3, lines4–6: it is commonly found that tuning parameters require quite different values at different model resolutions; therefore any suggestion that tuning using the LRM can be applicable to the HRM or vice versa requires proper justification.

[A1-9] Thank you for your comment. Indeed, the tuning parameters need different values for different model resolutions. We thought it over and decided to remove this comment from the revised manuscript. Alternatively, we modified the end of the abstract as follows: "Because one-tenth of the computer resources are required for the LRM (56-km grid) compared to the HRM (14-km grid), these findings in this study greatly help the modelers to decide whether the objectives will be archived by using a model with such higher resolution or not under the limitation of the available computational resources."

[C1-10] p.8, line 2: does this really mean monodisperse particles of a single fixed size as suggested, or the more usual unimodal size *distribution* with a fixedmode and width?

[A1-10] This means the monomodal size distribution with a fixed mode radius shown in the manuscript, but the width of the aerosols (and the size for sulfate) was not mentioned in the original manuscript. We modified it as follows: "…whereas those of BC, POM and sulfate are assumed to be monomodal with single fixed sizes (the radii are 0.1 μm for internally mixed BC with POM, 0.08 μm for secondary organic aerosol (SOA), 0.054 μm for external BC and 0.0695 μm for sulfate) and the width (1.8 for internally mixed BC with POM, 1.8 for SOA, 1.53 for external BC and 2.03 for sulfate). The sizes and widths are referred from Hess et al. (1998), Moteki et al. (2007) and Goto et al. (2008)."

[Additional references]
Hess, M., Koepke, P., and Schult, I.: Optical properties of aerosols and clouds: The software package OPAC, Bull. Am. Meteorol. Soc., 79, 831–844, doi:10.1175/1520-0477(1998)079<0831:OPOAAC>2.0.CO;2, 1998.
Moteki, N., Kondo, Y., Miyazaki, Y., Takegawa, N., Komazaki, Y., Kurata, G., Shirai, T., Blake, D. R., Miyakawa, T., and Koike, M.: Evolution of mixing state of black carbon particles: Aircraft measurements over the western Pacific in March 2004, Geophys. Res. Lett., 34, L11803, doi:10.1029/2006GL028943, 2007.
Goto, D., Takemura, T., and Nakajima, T.: Importance of global aerosol modeling including secondary organic aerosol formed from monoterpene, J. Geophys. Res., 113, D07205, doi:10.1029/2007JD009019, 2008.

[C1-11] p.8, line 10: what is meant by "flux of the ARI"?

[A1-11] According to your suggestion in C1-26, the terminology is also redefined. It is modified as follows: "RFari, the radiative forcing of the ARI".

[C1-12] p.9, lines 6–20: given that a pre-industrial reference is mentioned elsewhere for the radiative-forcing calculations, please state what emissions are used for the pre-industrial case as well as for present-day/2010.

[A1-12] In the preindustrial case, the emission fluxes from all anthropogenic sources as well as biomass burning are set at zero, whereas the others are identical to those used in present day. This sentence was inserted into the end of section 2.3 as follows: "In the preindustrial era to estimate both ERFari and ERFaci, the emission fluxes from anthropogenic sources and biomass burning for BC, POM and $SO_2$ are set to zero, but those from other sources, i.e., all natural sources, are identical to those used in the present era."

[C1-13] p.9, line 8: are biomass-burning emissions used for specific years in the simulation, or is this 2005–2014 period used to construct a climatology instead? If the latter, please consider the impact on the results (especially comparison to specific observations) of not capturing the real-year interannual variability in fire emissions.

[A1-13] We averaged the biomass-burning emission using 10 years (2005-2014 period) to construct a climatological field. Your concern is important and needs to be investigated, so we checked the interannual variability in emissions from biomass burning in GFED v4.1 and plotted some key figures as a supplementary figure (Figures S1 and S2). Using these figures, we would like to explain the possible impact of the climatological emissions from biomass burning. In the original manuscript, the model validation in specific years (less than 2 years) were performed;

(1) Surface BC, OC and sulfate mass concentrations by CAWNET in China for 2006-2007 (Fig 8 in the revised manuscript or Fig 11 in the original figure): In China, since aerosols emitted/formed from anthropogenic sources are much larger than those from the biomass burning (BB) in annual averages, the impacts of the BB climatological fields on results should be small.

(2) Surface sulfate mass concentrations at Barrow for 2008-2009 averages (Fig15f in the original figure): Figure S1(b) in supplement shows interannual variabilities of BC, POM and $SO_2$ in the Arctic (60°N-90°N). The 2008-2009 values for $SO_2$, normalized by the 2005-2014 average, showed a range of 0.5-1. This shows that the simulated sulfate can be overestimated by using the climatological values of the biomass burning. However, the simulated sulfate in this study at this site tends to be underestimated. Therefore, the use of climatological emissions of biomass burning does not affect the gap between the simulation and the observation.

(3) Vertical BC mass concentrations over the Pacific for 2009 (Fig 14 a-e in the revised manuscript and Fig 17a-e in the original manuscript): Supplemental Figure S1(a) shows the interannual variabilities of BC, POM and $SO_2$ for the global average. The 2009 amount of BC normalized to the average of 2005-2014 is 0.83, which is small compared to the differences in BC between the simulation and observation, especially at relatively low altitudes. However, in the Pacific Ocean, the effects of the biomass burning may be seen at relatively high altitudes. Some of the simulated BC concentrations at high altitudes are overestimated. Therefore, the use of climatological values for biomass burning may contribute to the gap between simulation and observation. This point was added to the revised manuscript in section 4.3.

(4) Vertical BC mass concentrations over East Asia for March-April 2012 (Fig 14f in the revised manuscript and Fig 17f in the original manuscript): Supplemental Figure S1(c) shows the interannual variabilities of the March-April mean BC in the domain of 90°E-150°E, 20°N-40°N. The 2012 amount of BC normalized to the average of 2005-2014 is 0.72, which can cause the overestimation of the simulation. However, the contribution of the biomass burning in this region is not so great. Furthermore, since the range of the observation results is up to twice, the simulated results in the current manuscript as well as the simulated results using the emission from biomass burning for a specific year are considered to be within the observation range (Fig17f).

(5) Vertical BC mass concentrations over the Arctic for March-April 2008 (Fig 14g in the revised manuscript and Fig 17g in the original manuscript): Supplemental Figure S1(c) also shows the interannual variabilities of the March-April mean BC in the domain of 60°E-90°E over the Arctic. The 2012 amount of BC normalized to the average of 2005-2014 is 3.67, which absolutely causes the overestimation of the simulation. This possibility was also mentioned in the original manuscript.

(6) Vertical BC mass concentrations over the Arctic for July-August 2008 (Fig 14h in the revised manuscript and Fig 17h in the original manuscript): Figure S1(c) in the supplement also shows the interannual variabilities of for July-August mean BC in the domain of 60°E-90°E over the Arctic. The

2012 amount of BC normalized to the average of 2005-2014 is 0.42, which can cause the overestimation of the simulation. However, the simulated BC concentrations are largely underestimated compared to the observation. Therefore, the gap between the simulation and the observation is not attributed from the use of climatological emission of the biomass burning.

(7) AOT and other parameters in 2012-2014 years: Supplemental Figure S2 illustrates global distribution of the differences in BC emission fluxes from biomass burning by GFEDv4 between the 2012-2014 and the 2005-2014 averages in Annual, January and July averages. The large differences are found in Siberia and Canada, where biomass burning often occurs in the summer season. In the Arctic average (60°E-90°E) shown in Figure S1(b), the 2012-2014 amount of AOT normalized to the average of 2005-2014 is approximately 1.5. This can influence the AOT validation using MODIS in these areas (Fig 5). Therefore, this point should be stated in section 3.2 in the revised manuscript as follows: "Over Canada and Siberia where biomass burning often occurs in summer, the NICAM-simulated AOT tends to be largely underestimated compared to the MODIS retrievals, partly due to use of climatological emission inventories for the biomass burning as pointed out in section 2.3 and Figure S2."

In section 2.3 in the revised manuscript, we modified several sentences as follows: "The emission amounts of total BC were 5.6 Tg yr$^{-1}$ from anthropogenic sources in 2010 according to the Hemispheric Transport of Air Pollution (HTAP)-v2.2 emission inventory (Janssen-Maenhout et al., 2015) and an climatological average of 1.8 Tg yr$^{-1}$ from biomass burning over 2005-2014 from the Global Fire Emission Database version 4 (GFEDv4; van der Werf et al., 2017). The interannual variabilities of the emission from the biomass burning are shown in the supplemental figures (Figures S1 and S2) to show impacts of the climatological averages on the results in a specific year, indicating that the impacts can be mostly ignored with only a few exceptions: AOT over Canada and Siberia in 2012-2014 average (mainly section 3.2) and BC mass concentrations over the Pacific and over the Arctic in March-April 2008 (section 4.3)."

[C1-14] p.9, lines 11–13: what does a conversion from "POM" to "particulate organic aerosols" mean? Why is such a factor necessary, and different for anthropogenic vs biomass-burning? Or is that actually supposed to be referring to conversion between organic *carbon* (OC) and POM?

[A1-14] Sorry for confusing you. It should be expressed as "a conversion from organic carbon (OC) to POM", so it was modified in the revised manuscript as follows: "Organic carbon (OC) is composed of both primary and secondary components; the emission amounts of primary OC were 20.3 Tg yr$^{-1}$ for anthropogenic sources (HTAP-v2.2) and 39.7 Tg yr$^{-1}$ from biomass burning (GFEDv4). These OC values are converted by multiplying the corresponding values for particulate organic matter (POM) by 1.6 for anthropogenic sources and 2.6 for biomass burning sources, whose values are used in several global aerosol models (Tsigaridis et al., 2014)."

[C1-15] p.10, line 4: is it correct that only Terra is used for MODIS AOD, and not Aqua? If so, why? Both are mentioned below for the CERES_EBAF data. Also, are Dark Target or Deep Blue products used, or both?

[A1-15] Thanks for your comment. According to your suggestion, we used both Terra and Aqua AOTs retrieved from the combination of both Dark Target (DT) and Deep Blue (DB), i.e., merged DT/DB method, in the revised manuscript. We modified Figure 5 (global distributions of AOT) in the revised manuscript. We found the global annual averages of the HRM-simulated AOT (0.175) and the LRM-simulated AOT (0.170) are within the differences between the MODIS/Aqua-retrieved AOT (0.163) and MODIS/Terra-retrieved AOT (0.184). However, as mentioned in the original manuscript, the regional distributions of both the HRM-simulated and LRM-simulated AOTs are different from those of the MODIS-retrieved AOTs. In the revised manuscript, we modified the first paragraph in section 3.2. Please see the revised manuscript. The CERES_EBAF data are estimated from the radiative fluxes using CERES sensors onboard both Terra and Aqua satellites to consider the diurnal variations of clouds. This point, the above points for MODIS AOT, and other points related to [A2-9] are modified in the revised manuscript as follows: "Table 1 summarizes the measurements used in this study for the model evaluation. Satellite observations greatly assist in better understanding the global model performance of optical properties. The Moderate Resolution Imaging Spectroradiometer (MODIS), a sensor on board the polar-orbiting satellites Terra and Aqua, observes both aerosols and clouds. The cloud products, i.e., COT only for warm-topped clouds and cloud

fraction (CF), and the aerosol products, i.e., AOT, in collection 6 are retrieved with a grid of 1°×1° by a NASA algorithm (Platnick et al., 2015). For clouds, the MODIS-retrieved COT has some positive biases especially in high latitudes, due to high solar zenith angle (Grosvenor and Wood, 2014; Lebsock and Su, 2014). For the AOT, the combination method of Dark Target (DT) and Deep Blue (DB) is used and can retrieve AOTs even over the desert areas (Levy et al., 2013), but it does not retrieve AOTs over high-albedo areas covered by snow and some specific areas, which include caatinga/cerrado surfaces over eastern Brazil in June-July-August and over Australia in all seasons (Sayer et al., 2014)." "The top-of-atmosphere (TOA) radiation fluxes, i.e., outgoing shortwave radiation (OSR), outgoing longwave radiation (OLR) and shortwave and longwave cloud radiative forcing (CRF), prepared by the CERES_EBAF_Ed2.8 level-3 product are obtained from a sensor of the Clouds and the Earth's Radiant Energy System (CERES) experiment onboard Terra and Aqua using 1°×1° grids by considering the diurnal variations of clouds (Loeb et al., 2009)."

Grosvenor, D., and Wood, R.: The effect of solar zenith angle on MODIS cloud optical and microphysical retrievals within marine liquid water clouds, Atmos. Chem. Phys., 14, 7291-7321. https://dx.doi.org/10.5194/acp-14-7291-2014, 2014.

Lebsock, M., and Su, J.: Application of active spaceborne remote sensing for understanding biases between passive cloud water path retrievals, J. Geophys. Res.: Atmospheres 119, 14, 8962-8979. https://dx.doi.org/10.1002/2014jd021568, 2014.

Levy, R. C., Mattoo, S., Munchak, L. A., Remer, L. A., Sayer, A. M., Patadia, F., and Hsu, N. C.: The Collection 6 MODIS aerosol products over land and ocean, Atmos. Meas. Tech., 6, 2989-3034, doi:10.5194/amt-6-2989-2013.

Sayer, A. M., Munchak, L. A., Hsu, N. C., Levy, R. C., Bettenhausen, C., and Jeong, M.-J.: MODIS Collection 6 aerosol products: Comparison between Aqua's e-Deep Blue, Dark Target, and "merged" data sets, and usage recommendations, J. Geophys. Res. Atmos., 119, 24, 13965-13989, doi:10.1002/2014JD022453, 2014.

[C1-16] p.12, lines 11–23: it's quite hard to identify the differences in these plots. Please include some kind of difference or statistical comparison plot (as is done in Fig. 6 for the aerosol fields).

[A1-16] Thanks for your comment. We plotted statistical comparisons in the supplement Figure S4 like Figure 6 in the original manuscript. In addition, we added global land/ocean averages to Figure 2 (we also fixed the global average values, as mentioned in the A1-1). Using these new figures, we largely modified the text about the wind comparison as follows: "Figure 2 illustrates the annual, January and July averages of the wind directions and speeds at a height of 10 m using the HRM-simulated, LRM-simulated and NCEP-reanalyzed winds. The global distribution of the statistical metrics, i.e., Pearson correlation coefficient (PCC), normalized mean bias (NMB) and root-mean-square error (RMSE), for the annually averaged wind speeds between the NICAM simulations and NCEP-reanalysis data are illustrated in the supplement (Figure S4). The global annual averages of both the HRM-simulated and LRM-simulated wind speeds (approximately 4.2 m s$^{-1}$; 4.169 m s$^{-1}$ for HRM and 4.242 m s$^{-1}$ for LRM) are slightly lower than those of the NCEP-reanalyzed wind speeds (4.487 m s$^{-1}$). The differences in the wind speed between the models and NCEP over land are smaller than those over ocean. The correlations between the NICAM (both the HRM and the LRM) and NCEP are moderate with a PCC of approximately 0.58 (0.577-0.580) for the global averages, whereas the differences in the PCC between land (0.582 for HRM and 0.590 for LRM) and ocean (0.576 for HRM and 0.577 for LRM) are small. The global annual average RMSEs between the NICAM simulations and NCEP are calculated to be approximately 1.45 m s$^{-1}$ (1.446 m s$^{-1}$ for HRM and 1.461 m s$^{-1}$ for LRM), approximately one-third of the global averages. The RMSEs are relatively high over the Southern Ocean (45°S-70°S), with values of at most 5.0 m s$^{-1}$. The NMBs are calculated to be -7.6% (HRM) and -5.8% (LRM) for the global averages. The RMSE and NMB over land are smaller than those over ocean. In Figures 2(a), 2(b) and 2(c), the spatial patterns of the HRM-simulated and LRM-simulated winds are generally in agreement with those obtained from the NCEP-reanalysis data, although there are some differences between the models and the NCEP reanalysis. The difference is predominantly caused by an underestimation over the Southern Ocean (within 45°S-70°S) with lower correlation (PCC), higher

uncertainty (RMSE) and more negative bias (NMB) than other areas (Figure S4). More negative NMB values over land are also found in both HRM and LRM (Figure S4(g) and S4(h)), even though the NCEP wind speeds are generally less than 3 m s$^{-1}$ over land. In January and July, the global averages of both the HRM-simulated and the LRM-simulated wind speeds are also lower than those of the NCEP-reanalyzed wind speeds by at most 10%. In January, the differences in the global averages of wind speed between the models and the NCEP reanalysis over land are larger than those over ocean, whereas in July those over land are smaller than those over ocean. In regions where sea salt is dominant over the Southern Ocean in both January and July and dust is the dominant over the Sahara in July, the differences in wind speeds between the HRM and LRM are relatively large. Although we expected the HRM-simulated wind speeds to be higher than the LRM-simulated wind speeds, our results do not confirm this behavior because the wind speed can be influenced by several complex mechanisms, such as clouds and radiation."

[C1-17] p.15, line 16: please quote the actual MODIS AOT value here.

[A1-17] We added it to the revised manuscript. As we mentioned in A1-1, the values of the global average for AOT are also corrected. Also, as we mentioned in A1-15, we used both MODIS/Terra and MODIS/Aqua in Figure 5. As a result, the explanation for Figure 5 was changed. Please see the first paragraph in section 3.2 in the revised manuscript.

[C1-18] p.16, line 23: are't these are ground-based, not satellite, observations?

[A1-18] Thanks for your comment. Right. We corrected it.

[C1-19] p.16, line 25–p.17, line 3: please consider the behaviour of the "coarsened" HRM data here, which should indicate whether the differences relate to the model resolution itself, or the fact that the observation sites are simply less representative of the coarser grid boxes.

[A1-19] This suggestion is very important. To clarify it, we evaluated the HRM results with 0.5° grid average by averaging 16 pixels of 0.125° grids (in the supplement of Table S5). As a result, we found that all statistical metrics in all cases were worse, for example, lower PCC (-0.014), higher RMSE (0.003) and larger NMB (-0.8%) in annual averages than those in the HRM results with the original grid (0.125°), but still higher than those in the LRM results. Therefore, in the revised manuscript we newly pointed out that "When the HRM-simulated AOTs with a grid converted to 0.5°×0.5° by averaging 16 pixels of 0.125° grids are evaluated using the in-situ measurements, the statistical metrics are worse than those in the original grids, i.e., lower PCC (-0.014), higher RMSE (0.003) and larger NMB (-0.8%) with regard to the annual averages (Table S5), but still higher than those in the LRM results. This finding suggests that the 0.5° grid is not fine enough to correspond to the representative value at the observation sites and the differences in AOTs between the HRM and LRM are not due to the grid conversion but the model resolution itself."
In addition, we also investigated the coarseness of the grid for the surface mass concentrations of sulfate, BC, POM, dust and sea salt. The same was found for sulfate, BC and POM only. This point was also added to the revised manuscript as follows: "As mentioned in the AOT comparison using in-situ measurements, although a 0.5° grid may not be fine enough to represent the observation sites, the difference between the HRM and LRM is not due to the analysis grid size but the model resolution itself (Table S6)."

[C1-20] p.22, line 7: why does underestimation of the emission inventory indicate the importance of using finer horizontal resolution? One does not obviously follow from the other.

[A1-20] In general, the emission inventory averaged on the coarse grid is smoothed from the original inventory, so strong emission fluxes from local points are not reflected on the coarse grid. Most of the smoothed sources are close to dense urban areas, as mentioned in revised summary and [A1-32]. Therefore, the underestimation of the BC emission inventory is partly caused by the smoothing effect, which requires a higher resolution simulation to use the emission inventory in the original high resolution grid. However,

this confused you, so we removed the original part and added it (related to A1-32) in the revised manuscript: "or the 14-km grid spacing is not sufficient to resolve such high concentrations in highly dense urban areas".

[C1-21] p.22, line 22: please define what you mean by "secondary" and "tertiary" products.

[A1-21] Surely it is not clear. Thanks. We added the following sentences to the relevant paragraph: "This finding suggests that the primary product, i.e., BC, is the most influenced by the grid size, but the secondary product, i.e., sulfate formed from oxidation of $SO_2$ (this is a primary product) and removal through precipitation, and the complex product, i.e., AOT comprising various aerosols including primary and secondary particles and being highly dependent on RH, are less influenced by the grid size. Therefore, the impacts of higher horizontal grid spacings on model performance for secondary products, such as sulfate, and complex products, such as AOT, are weaker than those for primary products, such as BC."

[C1-22] p.23, line 17: "incredibly" is too strong a word here.

[A1-22] Thanks for your comment. We changed it to "very".

[C1-23] p.23, line 24–p.24, line 1: the description of LRM-macro is rather confusing here. Please introduce both LRM-macro and VLRM-macro properly in the main model- description section instead of suddenly throwing them in during the discussion section.

[A1-23] Thanks for your suggestion. We removed this and inserted to the end of section 2.1 as follows: "In the sensitivity experiments for a comparison of aerosol mass concentrations over the Arctic in section 4.2, a cloud macrophysics module containing both a large-scale cloud condensation (Le Treut and Li, 1991) instead of the NSW6 cloud microphysics scheme and a cumulus parameterization (Chikira and Sugiyama, 2010) are adopted in the NICAM with 56-km and 220-km grid spacings. Hereafter, the sensitivity experiments are called low-resolution model (56-km) with the macrophysics module (LRM-macro) and very low-resolution model (220-km) with the macrophysics module (VLRM-macro). The VLRM-macro results have been evaluated against measurements in previous studies (Dai et al., 2014; Goto et al., 2015b; Dai et al., 2018)."

[C1-24] p.26, line 1: CALIPO −→ CALIOP.

[A1-24] Thanks for your correction.

[C1-25] p.26, lines 3–23: Figure 17 suggests that the LRM consistently matches the SP2- observed profile shape better than the HRM, especially at higher altitudes (ARCTAS-B being the exception in absolute terms due to a bias although the *shape* is still a better match with the LRM). However, the discussion seems to gloss over this result which goes against the general theme of "HRM performs better". This needs to be identified in the text, and consideration given to how it fits with the rest of the results and conclusions.

[A1-25] Thanks. According to your comments, we added the relevant sentences in section 4.3 as follows: "Above approximately 700 hPa, however, the differences between the HRM and LRM become large, which is consistent with the results of the comparison with CALIOP in Figure 13. Moreover, because the LRM-simulated BC concentrations are lower than those in the HRM, those in the LRM are closer to the observations than those in the HRM. As already mentioned in Table 2 in section 3.2, the differences in the BC concentrations between the HRM and LRM are caused by differences in the BC lifetime, especially for WIBC." The discussion of the different shapes, i.e., the overestimation of BC above 700 hPa, is supported by the discussion of the budgets shown in Table 2. Since the discussion using Table 2 was performed in section 3.2, we here added the possibility that BC lifetime is overestimated as one of the reasons to section 4.3 as follows: "The overestimation of the BC lifetime may be attributed to the underestimation of the wet deposition of WIBC in the HRM, overestimation of the convective mass flux above 500 hPa, which may

be improved by increasing the number of vertical layers in the model (Allen and Landuyt, 2014), possible overestimation of the climatological BC emission from biomass burning (Figure S1), and a lack of secondary aerosol activation by convective clouds and associated removal by precipitation (Yu et al., 2019)."

[C1-26] p.28, line 20–p.30, line 16: the terminology (ARF, IARF etc.) used throughout this section is non-standard and confusing. Please consider using terminology like RFari, ERFaci etc. consistent with IPCC usage for clarity. Most of the values quoted in this section also need uncertainty estimates or some other method to enable an assessment of whether their differences are significant or not, otherwise they are not that meaningful.

[A1-26] Sorry for confusing you. We modified it to use standard terminology such as RFari and RFaci in sections (i.e., section 2.2, 4.4 and 5) in the revised manuscript (not shown in detail here, so please see the revised manuscript).

[C1-27] p.29, lines 13–14: the second half of the sentence, "probably because the light- absorption… " doesn't make much sense. Please rephrase more clearly.

[A1-27] We modified it as follows: "which is probably because the light-absorption of carbonaceous aerosols is underestimated due to the underestimation of cloud scattering." We believe this is simpler and clearer.

[C1-28] p.30, lines 22–23: this appear to quote variability of "meteorological fields" in Tg yr$^{-1}$, which doesn't make sense for any usual meteorological quantity. Please check what is actually meant here.
[C1-29] p.31, lines 1–7 etc.: please explain how the "difference caused by meteorological variabilities" is being quantified here as this is not at all clear. I think this is based on some measure of the variation over the three years of simulation, but please clarify explicitly *what* measure of this is being used, and also its associated uncertainty or confidence range given the small number of years.

[A1-28 and A1-29] Sorry for confusing you. We intended to express the "meteorological difference" as the "interannual variability". In the revised manuscript, we used the word "interannual variability" and largely modified them in section 4.5 including the title "4.5 Interannual variability" and Figure 16 (Fig 19 in the original manuscript). We believe the revised Figure 16 helps readers understand what we want to show in this section and understand if the difference between HRM and LRM is greater or less than the maximum and minimum difference of each 1-year average over the 3 years. Details are not shown here, so please see the revised manuscript.

[C1-30] p.32, line 3: 3 years is a long simulation at such resolution, however it's still very short for any study involving interannual climate variability, as discussed above. Although this is one of the novel aspects of the work, describing it as "very pioneering" in this context is unjustified.

[A1-30] Thanks for your comment. We removed this.

[C1-31] p.33, line 6: does "carbon" here refer to BC, OC, or both together?

[A1-31] This includes all carbonaceous aerosols including WIBC, WSBC and POM. So, we changed it into "all carbonaceous aerosols".

[C1-32] p.33, lines18–19: why should the lack of agreement with observations over China be attributed to an urban-density issue at this resolution that doesn't apply elsewhere, rather than to a possible error in the emissions inventory, for example?

[A1-32] This comment is related to [A1-20]. We think there are two main possible reasons: (1) emission and (2) model resolution. Therefore, we added the possibility that the emission inventory is underestimated. As a result, this part was modified as follows: "This suggests that the emission inventory in China is underestimated or the 14-km grid spacing is not sufficient to resolve such high concentrations in highly dense urban areas."

[C1-33] p.34, lines 2–6: this terse list of "variabilities at relevant sites" isn't really appropriate for the presentation of conclusions. Please clarify what the actual conclusion is here instead.

[A1-33] Sorry, this expression is very unclear. As revised in the abstract, we modified it as follows: "At the polluted sites during polluted months, the differences in the simulated aerosol concentrations between the HRM and LRM are estimated with NMBs of -19% for surface BC, -5% for surface sulfate and -3% for AOT."

[C1-34] p.34, lines 13–24: this needs to take account of the apparent better performance of the LRM with respect to SP2 profiles mentioned above.

[A1-34] As we modified the relevant parts of your comment in [C1-25], we added the following sentence to the end of this paragraph in the revised manuscript: "This is also suggested by the validation of the vertical BC profile, which shows better performance of the LRM, whose lifetime of BC (4.97 days) is smaller than that in the HRM (6.37 days) and closer to the reference value (less than 5 days) from previous studies (Lund et al., 2018)."

[C1-35] p.35, lines 6–7: "negatively large[r]" is rather confusing. Does this mean "strongly/more negative"?

[A1-35] Thanks. We corrected "more negative" here.

[C1-36] p.35, lines 15–18: again, this conclusion is rather unclear. What distinguishes the "relevant" fields from the "others"? Please rephrase and clarify.

[A1-36] Thanks for your comment. After we revised Figure 16 in the revised manuscript (Figure 19 of the original manuscript) and section 4.5 in [A1-28 and A-29], we modified this part as follows: "The interannual variability is mainly reflect in the winds over land and RH, which cause relatively larger variabilities in dust (emission flux, column burden and AOT) and sea salt (mainly AOT). As a result, the total AOT and IRFari under clear-sky conditions and for longwave are more influenced by the interannual variabilities than those by the horizontal resolution. This suggests that the clouds are also significantly modulated by the interannual variabilities rather than the horizontal resolution. However, the results related to sulfate, POM and BC are strongly influenced by the horizontal resolution compared to the interannual variability, and discussions of the impacts of different horizontal grid spacings on these parameters can be facilitated using only a 1-year integration."

[C1-37] p.35, lines 23–25: it's not clear what this means. What tuning exactly has been carried out here using the LRM for the HRM? Does this tuning have a bearing on the results presented in the manuscript?

[A1-37] In our aerosol module, we used several tuning parameters but did not explicitly mention them in the original manuscript. Therefore, we added several comments to the revised manuscript related to the tuning parameters used in this study. First, main tuning parameters in our aerosol module are the interstitial fraction of aerosols to largely determine the amount of the wet deposition fluxes. This point was newly added to section 2.2 in the revised manuscript as follows: "Although aerosols coexist in both interstitial and inside clouds in the wet deposition, the interstitial fractions of aerosols are tuning parameters in the aerosol

module and set at 0.5 for dust, 0.2 for sea salt, 0.5 for all POM, 0.9 for external BC and 0.5 for sulfate in this study." All parameters used in the aerosol module in HRM are also used in those in LRM, so we added the following statement to section 2.2: "All parameters used in the HRM aerosol module also apply to those used in the LRM aerosol module." In addition, the dust emission function in our aerosol module uses some tuning parameters (Takemura et al., 2000), so that we added this point to section 2.3 in the revised manuscript: "
[revised manuscript text omitted]

---

## Author Comment (AC2) · 12 Jun 2020

Referee #2

[C2-1] Review of "Global aerosol simulations using NICAM.16 on a 14-km grid spacing for a climate study: Improved and remaining issues relative to a lower-resolution model" by Goto et al. for publication in Global Model Development.
The paper presents a fairly comprehensive overview of a pair of simulations performed with the NICAM.16 model with a coupled aerosol component based on SPRINTARS. Comparisons are made between a high-spatial resolution simulation performed at a horizontal 14 km resolution and a lower resolution simulation at 56 km. In addition to presentation of such a high resolution simulation the main novel aspect of the paper is that the simulation was run for several years, which is significant in terms of the comprehensiveness.

[A2-1] Thank you so much for reviewing our manuscript and give us much plentiful comments to improve our manuscript. According to your suggestions, we brushed up our manuscript. Through the revision, we moved and added some figures to the supplement including 9 tables and 16 figures. We also fixed errors in the calculation of global averages (and statistical parameters) used in wind speed, AOT and RF. In addition, we modified Figure 1 (typo: AOD→ AOT), Figure 2 (fixed errors of average values), Figure 3b (change: warm-topped cloud fraction → warm-topped COT), Figure 3c (change: warm-topped COT → cloud fraction in all clouds), Figure 4 (changed the name: Global Radiation→SSR), Figure 5 (added three panels obtained from MODIS/Aqua and fixed errors of average values), Figure 6 (typo: N=182, 186 and 272 → N=91, 83 and 136), Figure 7 (typo: Seasalt → Sea salt), Figure 9 (typo: Seasalt → Sea salt), Figure 15 (fixed errors of average values and added error bar) and Figure 16 (significantly modified, but the values used in this figure have not changed).

[C2-2] The paper is well written and fairly exhaustive in terms of comparisons made, but I am somewhat unsatisfied with the attribution aspects and suggest some needed modifications for publication. I do want to call out that I thought the presentation of the internal variabilities of the different resolutions was quite interesting and makes a case for bearing the costs of the higher resolution simulation, but it seems the conclusion of the paper is this is not necessary at the moment given the relative performances. Actually, I'm a little puzzled at what the overall conclusion is. Is it that the model performs well enough at the lower resolution to not justify the added cost? I wonder about specific case studies. Could the simulations be initialized to look at, say, a dust storm episode and dig more into the variability of such a case. For the overall conclusion, as stated what is recommended is that the tuning parameters associated with the aerosols in the LRM are applicable to those in the HRM. This is maybe true enough for dust and sea salt emissions, which are heavily tuned in most models. I wonder though if this is undermined by the apparent differences in the wet removal between the two runs.

[A2-2] Thank you for your comment. This manuscript shows many of the benefits of using HRM, including the obvious differences in the wet deposition, although some differences between HRM and LRM are small. As we mentioned in summary, a 14-km grid spacing (or finer) is needed to clearly resolve the scientific questions addressed in this study, when focusing on extreme phenomena related to clouds and precipitation and ACIs. In addition, the tuning parameters basically require different values for different model resolutions. We considered it again and decided to modify the relevant comment in the revised manuscript (the end of the abstract): "Because at least ten times of the computer resources are required for the HRM (14-km grid) compared to the LRM (56-km grid), these findings in this study help modelers decide whether the objectives can be achieved using such higher resolution or not under limitation of the available computational resources." In addition, the last part of summary was also slightly revised.

[C2-3] Further, I'm surprised the computational cost is only a factor of ten since the implied resolution differences suggest a factor of 16 more grid boxes in the high resolution run.

[A2-3] Yes, theoretically, the increase in HRM is more than 16 times the increase in LRM. However, for NICAM, the computational efficiency on the K computer used in this study tends to increase as continuous memory access increases. This is because that the K computer has relatively high memory performance. The Intel CPU does not exhibit such a tendency. Therefore, we added the following comment to the end of

summary: "As the computational cost is shown in Table S9 in the supplement, the computer resources required by the HRM are more than ten times higher (theoretically 16 times, but 10 times using the K computer, which is a high-performance computing resource with relatively high memory performance) than that required by the LRM when using the same supercomputer with the same number of processers." Table S9 shows in both [A2-7] and the modified supplement.

[C2-4] The discussion of the aerosol budget needs to be looked at more closely and given more discussion. In particular, I'm confused about what is shown in Table 2 especially with respect to black carbon components and for that matter the POM and sulfate. Differences in especially the lifetime of WIBC from nearly 9 to 6 days between the HRM and LRM runs are not explained by the budget given. Both runs have the same emissions, and the reported depositions for both runs are identical. So why is the lifetime different? Are there underlying mass conservation issues in the model that are not explored? WSBC has the same issue but the difference is less dramatic. For POM the lifetime also does not seem consistent with the loss and emissions numbers give. For sulfate I'm curious about the partitioning of aqueous production versus gas production, which is not spelled out. My take on the paper is that most of what is different is due to wet processes, but the budget numbers don't clearly bear that out.

[A2-4] Thank you for your comment. The lifetime is defined by Seinfend and Pandis (2006) as the global mean burden divided by the global mean emission amount, and by Textor et al. (2006) as the global mean burden divided by the global mean deposition amount. In usual global models including our model, the global annual amount of the emission flux is about the same as the amount of the total deposition flux. For POM and BC, the difference in the emission between HRM and LRM is almost zero, but not almost zero for other species: dust, sea salt and sulfate. Therefore, the difference in the atmospheric lifetime between the two experiments is highly dependent on the burden. As you know, the difference in the column burden between HRM and LRM is caused by that in the wet deposition. In addition, for some results, the differences in the global annual averages are small, but their spatial distributions are clearly different. Therefore, we newly added these figures to supplement as Figures S8-S15 and largely modified this part in the revised manuscript. For BC, we added the following comments to section 3.2: "The differences in the lifetimes between the HRM and LRM are large and estimated to be -22% for BC, -10% for WSBC and -33% for WSBC globally. The differences in their lifetimes or their column burdens between the HRM and LRM are mainly caused by wet deposition (Table 2 and Figures S13, S14 and S15). The wet deposition fluxes for aerosols in the HRM are generally smaller than those in the LRM, because the RPCW values in the HRM are smaller than those in the LRM. Therefore, over the outflow region, the wet deposition fluxes for BC, WSBC and WIBC in the HRM are smaller than those in the LRM. However, over land where the aerosol concentrations are large, the wet deposition fluxes in the HRM are larger than those in the LRM because the wet deposition fluxes are proportional to the aerosol concentrations (e.g., Seinfeld and Pandis, 2006). Near the source region of BC, for example in China, wet deposition in the HRM is larger than that in the LRM (Figure S13), mainly due to the larger concentrations, even though the RPCW values in the HRM are larger than those in the LRM." Also, we estimated sulfate production from both the gas and aqueous phases to be 16.8 TgS yr$^{-1}$ (gas, HRM), 16.1 TgS yr$^{-1}$ (gas, LRM), 41.7 TgS yr$^{-1}$ (aqueous, HRM) and 40.6 TgS yr$^{-1}$ (aqueous, LRM) and added them to the revised Table 2. The differences in the global annual averages are small, but regionally those in the aqueous phase are relatively large in East Asia as shown in Figure S8(e). The following comments were newly added to section 3.2 in the revised manuscript: "These differences between the HRM and LRM can be explained by the concentrations of both SO$_2$ and clouds, although the HRM-simulated clouds tend to be smaller than the LRM clouds, as shown in Figure 3. Therefore, these differences between the HRM and LRM are solely due to SO$_2$ concentration. This is also why the sulfate production rates through both the gas and aqueous phases in the HRM are greater than those in the LRM (Figure S8(d) and S8(e)). As a result, the HRM-simulated sulfate concentrations increase, but the wet deposition for sulfate in the HRM is larger than that in the LRM (Table 2 and Figure S11), as explained for BC that the wet deposition fluxes are proportional to the aerosol concentrations, even though the RPCW values in the HRM are larger than those in the LRM. In the end, the HRM-simulated sulfate in terms of the column burden is larger than in the LRM by 16% in a global average (Table 2), which mainly determines the differences in the lifetimes for sulfate."

Seinfeld, J. H. and Pandis, S. N.: Atmospheric Chemistry and Physics: From Air Pollution to Climate Change, 2nd ed., John Wiley and Sons, New York, USA, 2006.

[C2-5] Most of my other comments are more minor or for clarification:
Could you please make explicit: are the aerosols radiatively coupled to the AGCM? Are they fully interactive with the cloud scheme?

[A2-5] Yes, the aerosol module is radiatively coupled to AGCM as well as fully interactive with the cloud scheme. In section 2.1 in the revised manuscript, we newly added several words to section 2.1: "Cloud water and rain are fully interactive with cloud condensation nuclei (CCN), which are online calculated by the parameterization of Abdul-Razzak and Ghan (2000) as an indirect aerosol effect or aerosol-cloud interaction." In section 2.2, we modified it as follows: "To evaluate the aerosol direct effect in the NICAM, the instantaneous radiative forcing of the ARI (IRFari) is online calculated by the difference in the radiative fluxes with/without aerosol species in MSTRN-X."

[C2-6] Page 4, line 19: The NASA GEOS forecasting system is actually run at higher resolution in its operational forecasts with aerosols, and that system has been running for several years, although it is a quasi-operational system and so is not a single, consistent model experiment.

[A2-6] Thanks for the information. Surely, we know the existence of NASA GEOS forecasting system, but we couldn't find a clear reference to discuss the difference in the simulated results between the HRM and LRM. Therefore, we incorporated this point to section 1 as follows: "The NASA GEOS-5 aerosol forecasting system has been running at these high resolutions for several years (e.g., Gelaro et al., 2015), but to our knowledge, the published literature does not explain the difference in the simulated results between the HRM and LRM." In addition, we slightly modified the first paragraph of summary by changing 'all of these studies' into 'almost all of these studies' and by removing 'this work represents a very pioneering study' from the original manuscript.'

Gelaro, R., Putman, W. M., Pawson, S., Draper, C., Molod, A., Norris, P. M., Ott, L., Privé, N., Reale, O., Achuthavarier, D., Bosilovich, M., Buchard, V., Chao, W., Coy, L., Cullather, R., Silva, A., Darmenov, A., and Errico, R. M.: Evaluation of the 7-km GEOS-5 Nature Run, Tech. Rep. NASA / TM – 2014-104606, NASA, 2015.

[C2-7] Page 7, line 24: 10 bins for dust is kind of a lot for this kind of model. You do not break down system costs, but how much compute could be saved by running half as many dust bins?

[A2-7] Thanks for your comment. We added a table for the system cost in Table S9 in supplement. This table shows that the aerosol module contributes 10% of the total cost. The number of all tracers is 30, including 25 aerosols and their precursors, so by reducing half the dust bins, the number is 25. This reduction affects the cost of tracer advection, aerosol, surface flux and turbulence modules. In this case, the expected rate of the total cost is approximately 6% ({(402+296+182)*(30-25)/30}/2675=5.5% for HRM and {(56+27+16)*(30-25)/30}/278=5.9% for LRM).

Table S9: Calculation cost for HRM and LRM in units of second per 1-day integration. The values in parentheses represent contribution to the total cost.

|  | Process | HRM | LRM |
|---|---|---|---|
| Dynamics | Tracer advection | 56 (20%) | 402 (15%) |
| | Other | 61 (22%) | 325 (12%) |
| Physics | Microphysics | 56 (20%) | 1035 (39%) |
| | Radiation | 25 (9%) | 243 (9%) |
| | Aerosol | 27 (10%) | 296 (11%) |
| | Surface flux + Turbulence | 16 (6%) | 182 (7%) |
| | Other | 14 (5%) | 134 (5%) |
| Other | | 22 (8%) | 58 (2%) |
| Total | | 278 (100%) | 2675 (100%) |

[C2-8] Page 8, line 2: "one modal" -> "monomodal"

[A2-8] Thanks for your correction.

[C2-9] Page 13, line 17-18: It is incorrectly stated that HRM is closer to data than LRM; the opposite appears to be true, or only at equator is HRM so close to data for COT. This is also stated on page 14 lines 9-10. I'm missing something here. Related, given the apparent discrepancies in the cloud fraction and COT I don't understand how the radiation parameter in 3E and 3F looks so good, and similarly for Figure 4.

[A2-9] Thanks for your comments. First, it must be emphasized that the COT and CF shown in Fig 3b and 3c in the original manuscript were obtained from water-topped clouds. These were obtained under the limited conditions that the column pixels do not contain mixed-phase and ice-phase clouds. In contrast, SWCRF and OSR were obtained from CERES products, which accounts for all type of clouds and diurnal variations. Therefore, the CF and COT shown in Fig3b and 3c in the original manuscript were limited cases and they appeared to be inconsistent with the SWCRF and OSR results obtained by CERES. In the revised manuscript, we plotted the CF from all types of clouds (not only warm-topped clouds) and used both MODIS/Terra and MODIS/Aqua retrieved results. In addition, we added a comment on precipitation shown in Fig3a, because its result is very similar to that obtained from COT in total clouds in our model (not shown). The differences in CF (Fig3b in the revised manuscript) and COT (as precipitation Fig3a in the revised manuscript) between the models and observations, or between the HRM and LRM, are consistent with those in SWCRF, indicating that the performance of the HRM is better than that in LRM. To support the discussion, we added the global distributions of these parameters shown in Figure 3 to the supplemental Figure S5.

Although the validation of certain parameter, i.e., warm-topped COT, is not very common among GCM community, we left the COT only from warm-topped clouds in the revised manuscript for a better understanding of the model performance of clouds. As a result, the both HRM- and LRM-simulated COTs only from warm-topped clouds are underestimated compared to MODIS results (Figure 3c in the original manuscript and Figure 3b in the revised manuscript). The possible reasons for this bias are probably the underestimation of warm-topped COT itself in the NICAM and the overestimation of warm-topped COT in MODIS, especially in high latitudes (Grosvenor and Wood, 2014; Lebsock and Su, 2014). Another possible reason is that a bias of the simulated cloud height in the NICAM.

Therefore, the main message of this comparison is that the HRM performance of both SWCRF and OSR is better than the LRM performance, mainly due to the better performance of both CF and COT (or precipitation) in HRM, whereas the decomposed parameters such as COT in warm-topped clouds have not yet been adequately reproduced by both the HRM and LRM. These comments were reflected to the revised manuscript as follows: "The simulated clouds are also evaluated by zonal averages based on a comparison with satellite observations (MODIS/Terra and MODIS/Aqua). Because the cloud liquid water path (LWP) retrieved from satellites is highly uncertain (e.g., Lebsock and Su, 2014) and the simulated LWP is strongly correlated to precipitation (not shown), the comparison of the simulated precipitation shown in Figure 3(a) can be considered one of a validation of cloud parameters. In Figure 3(b), the warm-topped COTs are shown, and their global averages are estimated to be 7.9 (HRM), 10.2 (LRM), 15.1 (MODIS/Terra) and 15.0 (MODIS/Aqua). The distributions of both the HRM and LRM results are also not very close to the MODIS retrievals (Figure 3(b) and supplemental Figures S5(d,e,f)). The possible reasons are the underestimation of warm-topped COT itself in the NICAM and the overestimation of warm-topped COT in MODIS, especially in high latitudes (Grosvenor and Wood, 2014; Lebsock and Su, 2014). Another possible reason is that a bias of the simulated cloud height in the NICAM. The differences in warm-topped COT (Figure 3(b)) between thee HRM and LRM are consistent with those of precipitation (Figure 3(a) and supplemental Figures S5(d,e,f)). Figure 3(c) illustrates zonal averages of cloud fraction (CF) for all types of cloud (not just warm-topped clouds). The global averages of the CF are 0.63 (HRM), 0.59 (LRM), 0.74 (MODIS/Terra), and 0.75 (MODIS/Aqua). Both simulated CFs are underestimated compared to the MODIS result, but the HRM results tend to be closer to the MODIS results than the LRM results over low latitudes from 30°S to 30°N as well as high latitudes from 60°N to 90°N, whereas the LRM results tend to be closer to the MODIS results than the HRM results over higher latitudes from 90°S to 30°S. These differences can be found in the global distribution shown in supplemental Figures S5(g,h,i). Such discrepancy in clouds between global models, including the NICAM and the observations, can be found in

previous studies (e.g., Nam et al., 2012; Kodama et al, 2015); therefore, our case also includes some common problems."

Grosvenor, D., and Wood, R.: The effect of solar zenith angle on MODIS cloud optical and microphysical retrievals within marine liquid water clouds, Atmos. Chem. Phys., 14(14), 7291-7321. https://dx.doi.org/10.5194/acp-14-7291-2014, 2014.

Lebsock, M., and Su, J.: Application of active spaceborne remote sensing for understanding biases between passive cloud water path retrievals, J. Geophys. Res.: Atmospheres 119(14), 8962-8979. https://dx.doi.org/10.1002/2014jd021568, 2014.

[C2-10] Page 15, line 5: Please clarify use of word "global" here to refer to sum of diffuse+direct (i.e., could write: global (sum of diffuse+direct)). Later you refer to biasing of global averages (line 10) by the BSRN site locations. Where is the global averaged compared to BSRN even presented? I don't understand what you are trying to make a point of here.

[A2-10] Thanks you for comment. The global radiation confuses readers, so we changed 'global radiation' into 'surface solar radiation (SSR)', according to the terminology used in IPCC-AR5. In line 10 in the original manuscript, we would like to note that the differences in the global average obtained from BSRN results are not exactly consistent to those obtained from satellites, because the BSRN does not cover the ocean. This sentence was also modified in the revised manuscript as follows: "In addition, the BSRN sites do not cover the oceans, which cover a considerable proportion of the globe, thereby not exactly being consistent with the global average obtained from the satellites."

[C2-11] Page 15, line 13 and Figure 5: The masking used here is curious since the simulations are AMIP runs untethered to actual events. Please explain the nature of the masking (presumably snow covered surfaces, although not sure about in Brazil). Another point that bears some discussion about how the comparison is approached here: MODIS attempts to do a clear-sky aerosol retrieval, while presumably the model AOD is the all-sky AOD. In CTM-type runs where real events are simulated (and so, real clouds) it is found that by masking the model results with the MODIS cloud masks the AOD comparisons make more sense. You cannot do that here, although you could play games with excluding high cloud fraction grid cells from the comparisons. Or are you comparing a clear-sky calculated AOD (and how)? I suspect this is also relevant to the high AOD bias in the model over the southern ocean.

[A2-11] Regarding the first question, the reason we used masking in the AOT comparison was because the MODIS retrieved AOT was undefined in some areas. These areas are areas where the ground surface is covered with snow, and other areas are where AOT is negative due to errors in the retrieval method in some specific regions like Brazil. We used MODIS-retrieved AOT of collection 6 by the combination of Dark Target and Deep Blue methods of NASA algorithm (Levy et al., 2013; Platnick et al., 2015). The combined method can also retrieve AOT in the desert areas, but not in high albedo areas covered by snow and some specific areas, which are caatinga/cerrado surfaces over eastern Brazil in June-July-August and over Australia in all seasons (Sayer et al., 2014).

Levy, R. C., Mattoo, S., Munchak, L. A., Remer, L. A., Sayer, A. M., Patadia, F., and Hsu, N. C.: The Collection 6 MODIS aerosol products over land and ocean, Atmos. Meas. Tech., 6, 2989-3034, doi:10.5194/amt-6-2989-2013.
Sayer, A. M., Munchak, L. A., Hsu, N. C., Levy, R. C., Bettenhausen, C., and Jeong, M.-J.: MODIS Collection 6 aerosol products: Comparison between Aqua's e-Deep Blue, Dark Target, and "merged" data sets, and usage recommendations, J. Geophys. Res. Atmos., 119, 13965-13989, doi:10.1002/2014JD022453, 2014.

Regarding the second point, it was noted that the difference between the simulated AOT under all-sky and clear-sky conditions can cause a difference in AOT between our simulations and MODIS. Previous study (Dai et al., 2015) exactly showed this effect using the same model, NICAM, and the same parameter, AOT. It concluded that the differences between the simulated AOT under all-sky and clear-sky conditions were

within 10% at a global scale and at most 20% at a regional scale (we noted this point in section 2.4 in the original manuscript). Fig 3 in Dai et al. (2015) indicates that the absolute difference between the simulated AOT under all-sky and clear-sky conditions is less than 0.05 (70% in relative difference) over the Southern Ocean where one of the largest relative differences between the simulated AOT under all-sky and clear-sky conditions. Dai et al. (2015) also shows the temporal variation of the simulated AOT in various region including the Southern Ocean in Fig 5(g), which shows small differences in the AOT between all-sky and clear-sky conditions. Through this revision, we re-checked the simulated AOT under all-sky and clear-sky conditions, as shown in the supplement (Figure S3). Figure S3 showed that over the Southern Ocean, the difference between the simulated AOT under all-sky and clear-sky conditions were within 0.1 (absolute difference shown in panel c) and 30% (relative difference shown in panel d). However, the overestimation of the simulated AOT cannot be explained by the difference of the simulated AOT under all-sky and clear-sky conditions. Over the North Atlantic, this difference can partly explain the results in Figures 5 and S5 in the revised manuscript. Therefore, we modified/added the above points to section 2.4 and section 3.2 in the revised manuscript as follows: (section 2.4) "In addition, we compared the simulated AOT and aerosol extinction coefficient under all-sky conditions with the satellite-retrieved AOT and coefficients under clear-sky conditions because the differences in the simulated AOT between all-sky and clear-sky conditions are within 0.01 or 10% at a global scale (Figure S3), which is consistent with the previous study (Dai et al., 2015), but it should be noted that regionally the differences reach up to 0.1 over some regions, such as the North Atlantic (Figure S3)."(section 3.2)"As mentioned in section 2.4, because the NICAM-simulated AOT under the all-sky condition and the MODIS-retrieved AOT under the clear-sky condition are compared, the differences in the AOT between the NICAM and MODIS may be partly explained by the differences in the AOT between under the all-sky and clear-sky conditions, especially over the North Atlantic where the HRM-simulated AOT under the all-sky condition is larger than that under the clear-sky condition by up to 0.1 (Figure S3). Over the oceans within 45°S-70°S, however, there are no clear tendency, with a mixture of positive and negative biases (Figure S3)."

[Figure]

Figure S3: Global distributions of the 1-year averages of (a) the HRM-simulated AOT under the all-sky conditions, (b) the HRM-simulated AOT under the clear-sky conditions, (c) the absolute difference between the HRM-simulated AOT under the all-sky and clear-sky conditions, i.e., AOT(clear-sky) minus AOT (all-sky), and (d) the relative difference between the HRM-simulated AOT under the all-sky and clear-sky conditions, i.e., the ratio of the absolute difference to AOT (all-sky), with the original grid (0.125°×0.125°). The numbers shown in the upper-right corner in each panel represent the global averages.

[C2-12] Page 15, line 21: over land AOD is "most uncertain" in MODIS products

[A2-12] Thanks. We modified it.

[C2-13] page 17, line 15: strike "the most"

[A2-13] Thanks. We removed it (the sentences around this word were removed and Figure was moved to the supplement in the revised manuscript, because the information of Fig 8 used in the original manuscript was somewhat overlapped with that of Fig 9 in the original manuscript).

[C2-14] Page 18, lines 21-22: Here and elsewhere (like page 22, line 22) you implicate grid resolution as an explanation for differences but don't go far enough to say why. What process is different that you can point to?

[A2-14] The differences in the grid sizes cause the differences in the meteorological fields such as winds, which perturbate the emission fluxes of dust and sea salt. In the grid, vertical diffusion, horizontal transport, and cloud and precipitation fluxes are also perturbated. Therefore, the various processes are modulated and it is difficult to identify the process that is responsible for the differences in the aerosol distribution. Therefore, we described these effects as "grid resolution". In the revised manuscript, we modified as follows: "Therefore, the impact of the horizontal resolution (14-km and 56-km grid spacings), which determines the meteorological parameters including wind, vertical mixing, diffusion, clouds and precipitation fluxes, on dust is very small, but sea salt, sulfate and BC are strongly influenced."

[C2-15] Figure 15: What is going here with "macro"? Is this a separate model run? This isn't clear at all.

[A2-15] Yes, these are separate models using the different cloud module as sensitivity experiments in section 4.2. To clarify them, the description about "macro" was newly added to the end of section 2.1 as follows: "In the sensitivity experiments for a comparison of aerosol mass concentrations over the Arctic in section 4.2, a cloud macrophysics module containing both a large-scale cloud condensation (Le Treut and Li, 1991) instead of the NSW6 cloud microphysics scheme and a cumulus parameterization (Chikira and Sugiyama, 2010) are adopted in the NICAM with 56-km and 220-km grid spacings. Hereafter, the sensitivity experiments are called low-resolution model (56-km) with the macrophysics module (LRM-macro) and very low-resolution model (220-km) with the macrophysics module (VLRM-macro). The VLRM-macro results have been evaluated against measurements in previous studies (Dai et al., 2014; Goto et al., 2015b; Dai et al., 2018)."

[C2-16] Page 23, line 22: the reference should be figure 15.

[A2-16] This was a typo. We corrected it.

[C2-17] Page 32, line 13: The statement that the clouds are not underestimated with respect to MODIS is completely belied by Figure 3b and 3c. What am I not understanding here?

[A2-17] Figure 3b and 3c confuses you and the readers, sorry for this. We modified this statement as follows: "
[revised manuscript text omitted]